# Volatile and non-volatile pathogen cues shape host extracellular vesicles production in pre-infection response

Klaudia Kołodziejska [1,2], Agata Szczepańska [1], Satya Vadlamani [1,2,7], Ramakrishnan Ponath Sukumaran [1,2,7], Mariusz Radkiewicz[3], Henrik Bringmann [4], Nathalie Pujol [5], Wojciech Pokrzywa [6] & Michał Turek [1] ✉

In natural environments, animals encounter pathogen-derived chemicals long before infection occurs. How such anticipatory cues influence extracellular vesicle (EV) dynamics, which are central to immune regulation, intercellular communication, and stress responses, remains unknown. Using *Caenorhabditis elegans*, we show that pathogen-derived volatile and non-volatile compounds trigger distinct EV pathways through separate sensory and molecular mechanisms. Non-volatile secretome components, including the tripeptide Ile-Pro-Pro, activate immune-dependent EV production, whereas volatile metabolites elicit immunity-independent EV formation. Both responses require sensory input from ASK, ADL, and AWC neurons and converge on a neural circuit involving RMG, AIB, and AIA interneurons. GPCRs SRI-19, SRI-36/39, and SRR-6 mediate non-volatile responses, with SRR-6 acting in the intestine to regulate muscle EVs release. Notably, pre-exposure to pathogen volatiles enhances offspring survival during subsequent infection in an SRI-19–dependent manner, suggesting a predictive, intergenerational benefit of pathogen detection. In summary, our findings uncover that pathogen-derived chemical cues shape host EV production via specialized sensory circuits, revealing how animals anticipate infection and prime protective physiological responses.

Extracellular vesicles (EVs), including exosomes and microvesicles, are pivotal in the host immune response by mediating intercellular communication and regulating immune functions. Released by various cell types, EVs carry a diverse array of bioactive molecules such as antigens, signalling molecules, and genetic material, thus influencing immune cell behavior and activity[1,2]. Pathogen-derived EVs often contain pathogen-associated molecular patterns (PAMPs) that can activate immune responses[3], while host-derived EVs help modulate inflammation and immune activation, maintaining homeostasis and enhancing defense mechanisms against pathogens[4]. These roles for EVs have been well-documented during active infection, where pathogens are already invading the host. However, pathogens release both volatile

[1]Laboratory of Animal Molecular Physiology, Institute of Biochemistry and Biophysics, Polish Academy of Sciences, Warsaw, Poland. [2]Doctoral School of Molecular Biology and Biological Chemistry at Institute of Biochemistry and Biophysics, Polish Academy of Sciences, Warsaw, Poland. [3]Mass Spectrometry Facility, Institute of Biochemistry and Biophysics, Polish Academy of Sciences, Warsaw, Poland. [4]Biotechnology Center, Center for Molecular and Cellular Bioengineering, Technische Universität Dresden, Dresden, Germany. [5]Aix Marseille Univ, INSERM, CNRS, CIML, Turing Centre for Living Systems, 163 Avenue de Luminy, case 906, 13009 Marseille, France. [6]Laboratory of Protein Metabolism, International Institute of Molecular and Cell Biology in Warsaw, Warsaw, Poland. [7]These authors contributed equally: Satya Vadlamani, Ramakrishnan Ponath Sukumaran. ✉e-mail: m.turek@ibb.waw.pl

and non-volatile compounds, which can alert and prepare the host to their exposure[5–7]. Whether such compounds can modulate EV production prior to infection is not known.

The nematode *Caenorhabditis elegans* is a widely used model for studying host-pathogen interactions, particularly valuable for examining its responses to clinically significant human pathogens such as *Pseudomonas aeruginosa*, *Serratia marcescens*, and *Staphylococcus aureus*[8]. This organism offers several advantages due to its simplicity, genetic tractability, and well-characterized immune responses. *C. elegans* relies solely on innate immune pathways, making it an ideal organism for dissecting fundamental aspects of innate immunity without the confounding influence of an adaptive immune system[9,10]. Additionally, its short lifespan and large brood size enable high-throughput genetic and pharmacological screens[11]. The evolutionary conservation of many immune pathways between *C. elegans* and higher organisms further enhances its relevance for studying host-pathogen interactions[12,13]. Moreover, *C. elegans* uses specialised G protein-coupled receptor (GPCR)-mediated sensory circuits to detect environmental pathogens and orchestrate systemic outputs, including immune activation and behavioral avoidance, in response to microbial metabolites such as phenazine-1-carboxamide, pyochelin[14], and volatile 1-undecene[15].

*C. elegans* has also emerged as a valuable model for studying EV production. Due to its transparent body and well-characterized genetics, *C. elegans* allows tracking of EV production in real-time and investigation of the regulatory pathways involved[16]. One example of an evolutionarily conserved class of particularly large EVs that is investigated in worms are exophers. Exophers have been shown to be involved in waste removal and material exchange between tissues, with key functions in cellular stress responses, tissue maintenance, and reproduction[17–22]. Previous studies have demonstrated that exophers help protect neuronal activity in *C. elegans* by expelling toxic cellular debris[17], a process also observed in mammals, where mouse cardiomyocytes release exophers to discard defective mitochondria, maintaining metabolic balance and preventing inflammation[23]. In addition to waste disposal, we previously demonstrated that exophers facilitate inter-tissue material transfer to support reproductive fitness[18], under the control of social cues originating from other individuals in the population[22].

In this context, we aimed to investigate whether EV production in a host organism can be modulated by the mere presence of pathogens in the environment. Importantly, we focused on the response to pathogen volatile and non-volatile secretomes, allowing us to decouple this regulation from direct infection-related responses. Using exophers released by *C. elegans* body wall muscles as a model, we sought to explore the molecular mechanisms underlying pathogen secretomes-induced EV dynamics.

Our study uncovers a complex regulatory network governing exopher production in *C. elegans*, shaped by the type of interaction with pathogens. Pathogen infection or exposure to their secretomes increases exopher production. However, the molecular mechanisms regulating volatile metabolite-dependent and non-volatile secretome-dependent increases in EV production are distinctly controlled at sensory and neuronal levels. Moreover, the ability to distinguish between two types of signals triggers a context-specific physiological response, which may or may not involve immune activation, depending on the nature of the detected compounds. These findings provide new insights into the physiological mechanisms of EV production in *C. elegans* and suggest that similar mechanisms may be conserved across species.

## Results
### Infection with pathogens induces exopher production in hermaphrodite muscles
To test whether pathogen infection influences the production of EVs (exophers) in *C. elegans* (Fig. 1a), we exposed worms at adulthood day 1 to live *Pseudomonas aeruginosa* PA14 or *Serratia marcescens* Db10

bacteria for 3 hours. Next, we moved them back to plates seeded with a non-pathogenic *Escherichia coli* OP50 bacteria and, 24 hours later, we quantified the number of exophers released by body-wall muscle cells (BWMs) following a previously described methodology[24] (Fig. 1b). Our results show that worms exposed to both types of pathogenic bacteria produce significantly more exophers than worms that were kept on *E. coli* OP50 bacteria throughout the whole experiment (Fig. 1c).

Given that extracellular vesicles in other organisms are often part of the immune response[1,25], we next examined whether basal production of exophers as well as increased exopher production upon exposure to pathogenic bacteria depends on specific immune pathways. We tested various mutants deficient in key nematode immune response pathways[10,26], including the insulin pathway (*akt-1*, *daf-16*)[27], TGF-β pathway (*dbl-1*)[28], p38 MAPK pathway (*tir-1*, *pmk-1*)[29,30], and Intracellular Pathogen Response (IPR) pathway (*cul-6*)[31,32] (Fig. 1d). In some of the tested immune response pathways, we observed altered levels of exophergenesis, either decreased or increased, compared to wild-type worms. This was observed under standard growing conditions on non-pathogenic *E. coli* OP50 bacteria after knocking out at least one component of each pathway (Fig. 1e). Surprisingly, despite their altered basal levels, most of the tested mutants still exhibited a robust, statistically significant increase in exopher production upon pathogen exposure (Fig. 1f-h). The sole exception was the *cul-6* mutant with a compromised IPR pathway, which showed an increase in exopher production upon exposure to *P. aeruginosa* and a decrease in exopher production upon exposure to *S. marcescens* when compared to the control (Fig. 1i). Among the immune-deficient strains we tested, the magnitude of pathogen-induced exopher upregulation differed. Following *P. aeruginosa* exposure, the *dbl-1*, *pmk-1*, and *cul-6* mutants exhibited increases in exopher production comparable to wild-type levels, whereas *tir-1* mutants showed an even greater induction. In contrast, *daf-16* mutants displayed a significantly diminished response relative to wild-type, while *akt-1* mutants, defective in the negative regulator of DAF-16, exhibited an exaggerated increase (Supplementary Fig. 1a). Because AKT-1 inhibits DAF-16 in the insulin pathway, these data indicate that although exopher induction by pathogen presence is broadly robust, its amplitude is fine-tuned by DAF-16-dependent transcriptional programs as well as by p38 MAPK pathway.

A similar pattern emerged when worms were challenged with *S. marcescens*, with the exception that *tir-1* mutants responded like wild-type and *cul-6* mutants showed a significant reduction in exopher induction (Supplementary Fig. 1b).

Collectively, these results indicate that exopher upregulation in response to pathogens is not driven by a single canonical immune pathway but is primarily modulated through the insulin/DAF-16 axis, with other pathways contributing in a minor or compensatory manner.

Previous work has established that exopher biogenesis requires fertility and the presence of embryos in the uterus[18,33]. To determine whether pathogen exposure alone can bypass this requirement, we exposed sterile, FUdR-treated worms to *P. aeruginosa* and *S. marcescens*. In both cases, sterile animals failed to produce exophers (Supplementary Fig. 1c), demonstrating that fertility remains a prerequisite for pathogen-induced exophergenesis. When we examined embryo accumulation, *S. marcescens* exposure led to an increase in uterine embryos, whereas *P. aeruginosa*-exposed worms showed no change in embryo accumulation (Fig. 1j). Therefore, although both pathogens stimulate exopher production in fertile animals, only *S. marcescens* does so alongside embryo accumulation. This observation is critically important, as it shows that *P. aeruginosa* activates exophergenesis through a novel signalling pathway that is independent of the classic mechanism linked to embryo accumulation. This model allows for the isolation and study of the direct response to a pathogenic signal without the confounding influence of maternal signals. For this reason, we decided to focus our subsequent mechanistic analyses specifically on exophergenesis induced by *P. aeruginosa*.

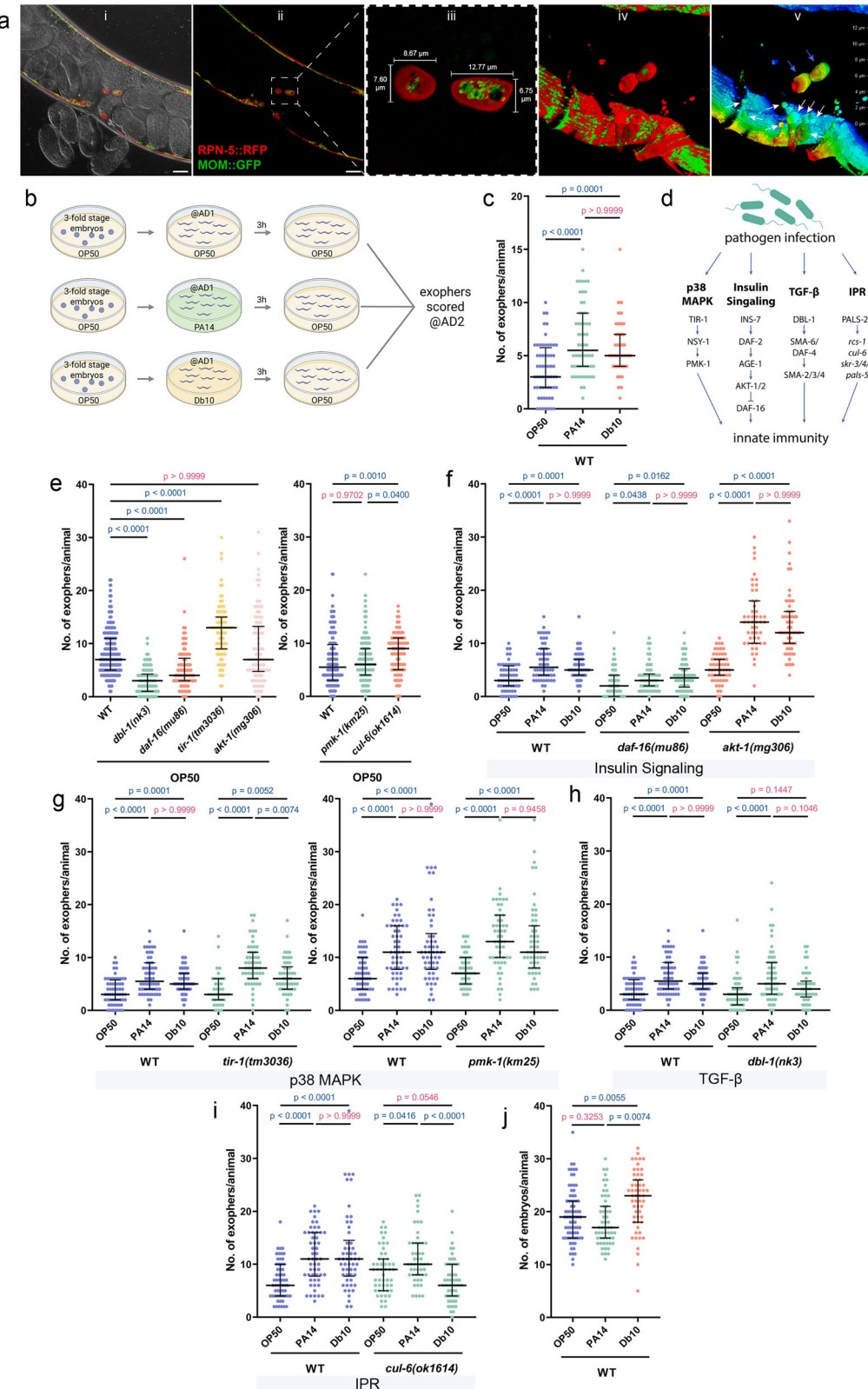

## Pathogen non-volatile and volatile secretomes trigger exopher production

Recent reports indicate that *C. elegans* can sense pathogen-derived volatile metabolites (airborne compounds emitted by the pathogen), which influence their physiology in a cell non-autonomous manner[15,34]. Previous studies have also shown that exposing *C. elegans* to the *P. aeruginosa* secretome (the full complement of water-soluble factors present in clarified culture supernatant including small metabolites, peptides, proteins, and EVs) leads to a robust immune response[35]. We hence aimed to determine whether the increased production of EVs in response to pathogens requires direct infection or can be triggered simply by exposure to volatile or soluble bacterial components.

We first investigated whether exposure to pathogen odours alone would be sufficient to induce exopher production in muscle cells. To

**Fig. 1 | Production of *C. elegans* muscle exophers is upregulated during pathogen infection. a** Confocal images of *C. elegans* expressing RPN-5::wrmScarlet (proteasome subunit, red) and MOM::GFP (mitochondrial outer membrane marker, green). Images were acquired using a Leica STELLARIS confocal microscope in LIGHTNING mode to enhance resolution. All five panels originate from the same animal and were generated from the same confocal z-stack. (i) Differential interference contrast (DIC) image aligned with a single confocal optical section, showing overall body morphology and the position of exophers. (ii) Overview fluorescence image of the mid-body region, illustrating the distribution of exophers along the body-wall muscles. (iii) Magnified view of two mitochondria-containing exophers from the region boxed in panel 2, with dimensions indicated to show their size variation and mitochondrial content. (iv) Three-dimensional surface rendering of the corresponding z-stack generated in LAS X 3D Viewer (Leica Microsystems), highlighting the morphology and spatial arrangement of exophers relative to surrounding tissues. (v) Depth-coded 3D heatmap representation of the same reconstruction. The color scale denotes the z-position of structures within the volume, with cooler colors (blue/cyan) indicating elements closer to the imaging plane and warmer colors (yellow/red) marking deeper regions. Blue arrows indicate mitochondria-containing exophers, and white arrows mark exophers emerging from body-wall muscles. This depth information emphasizes the extracellular location and three-dimensional topology of exophers. Scale bar: 20 μm. Representative of at least 23 animals from two independent replicates. **b** Experimental Setup: Synchronized AD1 worms were exposed to *Pseudomonas aeruginosa* (PA14) and *Serratia marcescens* (Db10), or to *Escherichia coli* (OP50) for 3 hours at 25 °C, followed by transfer to OP50-seeded plates. Exopher production was assessed on the second day of adulthood (AD2). Created in BioRender. Kolodziejska, K. (https://BioRender.com/xsyp3oe. **c** Short time infection with *P. aeruginosa* PA14 or *S. marcescens* Db10 leads to increased exopher production (*n* = 60, 58 and 59 worms (for respective columns), N = 2 independent experiments). **d** Investigated innate

immunity pathways: Various innate immunity pathways were assessed for their roles in regulating exopher production. **e** Worms deficient in key immune response pathways exhibited changes in exopher levels (*n* = 360, 90, 90, 90 and 90 worms (for respective columns), N = 3 independent experiments; *n* = 180, 120 and 91 worms (for respective columns), N = 3-4 independent experiments). **f–h** Infection-induced exopher production is not dependent on the (**f**) insulin signalling (*n* = 60, 58, 59, 56, 54, 58, 55, 50 and 60 worms (for respective columns), N = 2 independent experiments), (**g**) p38 MAPK (*n* = 60, 58, 59, 37, 55 and 50 worms (for respective columns), N = 2 independent experiments; *n* = 58, 54, 54, 54, 55 and 55 worms (for respective columns), N = 2 independent experiments), or (**h**) TGF-β pathways (*n* = 60, 58, 59, 54, 56 and 45 worms (for respective columns), N = 2 independent experiments). **i** Exopher production in response to *S. marcescens* Db10 infection is fully dependent on the IPR pathway, while production in response to *P. aeruginosa* PA14 infection is partially dependent on this pathway (*n* = 58, 54, 54, 51, 51 and 56 worms (for respective columns), N = 2 independent experiments). **j** Exposure to *S. marcescens* Db10, but not *P. aeruginosa* PA14, results in embryo accumulation in the uterus (*n* = 75, 55 and 55 worms (for respective columns), N = 3-4 independent experiments). Data information: The data shown in graphs (**f–i**) were collected together and share the same controls but are presented in separate panels to improve visualization and enhance clarity. Baseline differences between (**e, f–i**) arise from methodological variations which are detailed in the Materials and Methods section "Scoring of Muscle Exophers and Fluorescence Microscopy". Differences in basal exopher levels observed among wild-type controls are associated with the use of different transgenes for exopher visualization, as detailed in Supplementary Data 3. Data are presented as medians with interquartile ranges (**c, e–j**). Non-significant *p* values (*p* > 0.05) are highlighted in pink, while significant *p* values (*p* < 0.05) are shown in blue. Statistical significance was assessed using the Kruskal-Wallis test with Dunn's multiple comparisons test.

---

test this, we covered an NGM plate seeded with OP50 and containing age-synchronized L4-stage worms with another NGM plate containing *P. aeruginosa* and sealed both plates together with parafilm. Worms were incubated in this setup for 48 hours until they reached adulthood day 2, after which exophers were counted (Fig. 2a). Our results demonstrate that exposure to *P. aeruginosa* volatile metabolites indeed induces exopher production (Fig. 2b and c) and causes a slight embryo accumulation *in utero* (Fig. 2d).

Second, we aimed to determine whether the increased production of EVs in response to pathogens is triggered by exposure to the bacterial non-volatile secretome (Fig. 2e). As shown in Fig. 2f and g, direct infection with *P. aeruginosa* is not necessary; worms grown on plates with *P. aeruginosa* secretome also produced more exophers. Notably, this increase in exopher production is not associated with increased embryo accumulation in the worm's uterus (Fig. 2h). The fact that *P. aeruginosa* non-volatile secretome fails to increase embryo accumulation *in utero*, in contrast to the retention induced by volatile metabolites, suggests that non-volatile pathogen cues may actively suppress the embryo-accumulation response typically triggered by volatiles. This opposing outcome points to distinct downstream signalling branches that modulate reproductive physiology in response to different classes of pathogen-derived compounds. This also aligns with our observation that simultaneous exposure to volatile metabolites and the non-volatile secretome is non-additive for exopher production (Fig. 2i).

Consistent with previous results, FUdR-sterilized worms failed to upregulate exopher production in response to either *P. aeruginosa* volatile metabolites or non-volatile secretome (Supplementary Fig. 2a and d). This demonstrates that fertility is essential for exopher upregulation by both volatile and non-volatile pathogen cues. Moreover, because starvation alone is sufficient to increase exopher production[18,21], we monitored pharyngeal pumping to rule out reduced feeding as a confounding factor. We found no significant change in pumping rates following exposure to either pathogen volatile metabolites or non-volatile secretome (Supplementary Fig. 2b and e), confirming that the observed exopher upregulation is not driven by altered food intake.

We further explored whether neuronal exophers, previously shown to help maintain proteostasis in neurons, are similarly regulated by the pathogen-derived secretome. Using a touch-receptor neuron reporter strain[17], we exposed animals to either volatile metabolites or non-volatile *P. aeruginosa* secretome as described above and quantified neuronal exophers. Strikingly, we found that volatile metabolites, but not non-volatile secretome, significantly upregulate exopher formation in neurons (Supplementary Fig. 2c and f), indicating that the signalling machinery controlling EV production differs between muscle and neuronal cells.

## Volatile and non-volatile cues engage distinct immune-regulatory mechanisms

Next, we investigated whether the increased muscle exopher production in our new experimental setups is influenced by innate immunity signalling pathways. We first assessed each immune response mutant against its own unexposed baseline and then compared the magnitude of *P. aeruginosa*-induced changes to wild-type responses. On one hand, our data show that insulin, TGF-β, p38 MAPK, and IPR pathways are not individually required for the robust increase in exopher production triggered by volatile metabolites as each mutant mirrored wild-type induction (Fig. 3a and c), although we cannot exclude the possibility of redundant or compensatory interactions. However, there is a notable discrepancy in the role of these pathways when considering the non-volatile secretome-dependent increase in EV production. Specifically, while the insulin pathway does not by itself appear to be involved, the TGF-β, IPR, and partially p38 MAPK pathways are essential for the non-volatile secretome-dependent increase in exopher production (Fig. 3b and c). On the other hand, comparison of mutant versus wild-type animals showed that *pmk-1* and *akt-1* mutants displayed an even greater increase in exophers than wild-type upon volatile metabolites and non-volatile secretome exposure, while *dbl-1* mutants exhibited a blunted response to both types of compounds (Supplementary Fig. 3a and b). Together, these results indicate that *C. elegans*

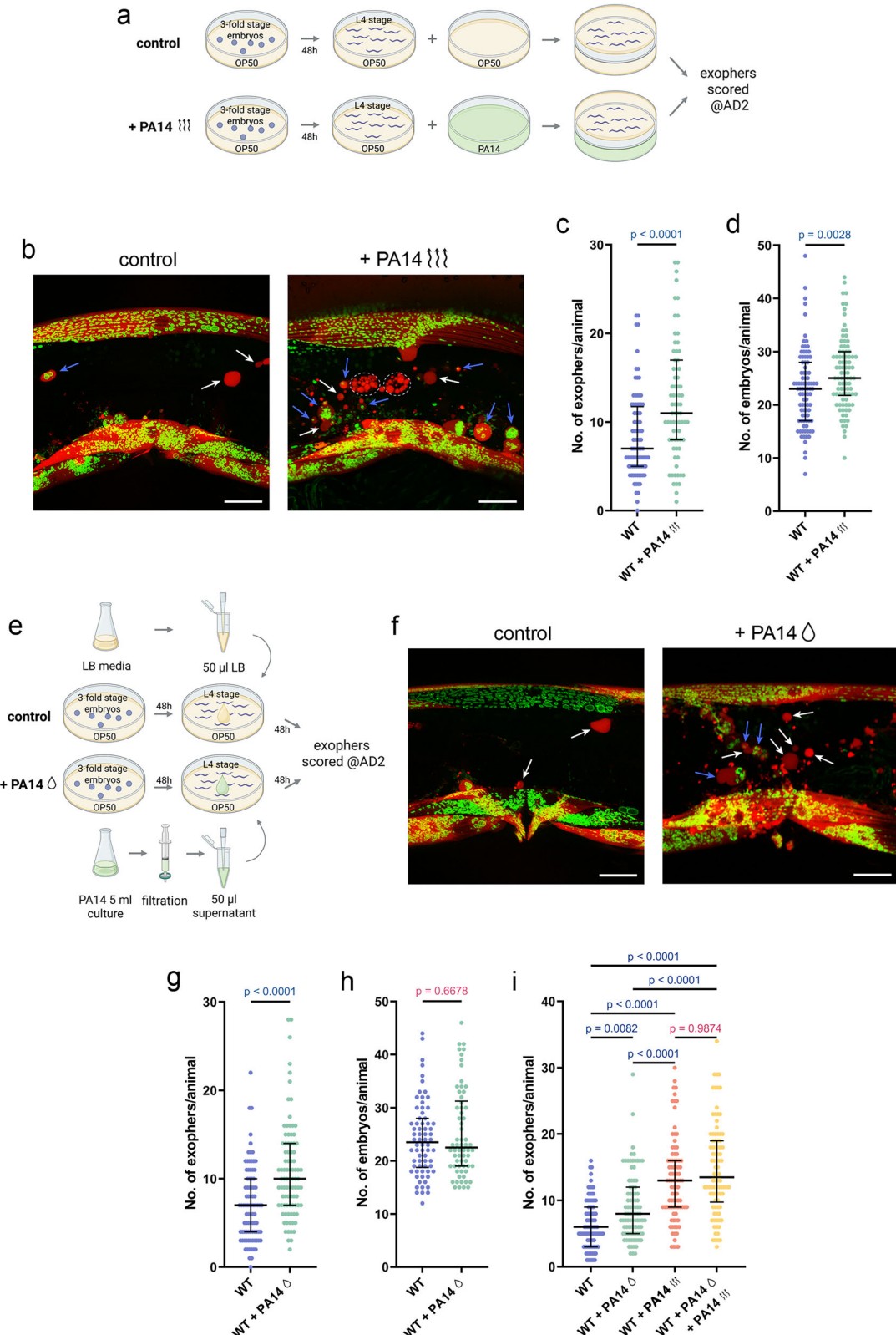

relies on distinct immune signalling mechanisms for EV production depending on the chemical nature of the stimulus: volatile *P. aeruginosa* metabolites trigger exophergenesis that is independent of multiple pathways yet quantitatively fine-tuned by insulin, TGF-β, and p38 MAPK signalling, whereas non-volatile secretome-driven upregulation requires TGF-β, IPR, and partially p38 MAPK pathways.

## Tripeptide Ile-Pro-Pro present in *P. aeruginosa* non-volatile secretome upregulates exopher production

During the course of our study, we observed a sudden decline in the reliability of our non-volatile secretome-based exopher upregulation assay. Troubleshooting traced this inconsistency to a switch in the membrane filters used for supernatant clarification. We confirmed this by testing five different filter types (two previously used and

**Fig. 2 | The production of muscle exophers in *C. elegans* is regulated by both volatile and non-volatile metabolites from pathogens. a** Experimental setup schematic: Synchronized L4-stage worms were exposed to PA14 volatile metabolites or OP50 for 48 hours, and exopher production was evaluated on the second day of adulthood (AD2). Created in BioRender. Kolodziejska, K. (https://BioRender.com/agnip73). **b** *C. elegans* hermaphrodites increase exopher production upon PA14 volatile metabolites, shown with red (RPN-5::wrmScarlet proteasome subunit) and green (mitochondrial outer membrane::GFP) fluorescence. Arrows: white - exopher, blue - exophers with mitochondria; dashed line circles - coelomocytes (identified based on their characteristic morphology and anatomical position within the worm body). Scale bars are 20 μm. **c** Exposure to PA14 volatile metabolites significantly increases exopher levels in worms (*n* = 88 and 77 worms (for respective columns), N = 3 independent experiments). **d** Volatile metabolites exposure causes mild embryo accumulation in the uterus (*n* = 90 and 90 worms (for respective columns), N = 3 independent experiments). **e** Experimental setup schematic: L4-stage worms were exposed to 50 μL of filtered pathogen supernatant, while controls received 50 μL of LB broth. Exophers were scored after 48 hours, at

AD2. Created in BioRender. Kolodziejska, K. (https://BioRender.com/slfj6mp). **f** *C. elegans* hermaphrodites increase exopher production upon PA14 non-volatile secretome, shown with red (RPN-5::wrmScarlet proteasome subunit) and green (mitochondrial outer membrane::GFP) fluorescence. Arrows: white - exopher, blue - exophers with mitochondria. Scale bars are 20 μm. **g** Exposure to PA14 non-volatile secretome results in increased exopher production (*n* = 90 and 90 worms (for respective columns), N = 3 independent experiments). **h** PA14 non-volatile secretome exposure does not induce embryo accumulation (*n* = 70 and 67 worms (for respective columns), N = 3 independent experiments). **i** Exposure to PA14 volatile metabolites induces higher exopher production compared to PA14 non-volatile secretome. No additive effect is observed when worms are exposed to both volatile and non-volatile secretomes (*n* = 90 worms (for each column), N = 3 independent experiments). Data information: Data are presented as medians with interquartile ranges. Statistical analyses were performed using the two-tailed Mann-Whitney test (**c**, **d**, **g**, **h**) and the Kruskal-Wallis test with Dunn's multiple comparisons test (**i**), with non-significant *p* values ($p > 0.05$) in pink and significant *p* values ($p < 0.05$) in blue.

three additional ones) and found that only two permitted robust exopher induction (Fig. 4a). We leveraged this filter sensitivity as a bioassay to pinpoint the active component(s) in *P. aeruginosa* supernatant (Fig. 4b). Using liquid chromatography–mass spectrometry, we compared the metabolite profiles of exopher-inducing versus non-inducing filtrates from the same bacterial liquid culture and identified 65 statistically significant, differentially abundant compounds (Supplementary Data 1). Interestingly, the active supernatant was enriched in several small peptides, dipeptides, tripeptides, and tetrapeptides, all of which contained at least one proline residue. Moreover, inspired by previous work demonstrating that N,N-dimethyltryptophan–containing dipeptide can modulate *C. elegans* physiology, we selected Ile-Pro-Pro (Fig. 4c), a tripeptide known to be produced by lactic acid bacteria[36], for functional testing.

When synthetic Ile-Pro-Pro was added to standard NGM medium, exopher production increased in a dose-dependent manner (Fig. 4d). At 50 μM, we observed a 17% rise over vehicle control, rising to 29% increase at 100 μM. These results identify the tripeptide Ile-Pro-Pro as a specific, pathogen-associated molecule capable of inducing exopher production in the host. While likely acting in concert with other secreted factors, Ile-Pro-Pro represents a key molecular signal through which *C. elegans* can sense the metabolic state of surrounding bacteria.

## Multiple types of sensory neurons regulate exopher production in response to pathogen secretome

Olfactory neurons in *C. elegans* detect numerous environmental signals that influence various aspects of the worm's life, including behavior, physiology, and reproduction[37]. Additionally, our recent findings have shown that olfactory neurons regulate pheromone-dependent increases in EV production[22]. To identify the sensory neurons responsible for regulating the pathogen non-volatile secretome-dependent and volatile metabolites-dependent increases in EV production, we utilized nematodes with genetic ablations of sensory neurons known to mediate pathogen recognition[38–40] and in exopher regulation[22]. Systematic analysis using animals carrying established genetic ablations of specific sensory neurons[41–45], revealed that removal of ADL, ASK, or AWC neurons completely abolished exopher upregulation induced by both volatile metabolites and the non-volatile secretome (Fig. 5a-c). Conversely, the removal of AQR/PQR/URX, which we previously identified as regulators of exophergenesis[22], had no effect in either assay, similar to ASJ ablation (Fig. 5a-c). Interestingly, we observed discrepancies among ASI, ASH, and AWB neurons. While ASI, ASH, and AWB neurons were crucial for the non-volatile secretome-induced increase, they do not play a role in the volatile metabolites-dependent increase in EV production (Fig. 5a-c). Notably, ablation of ADL or ASK also reduced baseline exopher levels (Fig. 5a and c), consistent with our previous work showing that these neurons

contribute to exophergenesis regulation at the basal level[22]. Finally, when comparing each ablation to wild type under non-volatile secretome exposure, all sensory-neuron mutants except ASI and AQR/PQR/URX showed a significantly attenuated exopher response, with ASK and ADL mutants exhibiting the largest reductions, whereas AQR/PQR/URX-ablated and ASI-ablated worms produced more exophers than wild type and the same number as wild type, respectively (Supplementary Fig. 4b). For volatile metabolites, AQR/PQR/URX and ASI ablations likewise enhanced exopher induction, AWB ablation had no measurable effect, and the response was reduced in all other neuron-ablated strains (Supplementary Fig. 4a). Altogether, these findings show that multiple types of sensory neurons regulate exopher production in response to pathogen-produced compounds in a context-dependent manner, with ADL, ASK, and AWC neurons playing pivotal roles.

## A hierarchical neural circuit regulates exopher production in response to pathogen secretomes

To map the downstream interneuron circuit, we took a dual approach. A literature search for a central integrator of the previously identified sensory neurons pointed to the RMG interneuron, a known "hub-and-spoke" network center[46]. Separately, a search of the *C. elegans* connectome for downstream partners highlighted the AIA and AIB interneurons[47]. Notably, AIA and AIB, along with the AIB-specific neuropeptide receptor NPR-9, have previously been implicated in regulating *C. elegans* immunity[48,49], making them prime candidates for mediating this physiological response.

Our functional data confirmed the critical role of the RMG hub. Genetic ablation of RMG not only reduced basal exophergenesis but also completely abolished the induced response to both volatile and non-volatile pathogen cues (Fig. 6a–c, and Supplementary Fig. 5a and b), establishing it as an essential, central integrator.

Next, we examined the roles of the candidate processing interneurons. Ablation of AIA revealed a striking, context-dependent function. While AIA removal caused a modest, non-significant reduction in basal exopher levels (Fig. 6d), AIA-deficient worms still responded to volatile metabolites with a significant but attenuated increase in exopher production (Fig. 6e, and Supplementary Fig. 5a). In contrast, exposure to the non-volatile secretome failed to elicit any induction (Fig. 6f, and Supplementary Fig. 5b). These results indicate that AIA is essential for non-volatile secretome-driven exophergenesis and plays a modulatory role in the volatile metabolites-mediated response (Fig. 6k).

Genetic ablation of AIB did not affect basal exopher production, whereas loss of the inhibitory receptor NPR-9, a condition that is known to activate AIB[48], reduced baseline exopher levels (Fig. 6g). Consistent with the results from the *npr-9* mutant, optogenetic

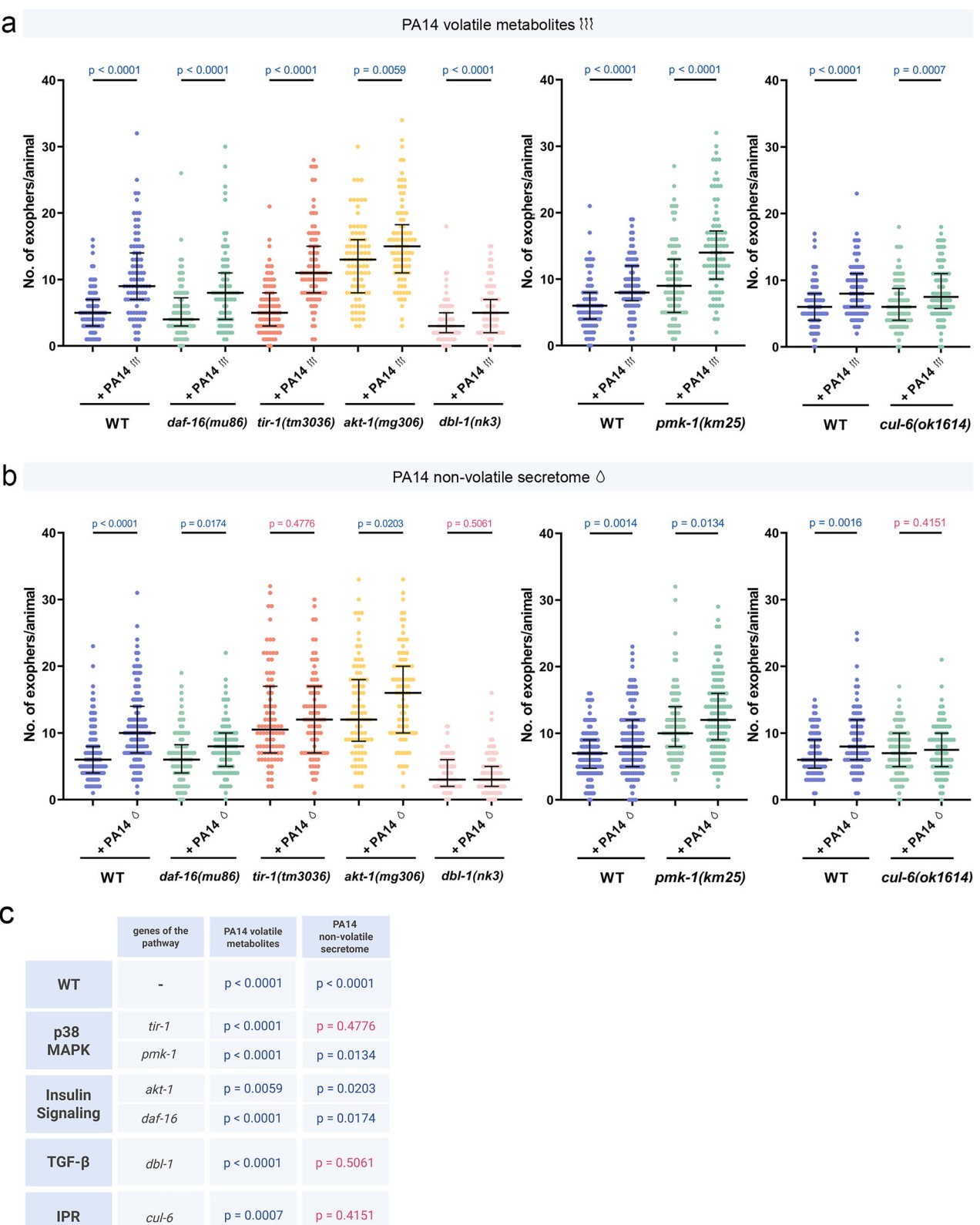

activation of AIB for 4 hours via ReaChR[50] also significantly decreased exopher production (Fig. 6h). Upon exposing worms to *P. aeruginosa* secretomes, the removal of the AIB neurons or NPR-9 receptor significantly reduced the increased exophergenesis induced by pathogen volatile metabolites (Fig. 6i, Supplementary Fig. 5a) and non-volatile secretome (Fig. 6j, and Supplementary Fig. 5b), with a more pronounced effect for volatile metabolites exposure. These results

suggest that the AIB interneurons and NPR-9 receptor are essential for mediating exopher production in response to pathogen signals (Fig. 6k).

In summary, our findings delineate a multi-layered circuit that processes pathogen threat information (Fig. 6k). Increased exopher production due to volatile metabolites sensing relies primarily on the ASK, ADL, and AWC sensory neurons, which signal through the AIB and

**Fig. 3 | Exopher production in response to pathogen volatile metabolites and non-volatile secretome is governed by distinct regulatory mechanisms.**
**a** Increased exopher production after exposure to PA14 volatile metabolites does not depend on innate immunity pathways ($n = 90, 90, 90, 80, 90, 90, 90, 90, 90$ and 90 worms (for respective columns), N = 3 independent experiments; $n = 80, 90,$ 90 and 90 worms (for respective columns), N = 3 independent experiments; $n = 88,$ 87, 88 and 90 worms (for respective columns), N = 3 independent experiments).
**b** Increased exopher production after exposure to PA14 non-volatile secretome depends on p38, TGF-β, and IPR pathways ($n = 120, 120, 90, 90, 90, 90, 90, 90, 90$ and 90 worms (for respective columns), N = 3-4 independent experiments; $n = 122$ worms (for each column), N = 4 independent experiments; $n = 90, 85, 90$ and 90 worms (for respective columns), N = 3 independent experiments). **c** Summary of experimental results: A table summarizing the experimental findings, including $p$ values. Created in BioRender. Kolodziejska, K. (https://BioRender.com/adabq21). Data information: Differences in basal exopher levels observed among wild-type controls are associated with the use of different transgenes for exopher visualization, as detailed in Supplementary Data 3. Data are presented as medians with interquartile ranges. Statistical analyses were performed using the two-tailed Mann-Whitney test, with non-significant $p$ values ($p > 0.05$) in pink and significant $p$ values ($p < 0.05$) in blue.

RMG interneurons, with AIA interneurons acting as a modulator (Fig. 6l). In contrast, increased exopher production due to non-volatile secretome sensing engages an expanded circuit that includes ASI, AWB, and ASH inputs, alongside ASK, ADL, and AWC, converging on the same AIB and RMG core, but with stronger AIA involvement (Fig. 6m).

## Identification and functional characterization of GPCRs mediating pathogen-induced exophergenesis

G-protein-coupled receptors (GPCRs) are essential mediators that enable cells to interact with their environment by transmitting signals. In both vertebrates and invertebrates, GPCRs play a key role in coordinating immune responses[51,52]. Because exopher induction is triggered by chemical cues from pathogenic bacteria, we hypothesised that specific GPCRs act as sensors linking pathogen-derived signals to neuronal circuits controlling exophergenesis. To identify such receptors, we adopted a transcriptomic approach combining published datasets and new RNA-sequencing experiments.

First, we analysed published transcriptomics data from worms exposed to *P. aeruginosa* compared to worms grown on *E. coli* OP50[53]. Among the most significantly upregulated GPCRs were *sri-36* and *sri-39*, with fold changes of 480 and 79.5, respectively. According to a study by Vidal et al. [54]., these GPCRs are expressed in ASK and ASI cells, which we have shown to play an important role in exopher upregulation (Fig. 5a-b). Additionally, our prior RNA sequencing, which identified the GPCR *str-173* as a regulator of socially-mediated exopher production, also ranked *sri-36* and *sri-39* among the top candidates associated with exopher regulation[22], further justifying their selection for validation.

To broaden this search and identify receptors associated with natural variation in exopher output, we conducted RNA sequencing on age-synchronized wild-type worms that exhibited either high (more than 20) or low (fewer than 2) number of exophers on adulthood day 2. A similar strategy, utilizing gene expression heterogeneity within isogenic populations, was recently employed to study the processes that generate lifespan variation[55]. The results of this RNA sequencing show that worms producing a high number of exophers differ significantly at the transcriptomic level from worms producing low number of exophers (Fig. 7a). Specifically, 609 genes were upregulated and 665 genes were downregulated (p adj <0.05; -1> log2 fold change >1) (Fig. 7b and Supplementary Data 2). GO enrichment analysis revealed upregulation of "cilium assembly" (Fig. 7d), consistent with the central role of ciliated neurons in exopher regulation[22]. Among the significantly altered genes were three GPCRs; *str-173*, *srr-6*, and *sri-19*, whose expression correlated with exopher number (Fig. 7c). As *str-173* had been validated previously as a pheromone-responsive regulator[22], *srr-6* and *sri-19* were selected for functional analysis alongside *sri-36* and *sri-39*, yielding four prime receptor candidates for experimental validation.

To examine their functions, we generated CRISPR/Cas9 mutants for each locus (Supplementary Fig. 6). Because *sri-36* and *sri-39* are separated by only 1.9 kb, with one gene situated between them, they were deleted together to produce an *sri-36/39* mutant. Under baseline conditions, exopher levels were similar to wild type in *sri-19* and *sri-36/39* mutants (Fig. 8a and e), while it was mildly upregulated in *srr-6* mutant (Fig. 8i). Upon exposure to pathogen cues, a clear functional divergence emerged. All receptor mutants exhibited a statistically significant increase in exopher production following exposure to volatile metabolites (Fig. 8b, f, j). The responses of *sri-36/39* and *srr-6* mutants were comparable to wild type (Supplementary Fig. 7c, e), whereas the *sri-19* mutants showed a significantly attenuated response (Supplementary Fig. 7a). In stark contrast, the response to the non-volatile secretome was completely suppressed in all tested mutants (*sri-19*, *sri-36/39*, and *srr-6*), indicating they are all essential for this pathway (Fig. 8c, g, k, Supplementary Fig. 7b, d, f). Underscoring its central role, the *sri-19* mutant also failed to upregulate exopher production upon exposure to live *P. aeruginosa* bacteria (Supplementary Fig. 7g).

Expression analysis provided the crucial link between these molecular players and the previously identified circuits. The wrmScarlet CRISPR/Cas9-mediated transcriptional knock-in of the *sri-19* gene revealed its expression in ADL and PHA neurons (Fig. 8d, and Supplementary Fig. 7h, i). Furthermore, transcriptional reporters using fluorescent proteins for the *sri-36* and *sri-39* genes indicated expression in anterior ciliated neurons (Fig. 8h), which, based on available literature[54], correspond to ASI and ASK neurons. However, we must note that we used a double deletion at the *sri-36/39* locus, the observed phenotype in exopher production could be driven by loss of only one receptor, with the other making little or no contribution. Nonetheless, the expression pattern of all three receptors is consistent with our data, underscoring the critical role of ADL, ASK, and ASI neurons in pathogen-induced exopher production and establishing its molecular mechanism.

Finally, the expression of the *srr-6* receptor in neurons of the retrovesicular ganglion (RVG), alimentary tract, and excretory system (Fig. 8l), as well as the expression of *sri-36* and *sri-39* in the alimentary tract (Fig. 8h), suggest that one or more of these tissues may also play a role in regulating exopher production in muscles. To test this, we drove *srr-6* expression specifically in the intestine. This intestinal rescue not only restored the non-volatile secretome-dependent exopher response but also reduced basal exopher levels, a result consistent with array-mediated overexpression, demonstrating that SRR-6 functions in the intestine to regulate muscle exopher production (Supplementary Fig. 7j) and revealing a novel gut-muscle signalling axis.

## Offspring of animals exposed to pathogen volatile metabolites show improved survival post-infection

We aimed to investigate the functional consequences of extracellular vesicles release when worms detect the pathogen in their environment. Prompted by a recent study showing that perception of *Pseudomonas vranovensis* volatiles confers intergenerational protection in *C. elegans*[56], we examined both offspring and maternal outcomes following pre-exposure to *P. aeruginosa* cues. Specifically, we assessed progeny survival and developmental rate during *P. aeruginosa* infection following parental pre-exposure to volatile metabolites or non-volatile secretomes, and evaluated how this preconditioning impacted hermaphrodite mothers' survival upon subsequent *P. aeruginosa*

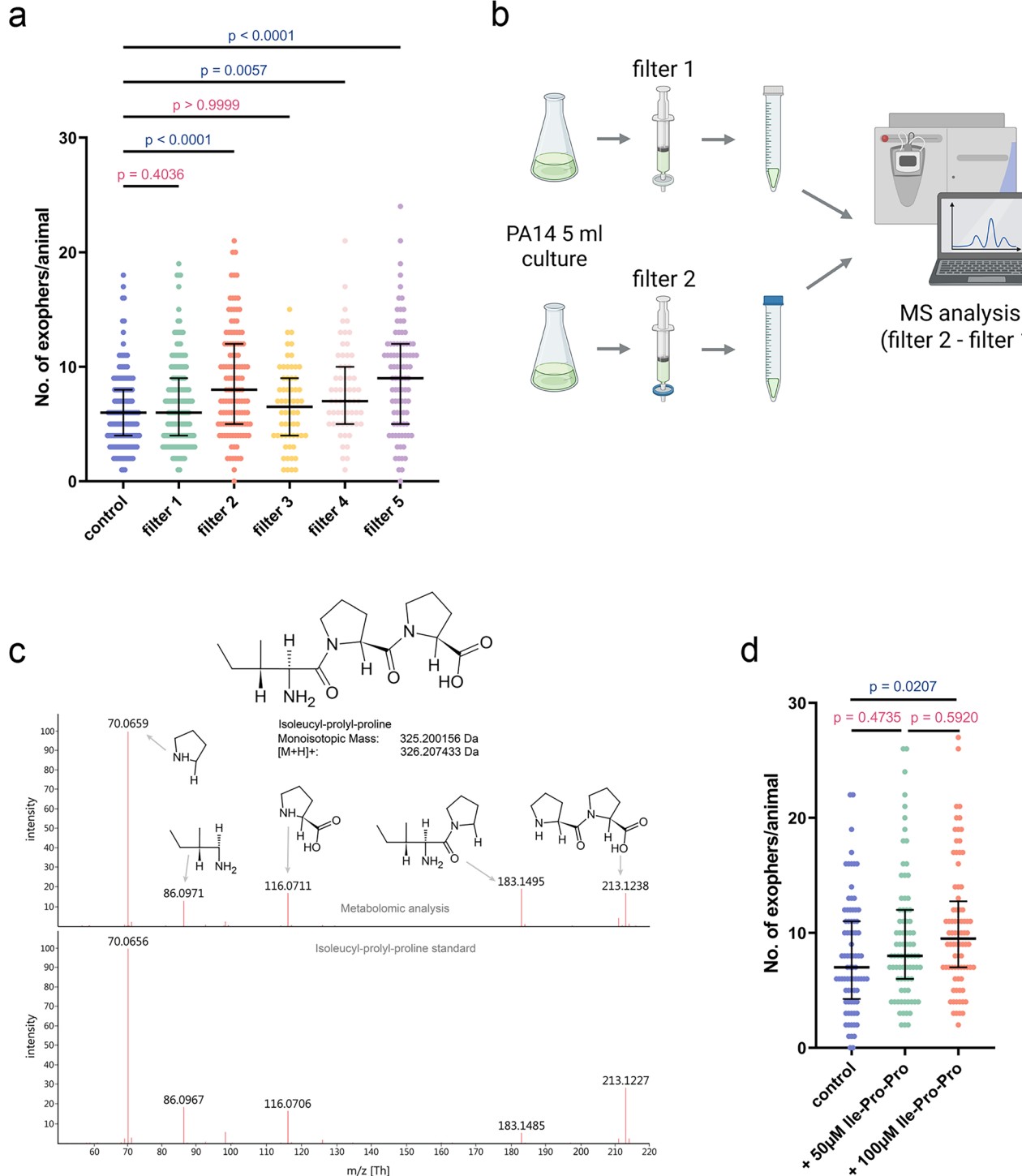

**Fig. 4 | Untargeted metabolomics revealed Ile-Pro-Pro tripeptide role in exopher upregulation. a** Evaluation of the robustness of pathogen supernatant in exopher induction revealed that filters 2 and 5 allowed efficient clarification while retaining strong exopher-inducing activity ($n = 150, 150, 120, 60, 60$ and 90 worms (for respective columns), N = 3-5 independent experiments). **b** Experimental setup schematic: Metabolite profiles of exopher-inducing (filter 2) and non-inducing (filter 1) filtrates from *P. aeruginosa* PA14 liquid cultures were compared using liquid chromatography–mass spectrometry. Created in BioRender. Kolodziejska, K. (https://BioRender.com/h5aij6j). **c** The fragmentation mass spectra of Isoleucyl-prolyl-proline acquired from the certified standard (below) and from the biological sample (above). Each significant signal is described by assigning the respective structure and mass of the fragment. Some of the fragments were generated as a result of neutral loss, others as [M + H]+ adducts. **d** The synthetic Ile-Pro-Pro tripeptide induces a dose-dependent increase in exopher production in exposed animals ($n = 80$ worms (for each column), N = 3 independent experiments). Data information: Data are presented as medians with interquartile ranges. Statistical analyses were performed using the Kruskal-Wallis test with Dunn's multiple comparisons test; non-significant *p* values ($p > 0.05$) are in pink color, significant *p* values ($p < 0.05$) are in blue color.

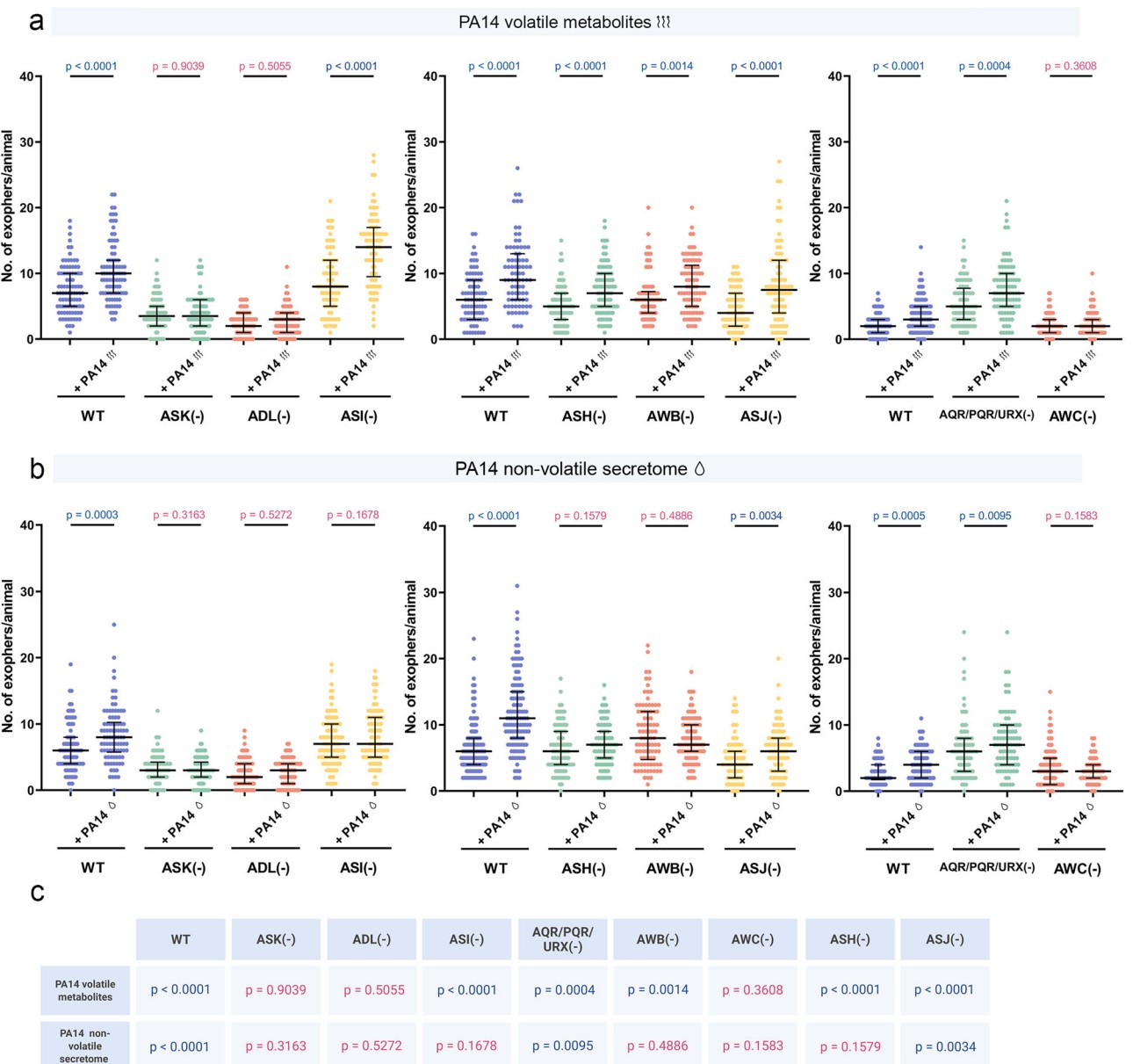

**Fig. 5 | Chemosensory neurons regulate exopher production in response to pathogen metabolite exposure. a** Exopher production upon exposure to PA14 volatile metabolites is dependent on ASK, ADL, and AWC neurons (*n* = 90, 90, 90, 88, 90, 88, 90 and 90 worms (for respective columns), N = 3 independent experiments; *n* = 80, 80, 90, 90, 90, 90, 90 and 90 worms (for respective columns), N = 3 independent experiments; *n* = 110, 120, 80, 90, 90 and 90 worms (for respective columns), N = 3-4 independent experiments). The data are presented in three separate graphs because three different exopher reporter lines, each exhibiting varying levels of basal exopher production, were used. The usage of different exopher reporter lines was due to crossing incompatibilities with specific neuronal ablation mutants. **b** Regulation of exopher production upon PA14 non-volatile secretome exposure involves multiple neurons, including ASK, ADL, ASI, AWB, AWC, and ASH (*n* = 90, 90, 90, 90, 90, 90, 120 and 120 worms (for respective columns), N = 3-4 independent experiments; *n* = 120, 120, 90, 90, 90, 90, 90 and 90

worms (for respective columns), N = 3-4 independent experiments; *n* = 90 worms (for each column), N = 3 independent experiments). The data are presented in three separate graphs because three different exopher reporter lines, each exhibiting varying levels of basal exopher production, were used. The usage of different exopher reporter lines was due to crossing incompatibilities with specific neuronal ablation mutants. **c** Summary of experimental results: A table summarizing the experimental findings, including p-values. Created in BioRender. Kolodziejska, K. (https://BioRender.com/snvoju0). Data information: Differences in basal exopher levels observed among wild-type controls are associated with the use of different transgenes for exopher visualization, as detailed in Supplementary Data 3. Data are presented as medians with interquartile ranges. Statistical analyses were performed using the two-tailed Mann-Whitney test, with non-significant *p* values (*p* > 0.05) in pink and significant *p* values (*p* <0.05) in blue.

infection (Fig. 9a). For these assays, P0 worms were exposed to *P. aeruginosa* volatile or non-volatile secretomes for 48 hours beginning at the L4 stage. To assess progeny outcomes, F1 embryos were collected at the 3-fold stage, transferred to *P. aeruginosa*–seeded plates at 20 °C, and scored after 48 hours for survival and developmental progression (scored as younger than L4, or L4 and older). For maternal survival, P0 adults (exposed to secretomes until day 2 of adulthood)

were transferred to fresh *P. aeruginosa* plates and monitored every few hours until death. Pre-exposure of P0 adults to *P. aeruginosa* volatiles significantly increased F1 survival under infection (Fig. 9b) without altering their rate of development to L4/adulthood (Fig. 9c). However, this effect did not persist in the F2 generation, where no significant differences in survival or development were observed (Supplementary Fig. 8a and b).

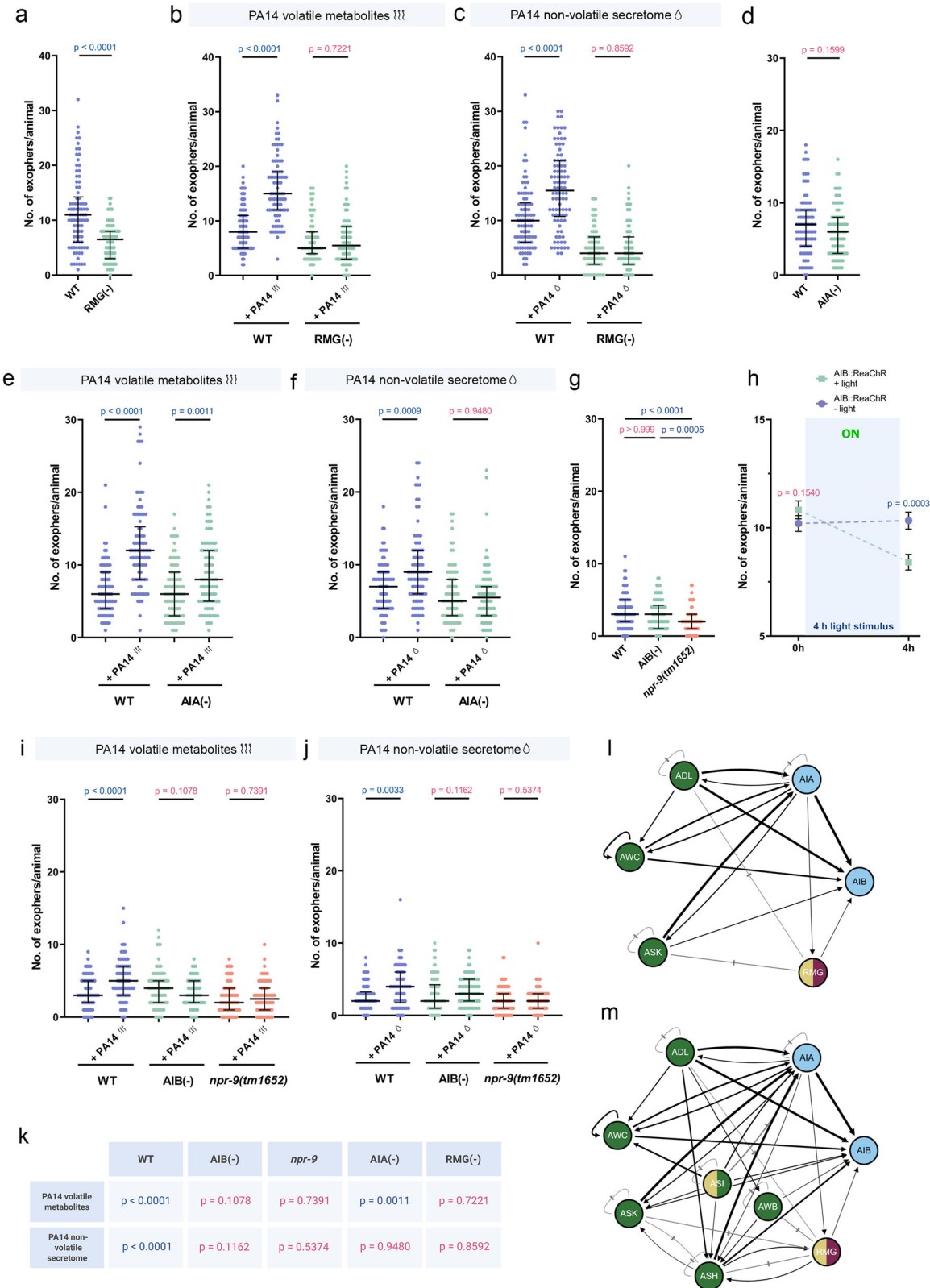

P0 pre-exposure to non-volatile secretomes had no effect on either progeny survival or developmental timing (Fig. 9d and e). For P0 mothers, volatile preconditioning produced a slight, non-significant decrease in survival on *P. aeruginosa* (Fig. 9f), whereas non-volatile preconditioning had no measurable impact (Fig. 9g).

We next asked whether the SRI-19 receptor, essential for exopher induction by *P. aeruginosa* and its secretomes, also influences progeny outcomes. Following P0 pre-exposure to *P. aeruginosa* volatiles or secretomes, *sri-19* mutants displayed no differences in F1 survival or developmental rate (Fig. 9h-k). This result indicates that the enhanced offspring resilience conferred by parental exposure to *P. aeruginosa* volatiles depends on the SRI-19-mediated pathway.

Elevated temperature of 25 °C both enhances exopher production in *C. elegans*[22] and increases *P. aeruginosa* virulence. Because wild worms

**Fig. 6 | AIA, RMG, AIB interneurons and AIB-specific NPR-9 receptor integrate sensory responses to mediate exopher production. a** Exopher production is reduced in worms with ablated RMG interneuron (*n* = 90 and 88 worms (for respective columns), N = 3 independent experiments). **b** RMG interneuron is required for exopher production triggered by pathogen volatile metabolites (n = 88, 88, 83, and 78 worms (for respective columns), N = 3 independent experiments). **c** Exopher production induced by pathogen-derived non-volatile secretome requires the RMG interneuron (*n* = 90 worms (for each column), N = 3 independent experiments). **d** Animals lacking the AIA interneuron produce a similar number of exophers as wild-type worms (*n* = 90 worms (for each column), N = 3 independent experiments). **e** Increased exopher production after exposure to PA14 volatile metabolites partially depends on AIA interneuron (*n* = 90 worms (for each column), N = 3 independent experiments). **f** Exopher production induced by PA14 non-volatile secretome is mediated by the AIA interneuron (*n* = 90 worms (for each column), N = 3 independent experiments). **g** Exopher production in strain lacking the AIB interneuron is comparable to that in wild-type worms and lower in worms lacking NPR-9 receptor (*n* = 90 worms (for each column), N = 3 independent experiments). **h** ReaChR-mediated optogenetic activation of AIB interneuron decreases exopher production (*n* = 150 worms (for each group), N = 5 independent experiments). **i** AIB interneuron and AIB-specific NPR-9 receptor control exopher release upon pathogen volatile metabolites (*n* = 91, 90, 84, 88, 90, and 90 worms (for respective columns), N = 3 independent experiments). **j** AIB interneuron and the NPR-9 receptor are required for exopher production induced by pathogen-derived non-volatile secretome. (*n* = 90 worms (for each column), N = 3 independent experiments). **k** Summary of experimental results: A table summarizing the experimental findings, including *p* values. Created in BioRender. Kolodziejska, K. (https://BioRender.com/3t9pqy9). **l** Neuronal circuit controlling the exopher release upon volatile PA14 metabolites exposure (connectome data from White et al.[47] and Witvliet et al.[75]). Created with wormwideweb.org. **m** Neuronal circuit regulating exopher release upon exposure to PA14 non-volatile metabolites (connectome data from White et al.[47] and Witvliet et al.[75]). Created with wormwideweb.org. Data information: Differences in basal exopher levels observed among wild-type controls are associated with the use of different transgenes for exopher visualization, as detailed in Supplementary Data 3. Data are presented as medians with interquartile ranges (**a–g, i, j**) or mean with SEM (**h**). Statistical analyses were performed using the Kruskal-Wallis test with Dunn's multiple comparisons test (**g**) and the two-tailed Mann-Whitney test (**a–f, h–j**), with non-significant *p* values (*p* > 0.05) in pink and significant *p* values (*p* < 0.05) in blue.

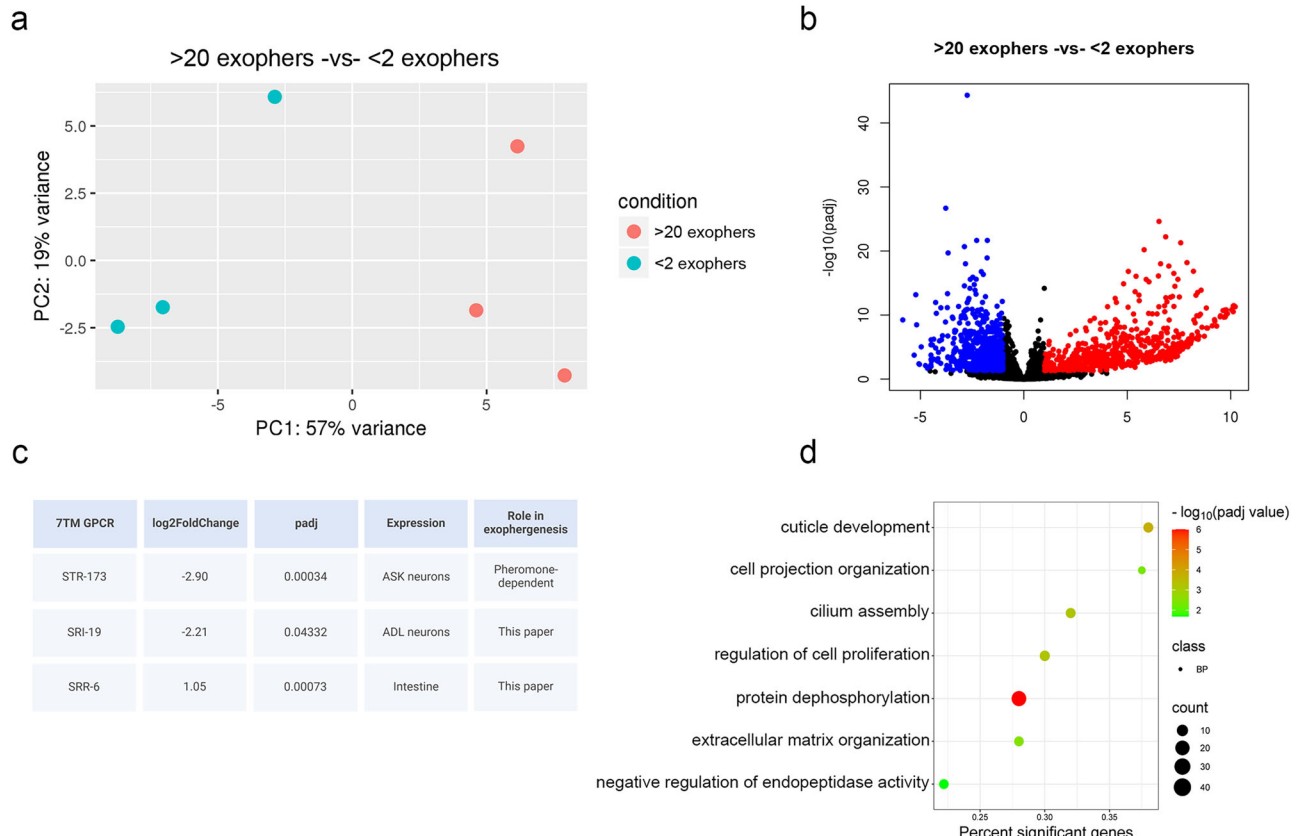

**Fig. 7 | Comparative transcriptomic analysis of worms producing high and low number of exophers per individual. a** Principal component analysis (PCA) comparing worms with high exopher production (>20 exophers) and low exopher production (<2 exophers). **b** Volcano plot showing differential gene expression between worms producing high (>20 exophers) and low (<2 exophers) number of exophers. **c** Table listing 7TM GPCR transcripts with significant changes in expression levels. Created in BioRender. Kolodziejska, K. (https://BioRender.com/gs6ncio). **d** Gene Ontology (GO) term analysis plot based on the transcriptome results.

encounter pathogens in a range of environmental temperatures, we assessed how F1 survival and development are affected at 25 °C. As shown in Supplementary Fig. 9a–d, these high-temperature outcomes differ markedly from those at 20 °C (Fig. 9b–e). Progeny of parents pre-exposed to *P. aeruginosa* volatiles developed faster (Supplementary Fig. 9b) but showed unchanged survival (Supplementary Fig. 9a), whereas progeny of parents exposed to *P. aeruginosa* non-volatile secretome suffered both reduced survival and delayed development

compared to unexposed controls (Supplementary Fig. 9c and d). In P0 adults, pre-exposure to *P. aeruginosa* volatiles significantly decreased survival upon subsequent *P. aeruginosa* infection (Supplementary Fig. 9e), while pre-exposure to non-volatile secretome had no effect (Supplementary Fig. 9f). Notably, *sri-19* mutants, accumulated fewer embryos *in utero* (Supplementary Fig. 9g), and are unable to upregulate exophers in response to *P. aeruginosa*, but survived significantly longer during infection (Supplementary Fig. 9h). Likewise, knockdown of the

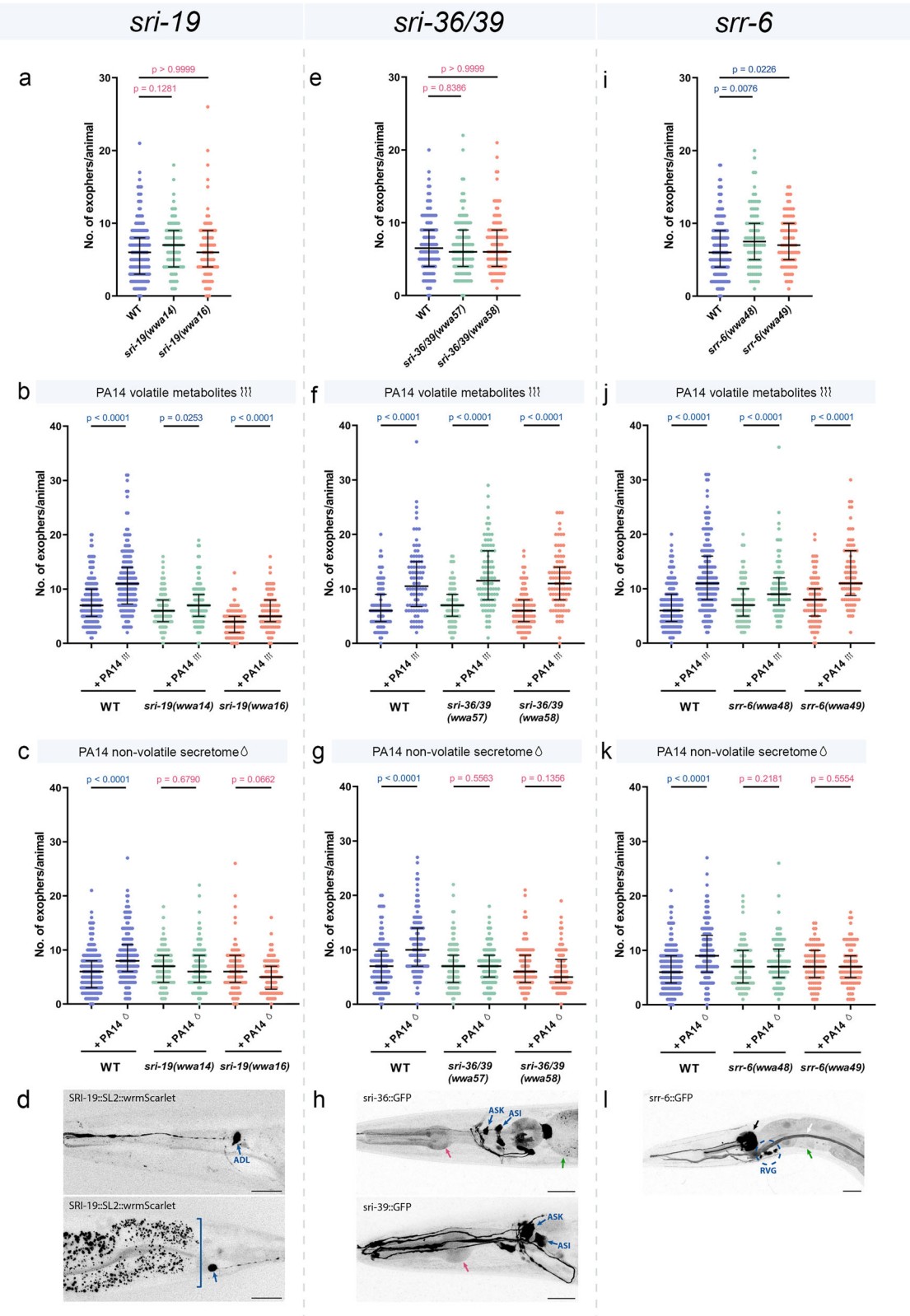

yolk receptor RME-2, which nearly abolishes exopher production[18], and reduces embryo accumulation *in utero* (Supplementary Fig. 9i) also conferred enhanced resistance to *P. aeruginosa* (Supplementary Fig. 9j). Together, these data demonstrate that parental sensing of *P. aeruginosa* volatiles enhances offspring resilience at 20 °C, while elevated environmental temperature abolishes this effect, underscoring its context-dependent nature. Finally, SRI-19's role in this pathway directly links the machinery of exopher biogenesis to survival outcomes, highlighting how EV production and organismal fitness are mechanistically intertwined.

## Discussion

Our findings support a unified model (Fig. 10), in which *C. elegans* employs adaptive exophergenesis as a sophisticated response

**Fig. 8 | Exopher production induced by PA14 non-volatile metabolites is regulated by multiple GPCRs. a** Exopher production in worms lacking the SRI-19 receptor is comparable to that in wild-type worms ($n$ = 180, 90 and 90 worms (for respective columns), N = 3 independent experiments). **b** Increased exopher production after exposure to PA14 volatile metabolites does not depend on SRI-19 receptor ($n$ = 180, 180, 114, 108, 90 and 90 worms (for respective columns), N = 3-4 independent experiments). **c** The SRI-19 receptor is required for exopher production induced by PA14 non-volatile secretome ($n$ = 180, 180, 90, 90, 90 and 90 worms (for respective columns), N = 3 independent experiments). **d** The *sri-19* gene encoding 7TM receptor is expressed in ADL neurons and PHA tail neuron, based on position and scRNAseq data[76], consistent with observations in at least 20 animals from three replicates. Square bracket mark gut autofluorescence. **e** Exopher production in worms lacking the SRI-36 and SRI-39 receptors is comparable to that in wild-type worms ($n$ = 120 worms (for each column), N = 4 independent experiments). **f** Increased exopher production after exposure to PA14 volatile metabolites does not depend on SRI-36 and SRI-39 receptors ($n$ = 90 worms (for each column), N = 3 independent experiments). **g** The SRI-36 and SRI-39 receptors are required for exopher production induced by PA14 non-volatile secretome ($n$ = 120, 120, 120, 120, 90 and 90 worms (for respective columns), N = 3-4 independent experiments).

**h** The *sri-36* and *sri-39* genes encoding 7TM receptors are expressed in neurons (including ASK# and ASI#) and the alimentary tract (red arrow – pharynx, green arrow – intestine), consistent with observations in at least 20 animals from three replicates. #based on position and Vidal et al.[54]. **i** Basal exopher levels in *srr-6* mutants is increased compared to wild-type worms ($n$ = 180, 90 and 90 worms (for respective columns), N = 3 independent experiments). **j** Increased exopher production after exposure to PA14 volatile metabolites does not depend on SRR-6 receptor ($n$ = 180, 180, 90, 90, 90 and 90 worms (for respective columns), N = 3 independent experiments). **k** The SRR-6 receptor is required for exopher production induced by PA14 non-volatile secretome ($n$ = 180, 180, 90, 90, 90 and 90 worms (for respective columns), N = 3 independent experiments). **l** The *srr-6* gene encoding 7TM receptor is expressed in neurons of the retrovesicular ganglion (encircled), pharynx neurons (black arrow), alimentary tract (green arrow – intestine), and excretory system (white arrow), consistent with observations in at least 20 animals from three replicates. Data information: Scale bars are 20 µm. Data are presented as medians with interquartile ranges. Statistical analyses were performed using the Kruskal-Wallis test with Dunn's multiple comparisons test (**a**, **e**, **i**) and the two-tailed Mann–Whitney test (**b**, **c**, **f**, **g**, **j**, **k**); non-significant $p$ values ($p > 0.05$) are in pink color, significant $p$ values ($p < 0.05$) are in blue color.

mechanism to pathogen-derived environmental cues. We demonstrate that worms effectively discriminate pathogen cues based on their chemical nature, volatile versus non-volatile. Volatile metabolites are small, low-molecular-weight compounds with high vapour pressures, diffusing rapidly through air or aqueous films to create broad, low-concentration plumes. In contrast, compounds present in non-volatile secretomes are larger, more polar molecules that remain close to their source, forming steep, localized concentration peaks. We propose that distinguishing between these two types of compounds could enable worms to accurately infer pathogen proximity and subsequently mount tailored physiological responses. This response is orchestrated by immune signaling in a stimulus-specific manner: exopher production in response to volatile metabolites is modulated by insulin, TGF-β, and p38 MAPK but proceeds independently of IPR, whereas non-volatile secretomes require TGF-β, p38 MAPK, and IPR. The partial overlap of these pathways[57,58] suggests coordinated or compensatory interactions that fine-tune exophergenesis according to the nature and intensity of the pathogen cue. The sensory neurons ADL, ASK, and AWC detect both cues, whereas ASI, ASH, and AWB neurons specialize in detecting non-volatile compounds. Signal integration occurs downstream via RMG hub, AIA, and AIB interneurons, with the AIB-specific GPCR NPR-9 functioning as one of the regulatory components. Both neuronal and non-neuronal GPCRs (SRI-19, SRI-36/39, and SRR-6) are essential for transducing the non-volatile secretome signal into exopher release, and SRI-19 is responsible for modulation of the response to volatile metabolites. Moreover, *P. aeruginosa* released tripeptide Ile-Pro-Pro functions as an inducer of exopher production. Functionally, we demonstrate that maternal exposure to volatile pathogen metabolites triggers an SRI-19-dependent increase in offspring survival at 20 °C, linking the regulation of exopher formation with survival outcomes and suggesting that both processes are coordinated through shared signaling mechanisms.

Our findings on pathogen-induced exophergenesis gain broader significance when considered alongside our recent work on its regulation by social cues, revealing a unifying framework in which *C. elegans* leverages the same physiological process, muscle-derived exopher production, to navigate radically different environmental contexts. In both cases, exophergenesis seems to serve as a form of maternal resource allocation that supports offspring fitness, yet the initiating stimuli represent opposite poles of the ecological spectrum: pathogenic threat versus reproductive opportunity. The male-produced ascaroside pheromone ascr#10 enhances maternal provisioning in anticipation of mating, whereas pathogen-derived metabolites provoke a terminal investment in progeny under threat of infection. This functional reuse of a conserved output module

underscores the evolutionary versatility of the exopher pathway. A striking point of convergence across these two systems is the ASK and ADL sensory neurons, which emerge as master integrators capable of detecting important environmental information. For instance, in the social context, ASK detects ascr#10 via the GPCR STR-173; in the context of pathogen exposure, ASK, together with ADL, detects non-volatile pathogen secretomes likely via distinct GPCRs such as SRI-36/39 and SRI-19. This dual role positions ASK and ADL as a key node in the sensory network that interprets both reproductive cues and pathogen-derived danger signals, illustrating the remarkable contextual plasticity of the sensory code. However, beyond this shared component, the upstream sensory receptors and downstream processing circuits diverge almost completely, demonstrating the evolution of parallel, specialised neural architectures for environmental decision-making. The pathogen-responsive circuit engages a distributed network comprising ASK, ADL, and AWC neurons, integrated through the RMG, AIB, and AIA interneurons hub mediated through canonical immune pathways such as TGF-β, p38 MAPK, and the IPR system. In contrast, the social cue circuit recruits the AQR, PQR, and URX sensory neurons and is modulated by neuropeptides including FLP-8 and FLP-21. This divergence could be explained by a broader principle of *C. elegans* sensory neuroscience: worms employ combinatorial coding and distributed receptor repertoires to extract meaning from complex environmental landscapes. Rather than relying on a single labelled-line logic, the nervous system interprets unique activation patterns across multiple neurons[59], allowing for nuanced discrimination between identity and concentration of an odorant stimulus. In this context, a similar mechanism could underlie the discrimination between pathogen threats and imminent reproductive opportunities. The evolution of dual sensory systems converging on exopher production, each with distinct inputs and physiological consequences, illustrates how a single effector process can be co-opted to serve adaptive roles across diverse ecological scenarios.

Importantly, the nature of the exopher response is determined not only by the chemical nature of the stimulus (volatile vs. non-volatile) but also by the associated signalling architecture. Non-volatile cues require canonical immune modules, including TGF-β, p38 MAPK, and the IPR pathway, signalling axes that are deeply conserved across metazoans and underscore an immune-centric, mode of defensive mobilisation. In stark contrast, volatile-triggered responses proceed largely independently of these immune effectors, reflecting an alternative regulatory strategy of pathogen anticipation. Interestingly, we did not observe an additive effect when both volatile and non-volatile pathogen-derived cues were present simultaneously. This suggests that these two types of signals may converge onto a shared regulatory

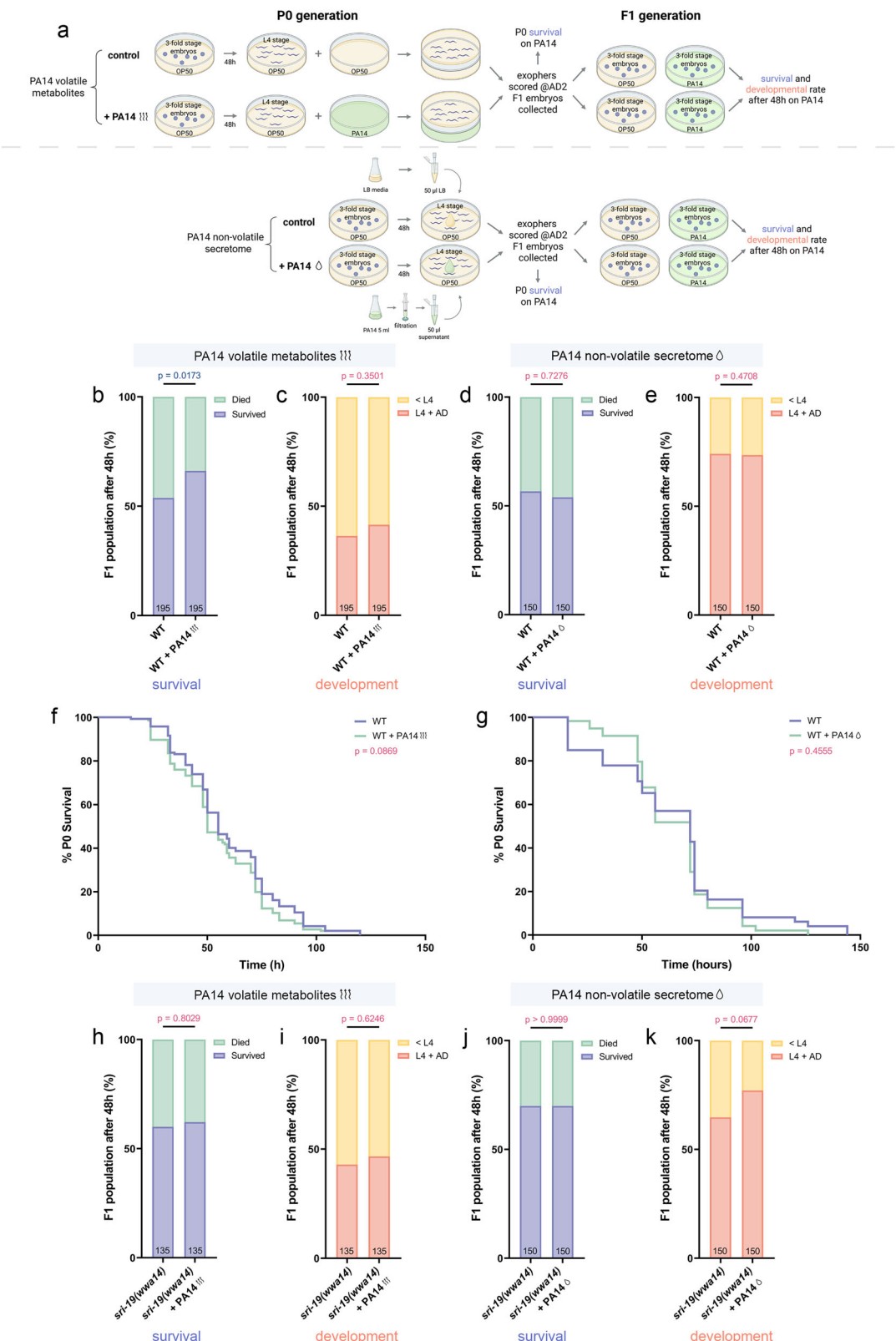

or effector pathway that defines an upper limit for exopher production, rather than acting through independent mechanisms. This distinction also raises the possibility that exophers may constitute distinct subtypes with unique molecular compositions and functions. The chance that quantitatively similar exophers could be functionally distinct opens new research avenues to isolate and characterize these vesicles in detail. By analysing their content in response to both volatile

and non-volatile secretomes, we could uncover their specific roles in organismal health and pathogen defense. This principle echoes findings in mammalian extracellular vesicle biology, where cargo composition and functional outcome are determined by the upstream signalling context[60].

In this study, we focused on key immune signalling pathways in *C. elegans*, including p38 MAPK, insulin signalling, TGF-β, and the

**Fig. 9 | The SRI–19 receptor is essential for the increased progeny resilience observed after maternal pre-exposure to *P. aeruginosa* volatile cues.**
**a** Experimental setup schematic: Age-synchronized L4-stage worms (P0 generation) were exposed for 48 h to either PA14 and OP50 volatile metabolites (top) or PA14 non-volatile secretome and LB media (bottom). After this exposure, exopher production was quantified at the AD2 stage, and embryos from the F1 generation were collected. The P0 worms were subsequently transferred to Slow Killing (SK) plates seeded with PA14, where their survival was monitored. F1 embryos were subsequently transferred to SK plates seeded with PA14, where their survival and development was quantified after 48 h post-infection. Created in BioRender. Kolodziejska, K. (https://BioRender.com/w2ui4aw). **b** Offspring of hermaphrodites exposed to PA14 volatile metabolites have better survival rate upon PA14 infection (*n* = 195 worms (for each column), N = 5 independent experiments). **c** No significant differences were observed in the developmental rate of F1 progeny between control animals and those whose parents were exposed to PA14 volatile metabolites (*n* = 195 worms (for each column), N = 5 independent experiments). **d** Exposure of parents to PA14 non-volatile secretome had no significant impact on F1 survival during subsequent PA14 infection (*n* = 150 worms (for each column), N = 4 independent experiments). **e** Developmental progression of F1 progeny during PA14 infection was not significantly affected by parental exposure to PA14 non-volatile secretome (*n* = 150 worms (for each column), N = 4 independent experiments).

**f** Wild-type P0 worms exposed prior to infection to PA14-derived volatile metabolites had similar survival rate comparing to the control group (*n* = 163 and 165 worms (for respective groups), N = 6 independent experiments). **g** Wild-type P0 worms pre-exposed to PA14-derived secretome showed no significant difference in survival compared to the control group upon subsequent PA14 infection (*n* = 52 worms (for each group), N = 2 independent experiments). **h** F1 progeny of *sri-19* mutant worms pre-exposed to PA14-derived volatile metabolites showed no difference in survival upon PA14 infection compared to the control group (*n* = 135 worms (for each column), N = 3 independent experiments). **i** Parental exposure to PA14 volatile metabolites did not affect the developmental timing of F1 progeny in the *sri-19* mutant background (*n* = 135 worms (for each column), N = 3 independent experiments). **j** No significant difference in survival was observed in the F1 generation of *sri-19* mutants following parental exposure to PA14 non-volatile secretome (*n* = 150 worms (for each column), N = 4 independent experiments). **k** Developmental progression of F1 progeny was unchanged in *sri-19* mutants whose parents were exposed to PA14 non-volatile secretome (*n* = 150 worms (for each column), N = 4 independent experiments). Data information: Data are presented as stacked bar plots (**b–e**, **h–k**) and Kaplan–Meier survival curves (**f**, **g**); Statistical analyses were performed using the Fisher's exact test (**b–e**, **h–k**) and long-rank (Mantel–Cox) test (**f**, **g**); non-significant *p* values (*p* > 0.05) are in pink color, significant *p* values (*p* < 0.05) are in blue color.

intracellular pathogen response (IPR) pathway, which are well established as evolutionary conserved central mediators of pathogen-induced stress responses. Although additional transcriptional regulators such as SKN-1[61], ELT-2[62,63], and HLH-30[64] are also known to participate in immune and stress-related processes, their functions are highly pleiotropic and extend beyond immunity to encompass broader roles, for instances in stress responses[65], detoxification[66], and autophagy[67]. Nevertheless, the potential involvement of SKN-1, ELT-2, and HLH-30 in pathogen compounds-dependent EV regulation remains an intriguing avenue for future investigation.

Interestingly, while volatile metabolites like 1-undecene have been shown to increase lifespan in *C. elegans* under non-pathogenic conditions[68] or to protect worms against a subsequent infection with *P. aeruginosa*[15], our results demonstrate that volatile-induced exopher production can lead at higher temperature to decreased survival during infection. The differences in the type, timing, and context of exposure likely contribute to this contradiction. In previous studies[15,68], a specific compound and shorter exposure periods were used. In contrast, our study involved prolonged exposure to a mixture of volatile metabolites from pathogens, which may trigger different physiological responses. The involvement of multiple G protein-coupled receptors in exopher production indicates that exophergenesis is driven by a complex mixture of pathogen-derived signals, rather than a single metabolite, although some receptors may share common ligands. This complexity may allow the organism to fine-tune its response depending on the nature of the threat it perceives, enabling a more nuanced and context-dependent reaction to environmental challenges.

Previous studies have shown that increased exopher release can benefit the organism by removing protein aggregates[17], extending lifespan[20], or promoting faster offspring development[18]. However, when exopher release is elevated in response to pathogen exposure at higher temperature, we observe an opposite effect, leading to decreased survival upon infection. Intriguingly, our data reveal tissue-specific exopher responses: muscle-derived exophers are induced by both volatile and non-volatile cues, whereas neuronal exophers respond exclusively to volatile signals. The apparent paradox between muscle- and neuron-derived exophers may reflect distinct functional roles: neuronal exophers promote longevity via cell-autonomous proteostasis, while muscle exophers serve as transporters of biological material. Moreover, this raises the possibility that exophers may function under a hormetic response: while moderate exopher production may be beneficial, too much exopher release might impose a burden on the organism, potentially due to

changes in the content of these vesicles under different environmental conditions.

Our findings suggest that pathogen-induced exophergenesis contributes to offspring fitness by promoting non-genetic maternal investment strategies. One possibility is that exophers serve as vehicles for protective cargos that are ultimately transferred to developing oocytes. Alternatively, elevated exopher output may reflect a systemic reconfiguration of maternal physiology that enhances oocyte provisioning through shared metabolic or stress-response pathways without the direct involvement of EVs in this process. Addressing this will require not only profiling exopher content but also functional testing, such as isolating exophers and injecting them into naïve mothers to assess whether they confer fitness advantages to offspring in the absence of infection. Resolving this will yield fundamental insights into the mechanisms of non-genetic, transgenerational immunity and into the broader role of EV as adaptive mediators of host-pathogen interactions.

In conclusion, our study highlights the complexity of exopher production in response to pathogen-derived signals. The ability of *C. elegans* to detect and respond to volatile and non-volatile compounds in a context-dependent manner suggests that exophers and possibly other EVs play a crucial role in mediating the organism's physiological response to environmental stressors. Future research should focus on isolating and characterizing exophers produced under different conditions to determine their specific molecular contents and functions. Understanding how these vesicles contribute to the long-term offspring fitness will provide deeper insights into the adaptive roles of EVs in host-pathogen interactions.

## Methods

### Worm Maintenance and Strains
*C. elegans* strains were maintained at 20 °C (unless otherwise indicated) on nematode growth medium (NGM) plates seeded with *Escherichia coli* OP50 as a food source[69]. Only hermaphrodites were used in the experiments. A comprehensive list of all strains and chemicals used in the study is provided in Supplementary Data 3 and 4, respectively.

### Pathogen procedures
**Pathogen bacteria media plates preparation.** Worms were exposed to either *Pseudomonas aeruginosa* PA14 or *Serratia marcescens* Db10 strains. Pathogen-seeded plates were prepared following standard protocols[70,71]. All the experiments with the pathogenic bacteria were conducted in Biosafety level 2 laboratory.

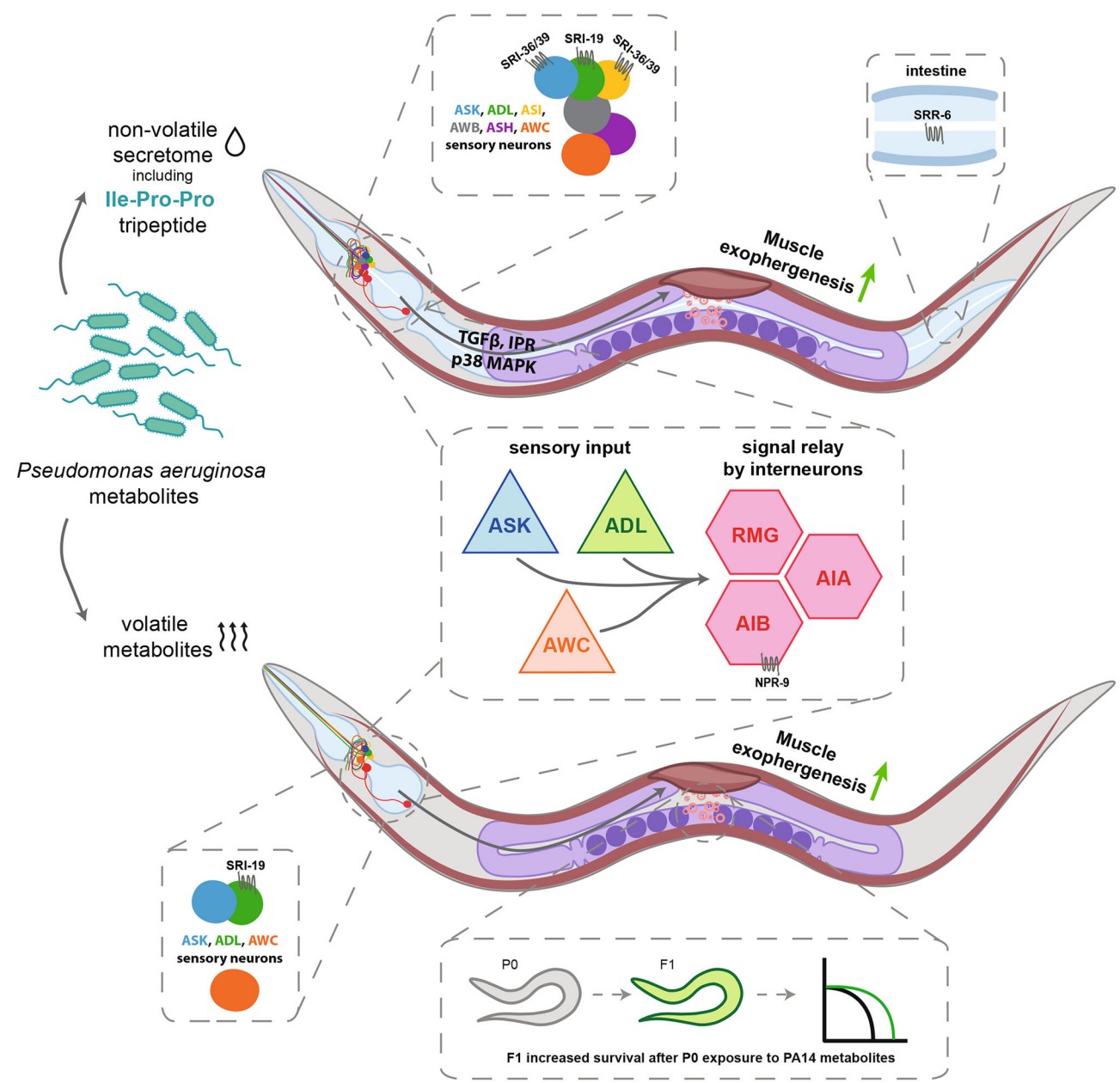

**Fig. 10 | Model of exopher regulation from muscles by pathogen non-volatile secretome and volatile metabolites.** Distinct volatile and non-volatile pathogen cues are processed through defined sensory and immune signaling networks to regulate exopher production and promote offspring survival.

***Pseudomonas aeruginosa* PA14 "slow-killing" (SK) plates preparation.** The PA14 strain was revived from frozen stock by streaking onto a 10-cm LB agar plate and incubating at 37 °C for 12–18 hours. Plates were stored at 4 °C and used within one week. A single PA14 colony was inoculated into 5 mL of LB broth and incubated overnight (maximum 16 hours) at 37 °C with shaking at 250 rpm. Following incubation, 10 µL of the overnight PA14 culture was spotted onto the center of 35 mm "slow killing" plates (SK plates), which were incubated at 37 °C for 24 hours, followed by 25 °C for another 24 hours to maximize virulence.

**Serratia marcescens Db10 plates preparation.** For *S. marcescens Db10*, the strain was similarly streaked from frozen stock onto a 10-cm LB agar plate, incubated at 37 °C for 12–18 hours, and stored at 4 °C for up to one week. A single Db10 colony was inoculated into 5 mL of LB broth and cultured at 37 °C with shaking for 8–10 hours. To prepare NGM infection plates, 25 µL of the *S. marcescens* culture was seeded onto 35 mm NGM plates, which were incubated at 37 °C overnight.

**Short-time infection assay.** Worms were age-synchronized by isolating 3-fold stage embryos (hand-picked under a stereomicroscope). Upon reaching the first day of adulthood (AD1), they were transferred either to control plates seeded with *E. coli* OP50 or to plates seeded with the pathogens PA14 or Db10. Worms were exposed directly to the pathogens for 3 hours at 25 °C, after which they were transferred back to OP50 seeded plates. Exopher levels were assessed the following day, on the second day of adulthood (AD2).

**Pathogen non-volatile secretome assay.** To collect non-volatile secretome, a single colony of PA14 was inoculated into 5 mL of LB media and cultured overnight for a maximum of 16 hours at 37 °C. After incubation, bacterial cells were removed by passing the culture through a 0.22 µm filter (Sarstedt, Filtropur S, no. 83.1826.001), and the bacterial supernatant was collected. Age-synchronized L4 stage worms (n = 30), grown on 60 mm NGM plates seeded with OP50, were treated with 50 µL of the filtered bacterial supernatant, while control

worms received 50 µL of LB broth. Plates were sealed with parafilm, and exophers were scored 48 hours later, when the worms reached AD2.

**Volatile metabolites chamber assay.** 3-fold stage embryos were placed on 35-mm NGM plates seeded with *E. coli* OP50, with 10 embryos per plate. Once the embryos developed into L4 stage worms, a chamber with volatile metabolites was assembled by placing a 35 mm SK plate containing PA14 or a control 35 mm NGM plate with OP50 on the bottom. The plate with synchronized worms was positioned on top, and the two plates were sealed together with parafilm. Worms were incubated with volatile metabolites of the bacteria for 48 hours. Exophers were evaluated on the second day of adulthood.

**Parental generation survival.** The parental generation (P0) of *C. elegans* was synchronized by transferring 10 3-fold stage embryos onto NGM plates seeded with *Escherichia coli* OP50. When the worms reached the L4 larval stage, they were subjected to the "Volatile Metabolites Chamber Assay" or "Pathogen non-volatile secretome assay" as previously described. After 48 hours of exposure to the volatile or non-volatile secretomes, exopher production was quantified in the P0 generation at AD2. The P0 worms were then moved to SK plates seeded with PA14 and maintained at 20 °C (Fig. 9f, g) or 25 °C (Supplementary fig. 8e-h). The use of 5-fluoro-2′-deoxyuridine (FUdR) was omitted from all plates. Survival of the P0 population was monitored at multiple time points using a touch-provoked movement assay. Worms that did not respond to touch were recorded as deceased, removed from the killing plates, and documented. Those that responded to touch were considered alive and were counted accordingly. Survival data were collected at least twice daily until the death of the entire population.

**Offspring survival and development assay.** Once the parental generation (P0) of worms reached the L4 larval stage, they were subjected to the "Volatile Metabolites Chamber Assay" or "Pathogen non-volatile secretome assay". After 48 hours of secretome exposure, exopher production was quantified in the P0 generation on the second day of adulthood (AD2).

To assess intergenerational effects, 3-fold stage embryos of the F1 generation, derived from P0 worms were collected and transferred to 35 mm SK plates seeded with PA14 bacteria (10–15 embryos per plate). The plates were incubated at 20 °C (Fig. 9b-e, h-k) or 25 °C (Supplementary fig. 8a-d). Offspring survival and development were assessed after 48 h by scoring the total number of worms and determining their developmental stages.

**Embryo accumulation assay.** First, AD1 age-synchronized worms were placed on either a lawn of pathogenic bacteria (PA14 and Db10) or a control lawn of OP50 for 3 hours. After exposure, worms were transferred to OP50 plates and cultured until they reached the AD2 stage. In the next step, hermaphrodites were treated with 1.8% hypochlorite solution. Once the worms dissolved, the embryos retained in the uterus were counted.

**RNA Interference Assay.** RNA interference (RNAi) in *C. elegans* was performed using the standard RNAi feeding method[72]. Briefly, RNAi plates were prepared by supplementing NGM with 12.5 µg/mL tetracycline, 100 µg/mL ampicillin, and 1 mM IPTG. These plates were seeded with *E. coli* HT115 bacteria expressing double-stranded RNA (dsRNA) targeting the gene of interest. Control worms were fed *E. coli* HT115 containing the empty vector L4440.

Age-synchronized 3-fold stage embryos were transferred onto the freshly prepared RNAi plates and cultured until day 2 of adulthood, at which point exophers were quantified. The developmental stage of the worms was confirmed at the L4 stage, and any individuals that were developmentally advanced or delayed were excluded from the experiment.

**Metabolomic analysis.** All solvents were prepared prior to analysis: extraction solvent was prepared by mixing 1 mL of LC-MS grade methanol, 4.5 mL of LC-MS grade acetonitrile and 4.5 mL of LC-MS grade isopropanol; 90% acetonitrile was prepared by mixing 9 mL of LC-MS grade acetonitrile with 1 mL of MQ grade water; mobile phase A (MPA) was prepared by adding 1 mL of formic acid and filling it to 1 L with MQ grade water; mobile phase B (MPB) was prepared by adding 1 mL of formic acid and filling it to 1 L with LC-MS grade acetonitrile. Metabolite extraction was achieved by transferring 50 µL of the sample to an Eppendorf LoBind tube and adding 250 µL of ice-cold extraction solution of methanol:acetonitrile:isopropanol (2:9:9, v/v/v). Samples were shaken for 10 min at 1500 RPM in RT and subsequently centrifuged for 10 min at 14 000 RPM at 4 °C. Supernatants were transferred to new polypropylene sample tubes and evaporated to dryness under nitrogen stream at 50 °C using Caliper TurboVap LV evaporator. Dry residue was reconstituted by addition of 100 µL of 90% acetonitrile, vortexed and centrifuged.

LC-MS analysis was performed using ACQUITY UPLC system (Waters) coupled with Q-Exactive Orbitrap high resolution mass spectrometer (Thermo Scientific). Chromatographic separation was achieved using ACQUITY UPLC BEH HILIC column (130 Å, 1.7 µm, 2.1 mm × 100 mm). MPA was 0.1% formic acid in MQ grade water and MPB was 0.1% formic acid in LC-MS grade acetonitrile. The initial gradient conditions were 100% B with mobile phase flow rate at 0.4 mL/min and was held for 1 min. The mobile phase composition changed to 90% B over next 6 min, then dropped to 50% B at 9 min, and to 10% B at 10 min to be kept for next 4 min. Column was re-equilibrated by changing the gradient back to initial conditions. The total method run time was 20 min. The mass spectrometer operated in separate positive (ESI+) and negative (ESI-) electrospray ionization. The tune parameters were as given: sheath gas flow rate 55.0 [a.u.], auxiliary gas flow rate 15.0 [a.u.], sweep gas flow rate 4.0 [a.u.], spray voltage in ESI+ was 4.0 and in ESI- was 3.5, S-lens RF 30.0 [a.u.]. Capillary temperature was set to 320 °C and auxiliary gas temperature to 350 °C. Data acquisition was proceeded in Data Dependent Acquisition (DDA) mode, MS1 resolution set to 70,000, MS1 AGC Target $1.0 \times 10^6$, MS 1 Maximum Injection Time 90 ms, scan range 120 to 1600 m/z, and MS2 resolution was set to 35,000, MS2 AGC Target to $1.0 \times 10^5$, MS2 Maximum Injection Time 100 ms, TopN was 8, isolation window 2.0 m/z and normalized stepped collision energy of 20, 30 and 40 was used to collect fragmentation spectra.

Raw data were uploaded to Compound Discoverer 3.3.3 (Thermo Scientific). An in-house protocol for metabolomic search and identification was employed. The workflow tree consisted of: Input Files → Select Spectra (any polarity, precursor mass 100–1500 Da, MS(n-1) precursor selection) → Align Retention Times (adaptive curve alignment model, max mass tolerance 10 ppm, max shift time 2.0 min) → Detect Compounds (mass tolerance 15 ppm, signal/noise> 5, max peak width 0.3 min, base ions [M + H]⁺ and [M-H]⁻) → Merge Features (mass tolerance 15 ppm and RT tolerance 0.5 min) → Group Compounds (mass tolerance 10 ppm, RT tolerance 0.2 min, all peaks used for area integration, preferred ions [M + H]⁺ and [M-H]⁻) → Fill Gaps (mass tolerance 5 ppm, signal/noise> 3) → Normalize Areas (Median Absolute Deviation model) → Mark Background (max sample/blank=5). Predict Composition node was set to mass tolerance of 5 ppm, isotope instensity tolerance of 30%, and minimum pattern coverage of 90%. Metabolomika Pathways, mzVault, mzCloud and ChemSpider databases were selected for analytes identification and assignment. Identifications were analyzed using mzLogic node with minimum match threshold of 30 and high resolution fragment mass tolerance 15 ppm.

XCalibur 3.0.63 (Thermo Scientific) software was used to design LC-MS workflow and acquire sample batch, as well to analyze raw mass

spectra using built-in Qual Browser. OpenMS 3.4.0 software was used to visualize the raw fragmentation spectra. ACD/ChemSketch 2024.2.3 (ACD Labs) was employed to determine the fragmentation pattern of isoleucyl-prolyl-proline tripeptide.

**Ile-Pro-Pro tripeptide assay.** Worms were age synchronized from 3-fold stage embryos and cultured on 60 mm NGM plate seeded with *E.coli* OP50 bacteria. A total of 50 μL Ile-Pro-Pro solution (containing 50 μM or 100 μM) was applied to plates with worms on adulthood day 1. Plates were left open under the hood to dry for 30 minutes and incubated for further 24 hours at 20 °C. BWM exophers were scored on day 2 of adulthood, after 24 h exposure to tripeptide. Synthetic, concentrated stock solution of Ile-Pro-Pro was stored in $H_2O$ at −80 °C. The stock was diluted to working solutions with M9 buffer.

**Transcriptome analysis.** *C. elegans* RNA extractions, library preparations, and sequencing were performed by Azenta US, Inc (South Plainfield, NJ, USA) as follows:

**RNA extraction.** Total RNA was extracted using Qiagen RNeasy Plus mini kit (Qiagen, Hilden, Germany) following the manufacturer's instructions.

**Library preparation with polyA selection and Illumina sequencing.** RNA samples were quantified using the Qubit 2.0 Fluorometer (Life Technologies, Carlsbad, CA, USA), and their integrity was assessed with the Agilent TapeStation 4200 (Agilent Technologies, Palo Alto, CA, USA). RNA sequencing libraries were constructed using the NEBNext Ultra II RNA Library Prep Kit for Illumina, following the manufacturer's protocol (NEB, Ipswich, MA, USA). In brief, mRNA was enriched using Oligo(dT) beads and subsequently fragmented at 94 °C for 15 minutes. First and second-strand cDNA synthesis was performed, followed by end repair and adenylation of the cDNA fragments at their 3' ends. Universal adapters were then ligated to the cDNA fragments, and libraries were enriched through a limited-cycle PCR with the addition of indexing sequences. The resulting sequencing libraries were validated using the Agilent TapeStation and quantified with both the Qubit 2.0 Fluorometer (Invitrogen, Carlsbad, CA, USA) and quantitative PCR (KAPA Biosystems, Wilmington, MA, USA). The libraries were multiplexed and loaded onto the Illumina NovaSeq 6000 platform according to the manufacturer's instructions. Sequencing was performed with a 2×150 paired-end (PE) configuration using v1.5 reagents. Image analysis and base calling were carried out using the NovaSeq Control Software v1.7. The raw sequence data (.bcl files) generated were converted into fastq files and de-multiplexed using the Illumina bcl2fastq software (version 2.20), allowing for one mismatch in index sequence identification.

**Sequencing data analysis.** Following quality assessment of the raw sequencing data, reads were trimmed to remove potential adapter sequences and low-quality nucleotides using Trimmomatic v0.36. The cleaned reads were then aligned to the Caenorhabditis elegans reference genome from ENSEMBL using the STAR aligner v2.5.2b. STAR is a splice-aware aligner that identifies splice junctions and integrates them into the alignment process to enhance the accuracy of read mapping. This step resulted in the generation of BAM files. Unique gene hit counts were calculated using the featureCounts function from the Subread package v1.5.2, considering only reads uniquely aligned to exon regions.

The gene hit counts were subsequently used for downstream differential expression analysis. DESeq2 was employed to compare gene expression levels between different sample groups. The Wald test was used to calculate p-values and log2 fold changes. Genes with an adjusted $p$ value < 0.05 and an absolute log2 fold change >1 were considered differentially expressed in each comparison.

**Scoring of Muscle Exophers and Fluorescence Microscopy.** Exopher quantification was performed as described in Turek et al. [18] and Banasiak et al. [24] (Bio-Protocol, detailed, step-by-step guide). In brief, exophers were scored using a Zeiss Axio Zoom.V16 stereomicroscope equipped with filter sets 63 HE and 38 HE. Worms were age-synchronized by isolating 3-fold stage embryos (hand-picked under a stereomicroscope) and maintained until day 2 of adulthood. On the second day of adulthood, worms were observed on NGM plates, and the number of exophers per freely moving worm was counted.

Figure 1f-i experiments were performed using different experimental setup. Exophers were scored using a Leica DMRBE microscope equipped with a 20× objective and RFP filter. Worms were age-synchronized by isolating 3-fold stage embryos or L4 stage larvae and maintained until day 2 of adulthood. During the infection period with PA14 or Db10 (or OP50 as a control), worms were briefly incubated at 25 °C for three hours. On day 2 of adulthood, animals were immobilized on 2% agarose pads with a 50 mM NaCl drop (10–20 μl), and the number of exophers per worm was scored.

**Scoring of neuronal exophers.** Neuronal exophers were scored using a Leica STELLARIS 8 FALCON confocal system with a 40× oil immersion objective. For imaging, worms were immobilized on 5% agarose pads (dissolved in M9), 10 μM tetramisole as an anesthetic.

**Optogenetics assays.** For optogenetic activation, both control and experimental 35 mm NGM plates seeded with *E. coli* OP50 bacteria were covered with 0.2 μM all-trans retinal (ATR) and left to dry at 20 °C and darkness for 16 hours. Ten to twelve age-synchronized worms were picked per plate from optogenetic strains expressing ReaChR in AIB at adulthood day 2. After a 3-hour incubation at 20 °C and in darkness, BWM exophers were scored. Next, experimental plates were placed under the source of white light (light intensity measured at 594 nm = 0.02 mW/mm2) for 4 hours. Control plates were shielded from light. The muscle exophers were counted immediately after illumination. Control and treated groups were randomized before the start of the experiment.

**Fluorescence Microscopy.** Representative fluorescence images in Fig. 1a, Fig. 2b, and Fig. 2f were acquired using a Leica STELLARIS 8 FALCON confocal microscope equipped with an HC PL APO CS2 40×/1.30 oil-immersion objective. GFP and RFP fluorescence were excited using 494 nm and 575 nm laser lines, respectively. Images in Fig. 1a were captured in LIGHTNING deconvolution mode to enhance spatial resolution. The left and middle panels show single optical sections, while the right panels present a three-dimensional reconstruction of the corresponding z-stack generated in LAS X 3D Viewer (Leica Microsystems). Images in Fig. 2b and Fig. 2f were acquired in standard confocal mode and represent maximum-intensity projections of z-stacks.

Representative fluorescence images in Fig. 7d, Fig. 7h, Fig. 7l, and Supplementary Fig. 7i were obtained using a Zeiss LSM 800 laser-scanning confocal microscope equipped with an EC Plan-Neofluar 40×/1.30 Oil DIC M27 objective. GFP and RFP were excited with 488 nm and 561 nm laser lines, respectively, and all panels represent maximum-intensity projections of confocal z-stacks.

For imaging, worms were mounted on 5% agarose pads and immobilized using either 10 μl of 0.05 μm polystyrene microspheres (PolySciences) or 25 μM tetramisole as an anesthetic.

**Generation of *sri-19*, *sri-36/39*, and *srr-6* mutant strains.** CRISPR/Cas9 genome editing was employed to generate the *sri-19* mutants (*sri-19(wwa15)* and *sri-19(wwa16)*) following previously established protocols[73]. The crRNA sequence used to target the *sri-19* locus was ATAATTGGTGGATATACAAATGG. Mutations were mapped to the first exon, resulting in frame shifts that likely produce molecular null alleles.

To create *srr-6* mutants, a previously described STOP-IN strategy[74] was used, which introduces an early stop codon within the first exon. The crRNA sequence used to target the *srr-6* locus was ACTC-CAAGTCCTGAAGTCGT, and the STOP-IN cassette sequence was GGGAAGTTTGTCCAGAGCAGAGGTGACTAAGTGATAA, flanked by AATCTTTTTCAACAATGATTACTCCAAGTCCTGAAGTCGT (left) and GGGTGACGAGGTTTCGTGTCCATCTATTATTGAAATACCT (right).

For the deletion of both the *sri-36* and *sri-39* genes, the entire 5968 bp region encompassing both genes (including the intercalated gene F33H12.7) was removed and replaced with a STOP-IN cassette (GGGAAGTTTGTCCAGAGCAGAGGTGACTAAGTGATAA). The flanking sequences were TGGTTACTTTTTATTTTTATTTTTC (left) and AATCCTAATCTAATATGAATCAATT (right). The crRNA sequences used for this deletion were ACAATGCTAGTTGTAGGTAC and ATTCATATTAGATTAGGATT.

All mutant strains were backcrossed twice with N2 wild-type worms to remove background mutations and verified using Sanger sequencing.

### Generation of *srr-6* transgenic rescue strains

**Plasmids cloning.** Rescue construct was generated by simultaneously inserting two fragments, promoter and gene of interest, into the linearized pMT27 vector using the SLiCE method (Seamless Ligation Cloning Extract). The pMT27 plasmid contains the *unc-54*-3′UTR and an *unc-119(+)* rescuing sequence. Restriction sites used for plasmid linearization were HpaI and PciI). Promoter region was amplified by PCR from plasmid templates. For the gene insert, cDNA from *C. elegans* was used as a template.

The SLiCE reaction (10 µl total volume) consisted of 100 ng of linearized vector and promoter/gene inserts in a molar ratio approximately equimolar to the vector, ensuring ~30 bp overlapping sequences between vector and inserts. The reaction mixture also contained NEBuffer 2.1 (NEB) and 0.5 µl of T4 DNA polymerase (NEB). The mixture was incubated at room temperature for 2 minutes, then placed on ice to stop the reaction, followed by bacterial transformation using the MH1 cloning strain. Plasmid sequences were verified for correctness using Nanopore sequencing.

**Transgenic lines preparation.** Transgenic strain was generated by germline transformation by injecting rescuing construct for *srr-6* gene (100 ng/µl of eft-3p::srr-6::unc-54-3′UTR and 10 ng/µl of pGLOW77(-myo-2p::mNeonGreen) from Addgene as co-injection marker) to *srr-6(wwa49)* mutant.

**Data Analysis and Visualization.** Data analysis was performed using Microsoft Excel and GraphPad Prism 9. Graphical representations of the results were generated in GraphPad Prism 9.

**Statistical Analysis.** No prior statistical methods were applied to determine sample sizes. Worms were randomly allocated to experimental groups, and all experiments were conducted in a blinded fashion for the datasets presented in Fig. 2d, i; Fig. 3a, b; Fig. 4a, d; Fig. 5a; Fig. 6a–j; Fig. 8a–k; Fig. 9b–k; Supplementary Fig. 2a, d; Supplementary Fig. 3a, b; Supplementary Fig. 4a, b; Supplementary Fig. 5a, b; Supplementary Fig. 7a–g, j; Supplementary Fig. 8a–h. Because the data were not normally distributed, we employed nonparametric statistical tests. For pairwise comparisons, the two-tailed Mann–Whitney tests were used, while Kruskal-Wallis tests followed by Dunn's multiple comparisons test were applied for multi-group comparisons. A *p* value of <0.05 was considered statistically significant.

### Reporting summary

Further information on research design is available in the Nature Portfolio Reporting Summary linked to this article.

### Data availability

The authors declare that the main data supporting the findings of this study are available within the article and its supplementary files. The RNAseq data generated in this study have been deposited in the GEO database under accession code GSE312284. Data underlying the figures is deposited at RepoOD with (https://doi.org/10.18150/YBLA2I) (https://repod.icm.edu.pl/dataset.xhtml?persistentId=doi:10.18150/YBLA2I). Source data are provided with this paper.

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

## Acknowledgements

Some strains were provided by the CGC, which is funded by NIH Office of Research Infrastructure Programs (P40 OD010440). We thank Justyna Polaczyk, Monika Woźniak, Natalia Wasiak, Małgorzata Śledź, Filip Kozłowski, and Tomasz Strumiński for assistance with worms maintenance; Adam Kłosin for discussions and comments on the manuscript. Work in the MT Laboratory was mainly funded by a National Science Center SONATA grant (2019/35/D/NZ3/04091 to MT) and additionally supported by a National Science Center SONATA BIS grant (2021/42/E/NZ3/00358 to MT). This work was supported by the Norwegian Financial Mechanism 2014-2021 operated by the Polish National Science Center (2019/34/H/NZ3/00691 to WP) and by the French National Research Agency (ANR-22-CE13-0037-01 to NP).

## Author contributions

Conceptualization: M.T. Data curation: M.T.; Formal analysis: K.K., A.S., S.V., R.P.S., M.R., M.T., W.P., N.P., H.B.; Funding acquisition: M.T., W.P., N.P.; Investigation: K.K., A.S., S.V., R.P.S., M.R., M.T.; Methodology: M.T., N.P., W.P., K.K., A.S., S.V., R.P.S., M.R., H.B.; Project administration: M.T.; Resources: M.T., N.P., M.R., W.P., H.B.; Supervision: M.T.; Validation: M.T., W.P., K.K., A.S., N.P., S.V., R.P.S., M.R.; Visualization: K.K., A.S., S.V., R.P.S, M.R., M.T.; Writing— original draft: M.T.; Writing—review & editing: M.T., W.P., K.K., A.S., N.P., S.V., R.P.S., H.B., M.R.

## Competing interests

The authors declare no competing interests.
