## [Transparent Peer Review file · Nature Communications]

Volatile and non-volatile pathogen cues shape host extracellular vesicles production in pre-infection response

Corresponding Author: Dr Michał Turek

Version 0:

Reviewer comments:

Reviewer #1

(Remarks to the Author)

In this manuscript by Kołodziejska et al., the authors found that both the volatile and non-volatile pathogen metabolites stimulate exophers generation in *C. elegans*. They dissected the chemosensation of volatile and non-volatile pathogen metabolites, including the chemosensory neurons as well as the potential GPCRs involved, and proposed the sensory signals are integrated by the AIB interneuron via NPR-9. Finally, they proposed the significance of such regulation in enhancing offspring survival against pathogen in the basis of compromising maternal survival. This work addresses an important mechanism on how species fight against pathogen infection in the aspect of population level, and unravelled the potential mechanism of metabolites perception. Besides, the manuscript is clearly written with a clear logical flow. However, there is still critical evidence lacking, that should be completed before publication:

1. Whether the regulation of exopher genesis in muscle is similar as in other tissue, such as neurons? How do the metabolites sensed by the chemosensory neurons and integrated by AIB neurons pass to muscle cells? What molecular mechanisms mediate this intercellular communication? Are there additive effects of volatile and non-volatile metabolites on exopher generation?
2. Why are multiple chemosensory neurons required for both volatile and non-volatile metabolites? For example, ASK, ADL, and AWC – do they use the same sets of GPCRs to sense both volatile and non-volatile metabolites?
3. For the neuron-specific function experiments, the authors should perform rescue experiments to confirm. For example, rescue SRI-19 in ADL neurons.
4. More evidence is required to confirm that AIB interneurons integrate both volatile and non-volatile signals via NPR-9. For instance, the authors could rescue NPR-9 in AIB neurons. Additionally, does activation of AIB neurons phenocopy the effect of volatile and non-volatile metabolites on exopher generation as well as offspring survival on pathogens?

Minor:

1. Why do the two sri-19 alleles present different behavior upon PA14-induced exopher generation?
2. Line 264, the citation should be Fig. 8d instead of Fig. 8c.
3. The PA14-induced exopher generation is even increased in the akt-1 mutants. The authors should at least discuss the potential reason.
4. The format of citation 44 should be corrected.

Reviewer #2

(Remarks to the Author)

Review report for manuscript "Pathogen threat proximity shapes host extracellular vesicle production in pre-infection response"

< Summary >

Kołodziejska et al. investigated how environmental signals from pathogens regulate the production of muscle-derived exophers in *C. elegans* hermaphrodite. The authors put forth a compelling set of findings on mechanisms of exophergenesis regulation in response to direct or indirect pathogen exposure:

- Dependency of exophogenesis on four immune response pathways (p38 MAPK, Insulin signaling, TFP-beta, IPR) by analyzing exopher production of the mutants
- Distinct effect of exposure to pathogen, pathogen supernatant or pathogen volatile to exophogenesis
- Identification of neuronal cells and molecules required for regulation of exophogenesis from muscles
- Effect of changed exophogenesis level upon survival under pathogen infection state

However, multiple conclusions throughout manuscript appear to extend beyond the experimental evidence. The details are addressed in comments.

The research report establishes, for the first time, that pathogen exposure can upregulate exophogenesis and release from muscle cells. Given that this is the inaugural report on pathogen-responsive muscle exophers, it is crucial to document their characteristics. Fluorescence imaging data and detailed descriptions are vital in this context. Although the methods for observing muscle-derived exophers are well-established by previous research, providing image data and comprehensive descriptions of the exophers is essential.

The authors extensively addressed the mechanisms regulating exophogenesis in response to volatile and non-volatile pathogen substances. While the data provide evidence for the roles of specific molecules and cells in the production of muscle-derived exophers, the connection between the findings is not fully established. Additionally, the authors explore the intriguing phenomenon of muscle-derived exophers influencing lifespan regulation under pathogen-infected conditions. However, the study currently lacks key experiments needed to address several pivotal questions.

To meet the standards of Nature Communications and ensure the claims are fully supported by the data, significant additional experiments would be required. These experiments should validate the key conclusions and provide a stronger link between the proposed mechanisms and the observed phenomena, thereby enhancing the impact and rigor of the study.

< Major comments by sections >

< Introduction >

1. Text line 88 "Our research demonstrates that worms can differentiate the proximity of a pathogen threat based on the type of metabolites they detect."

The results of this research show that animals can differentiate the pathogen threat via pathogen-derived supernatant or volatiles. However, whether they differentiate the "proximity" of pathogen based on the information has not addressed in this research.

< Figure 1 and associated text >

2. The consistency of results between Figure 1d and Figure 1e-g

In Fig. 1d, the authors suggest that the deficiency of genes involved in the immune response pathway results in significant changes in the amount of exopher production from muscles. However, the experimental results under the same conditions (i.e., OP50 condition) for *daf-16*, *tir-1*, and *dbl-1* mutants shown in Fig. 1e-g indicate apparently similar amounts of exopher production compared to wild-type animals. The authors should address the clear inconsistencies observed in the results. Clarifying whether these differences arise from the experimental design or other confounding factors is essential for supporting authors' conclusions of increased exopher production in the mutants upon pathogen exposure.

3. Figure 1i and text line 127 "This indicates that the *P. aeruginosa*-dependent increase in exopher production operates independently of the previously described embryo-maternal signalling, suggesting the involvement of a novel regulatory pathway."

It is difficult to assert that the exopher production in response to PA14 is independent of embryo-maternal signaling only based on the observation that the increase of exopher production was not accompanied by the increase of egg retention. To substantiate the claim, it is crucial that experiments be conducted to determine whether depleting embryos affects exopher production. Such studies are necessary to definitively conclude the independence of these processes.

< Figure 2 and associated text >

4. Figure 1i demonstrated that *Pseudomonas aeruginosa* does not lead to an increase in embryo accumulation in utero, contrasting sharply with the effects of pathogen volatiles as illustrated in Figure 2c. These findings raise the possibility that non-volatile pathogenic stimuli may suppress the embryo accumulation typically induced by exposure to pathogen volatiles. The authors are encouraged to provide a detailed explanation for these divergent outcomes.

5. In this manuscript, the terms 'bacterial supernatant' and 'bacterial non-volatile metabolites' appear to be used interchangeably. However, as bacterial supernatant also contains a variety of substances including extracellular vesicles, which are not metabolites. Thus, it is recommended for the authors to define the range of the terms or discuss the limitation of the experiments. A similar level of explanation is also preferred for the bacterial volatiles used in this research.

< Figure 3 and associated text >

6. Figure 3a, c and text line 152 "Our results indicate that none of the tested pathways—including insulin, TGF β , p38 MAPK, and IPR—significantly contribute to the volatile metabolite-dependent increase in EV production (Fig. 3a and c)."

It might seem as though the authors are completely excluding the possibility that the pathways contribute to the pathogen volatile-dependent increase in EV production. Considering potential for other scenarios, such as redundant contributions from these pathways, a more nuanced tone that leaves some room would be more appropriate.

7. Figure 3b and c. The requirement of several immune response pathways for this EV production raises interesting questions. Are these pathways working together to regulate exophergenesis? Discussing the interrelationships among these pathways within the mechanism regulating exophergenesis could yield valuable insights.

8. Text line 161 "... bypassing these pathways entirely for airborne signals."
Similar with Major comment 8, excluding all the possibilities entirely is too definitive.

< Figure 4 and associated text >

9. The authors have not detailed the information for genetic ablation of each neuron in Methods section. Although the information is included in Supplementary Table 2 "Caenorhabditis elegans strains", a brief description of the type of ablation within Methods and/or Results with reference to Table 2 would greatly enhance reader accessibility.

10. Figures 4a-c demonstrate that the ablation of ADL or ASK neurons not only abolishes the upregulation of exopher production in response to pathogen volatiles or supernatant, but also the ablation itself considerably reduces exopher production already. The authors are not addressing this aspect.

11. Figure 4a-c identify neurons required for upregulation of exopher production via stimulation from pathogen volatile or supernatant. The results are indicating that lack of ADL or ASK neurons abolishes the upregulation. However, the data also reveal that ablation of each neuron itself results in a considerable reduction in exopher production, implying that the exopher production is constantly downregulated already by the neuronal ablation. Additionally, ASK-ablated strains show prolonged survival irrespective of pathogen volatile exposure experience, as depicted in Figure 8b. These findings suggest an underlying complexity in the role of these neurons towards exophergenesis regulation that likely precedes their involvement in pathogen response. Given these complexities, it would be beneficial for the authors to further explore and discuss the implications of neuronal ablation on general physiological functions. Clarifying these intricate relationships will enhance the manuscript's depth and help substantiate the authors' conclusions.

12. In Figure 4, the authors broadly investigated the contribution of multiple sensory neurons to the regulation of exopher production from muscles. It is noteworthy that these neurons are part of the "Hub-and-Spoke" system, with RMG functioning as the central hub (Ref. 35 and Macosko EZ et al., 2009). This raises the possibility that RMG may play a central role in regulating exophergenesis, potentially through its interactions with the sensory neurons studied in this research. Investigating exopher production in neuron-ablated animals including RMG will enhance the understanding of circuit mechanisms underlying the nervous system's regulation of exophergenesis from muscles to substantiate novelty of this study.

< Figure 7 and associated text >

13. The arrangement of panels in Figure 7 is somewhat challenging to interpret. It may be more effective to reorganize the figure by grouping panels according to their respective conditions. Additionally, while Figures 7, 3, and 4 share a similar experimental design, Figure 7 includes statistical comparisons under basal conditions, which are not addressed in Figures 3 and 4. Since the basal phenotypes of the mutant animals used could significantly impact the interpretation of their pathogen substance-responsive phenotypes, providing information on basal phenotypes is strongly recommended.

14. Text line 230 "Notably, the basal level of exophergenesis in GPCR mutant worms under standard conditions remained unchanged (Fig. 7a, e, i). These findings collectively indicate that these receptors are critical for pathogen-induced exopher production, but not under non-stressed conditions."

It is stated that the basal level of exophergenesis in GPCR mutant animals remains stable under standard conditions; however, the results from sri-36/39 (wwa57), ssr-6 (wwa48), and ssr-6 (wwa49) show significant variations. The authors need to reconcile these observations with their initial claims. Furthermore, as the mutants itself, without pathogen stimuli, showed significant change of exophergenesis, it is misleading to claim "that these receptors are critical for pathogen-induced exopher production, but not under non-stressed conditions.". The authors must clearly delineate the observed phenotypes and articulate their contributions in light of the results obtained.

< Figure 8 and associated text >

15. The manuscript currently lacks a detailed description of the methods used for the parental survival assay. This information should be added to methods and described briefly in figure legends.

16. Figure 8a presents an illustration that is challenging to interpret, despite a detailed legend that explains the experimental scheme. For instance, the notation 'exophers scored @AD2 F1 embryos collected' explains the procedure for Figures 8e and f. However, at first glance, it appears to pertain to the experiments shown in Figures 8b-d. To enhance clarity and ensure that readers can easily follow the experimental design, the illustration requires improvement.

17. Figure 8b. The effect of ASK neuron ablation on the prolonged survival of *C. elegans* under *P. aeruginosa*-infected conditions was previously reported by Venkatesh et al. (Life Science Alliance, 2023). The authors should reference this study to provide appropriate context and acknowledge prior findings.

18. Figure 8b illustrates that exposure to pathogen volatiles, which increases exopher production from muscles, decreases the survival rate under *P. aeruginosa* infection. The manuscript notes that prolonged survival due to ASK neuron ablation,

which increases exopher production (Fig. 4a-c), supports the inverse correlation between exopher production and survival under infection. These findings raise critical questions. Yet, several aspects need to be addressed to fully support the conclusions claimed by the authors. These include:

a. It is evident that ablation of the ASK neuron downregulates exopherogenesis from muscles (Fig. 4a-c). This raises intriguing questions, could the survival-extending effect observed with ASK ablation be attributed to the altered exopherogenesis? What will be the roles of the ASK neuron in regulating both exopherogenesis and survival? While the *rme-2* RNAi experiment in Figure 4c supports the connection between exopher production and survival, the degree of survival extension observed with ASK ablation appears to differ from that of *rme-2* RNAi, implying a contradiction, which raises a concern that ASK's pathogen susceptibility effect may be largely through pathways other than exophers. Addressing this crucial question will require additional experiments. For example, testing *rme-2* RNAi in ASK-ablated animals or testing pre-exposure of pathogen under *rme-2* RNAi condition could provide valuable insights for understanding the complex biology.

b. Previous research by Alcedo and Kenyon (Neuron, 2004) suggested that ASK has a longevity-promoting effect based on combinational cell ablation experiments conducted under pathogen-free conditions. Interestingly, under pathogen-present conditions, ASK exhibits longevity-reducing effects. Given these contrasting observations, it would be valuable for the authors to explore and discuss how ASK and ASK-dependently regulated exophers contribute to longevity mechanisms in these opposing ways, depending on the presence or absence of pathogens.

19. Figure 8c indicates that the knock-down of the *rme-2* gene, known to decrease exopher production, also extends animal survival under infection, supporting the notion that reduced exopher production might enhance survival. Intriguingly, previous research has shown that the exopherogenesis event from touch neuron extends lifespan (text line 332 and Ref. 16), highlighting a potential protective role of exophers in non-pathogenic contexts. This raises a crucial question, will enhancing exopherogenesis from muscles through factors other than pathogen exposure, such as the knock-down of *vit-1* or *emb-27* genes, impact the survival of the animals under the condition tested in Figure 8c? This experiment can significantly strengthen understanding for the roles of muscle-derived exophers in survival under pathogenic conditions.

20. Figure 8c-d. The manuscript lacks clarity on whether the animals are exposed to pathogen volatiles prior to the survival assay. This critical information is not described in the legends, nor is it detailed in the Methods section. The information should be addressed in Methods and depicted briefly in legends.

21. Figure 8d and text line "Additionally, *sri-19* mutants, which do not show increased exopher production upon contact with *P. aeruginosa* non-volatile metabolites, survived significantly longer during infection (Fig. 8c) which demonstrates that on mechanistic level survival is interconnected with the regulatory elements for EV production."

The authors examined exopher production in *sri-19* mutants in response to pathogen supernatant or volatile as shown in Figure 7b and c. Assuming that the protocol in Figure 8d adheres to that outlined in Figure 8a, the primary factor influencing the survival of the mutant animals under pathogen infection would be the altered level of EV production induced by prior exposure to pathogen volatiles. However, Figure 7b indicates that EV production can be significantly upregulated even in the absence of the *sri-19* gene. These results raise a few questions:

a. Does pathogen infection significantly enhance EV production in *sri-19* mutants?
b. Which will dominantly impact on the survival of *sri-19* mutants, pathogen supernatant or pathogen volatile?
Answering these two questions could provide a fundamental basis to demonstrate that survival is mechanistically interconnected with the regulatory elements of EV production.

22. In the study, survival assays utilized different initiation stages for P0 and F1 generation animals. P0 animals are exposed to *P. aeruginosa* starting from the day 2 adult stage, whereas F1 animals begin exposure at the L4 stage. It is crucial for the authors to justify the choice of different stages for these experiments (why not adult day 2 for L1 animals) or clarify whether the pathogen's effects on the lifespan of *C. elegans* are consistent across these developmental stages.

23. Similar to Figure 8 discussed above, Fig. 8e and f lack sufficient labels to depict different conditions. Figure legend lacks information and it should describe what are shown in different subpanels. Additionally, an explanation of the error bars for Fig. 8e and f should be provided to ensure clarity.

24. The authors have sorted the P0 population into two categories based on criteria outlined on text line 280. However, the rationale for selecting these specific criteria is not provided. Detailed explanations are necessary for:

a. Why the numbers of exophers (7 and 10) were chosen as criteria for categorizing the biological replicates into high/low exopher groups. Although Supplementary Figure 2 and the associated text provide some basis for choosing these criteria, the process for selecting these specific numbers is not described. Furthermore, the authors should explain why the criterion changed from the median values 6 and 10 in Fig. S2 to 7 and 10 in Fig. 8e and f. This is a critical problem that could be a potential violation of neutrality in the statistical tests.

b. How the populations were handled for breeding and observation of exophers? Providing experimental details is crucial because this is a set of experiments on which authors' major claim "volatile-induced exopher production enhances offspring survival against pathogen" is based.

c. We assume that Fig 8e shows the results of high-exopher group and Fig 8f show the low-exopher group. Then what are shown on the left end panels of Fig. 8e and f? Which threshold was used for which test? Why are there data that are both lower and higher than the thresholds?

25. The left panel of Figures 8e and f represents exopher production across all examined individuals from multiple replicated groups. However, to clearly demonstrate that each P0 group used to generate categorized F1 groups meets the established criteria, it is essential to post the specific amounts of exopher production observed in each individual population. Providing this data will enhance transparency and allow for a more precise assessment of the experimental results.

26. The authors propose that the increased exopher production due to exposure to pathogen volatile decreases the survival of P0 animals under pathogen exposure, while the survival of the F1 generation under the same conditions is enhanced at the cost of the parent generation's survival. Figures 8e and 8f aim to demonstrate this hypothesis by comparing WT and WT+PA14 populations of P0 and F1 animals via survival assays.

However, when analyzing the results within populations — for example, comparing WT animals above the criterion (Fig. 8e, those producing more exophers) with WT animals below the criterion (Fig. 8f, those producing fewer exophers) — the results reveal conflicting phenotypes. Specifically, P0 animals above the criterion (Fig. 8e, middle panel) exhibit longer survival than those below the criterion (Fig. 8f, middle panel), contradicting the notion that increased exophers reduce the survival of P0 animals under pathogen conditions proposed by the authors. Conversely, F1 animals derived from P0 animals above the criterion (Fig. 8e, right panel) show shorter survival than those derived from below the criterion (Fig. 8f, right panel), which also conflicts with the authors' interpretation. The trend is consistent across WT and WT+PA14 populations in Figures 8e and 8f.

These findings not only conflict with the authors' major novel claim in the research, but also contradict with the results presented in Figures 8b-d. The authors must address this significant discrepancy transparently and provide a detailed explanation of their rationale for the data presentation and interpretation. Furthermore, a statistical analysis of the data within the same population, specifically comparing subgroups divided by the criteria as mentioned earlier, is necessary.

Additionally, it is crucial to demonstrate that the survival-decreasing effects of pathogen volatile-responsive exophers in P0 animals under pathogen exposure are consistently observed within the experimental framework of offspring survival. Clarifying these points is essential to resolve the discrepancies and ensure the validity of the conclusions.

< Discussion >

27. Text line 316 "This distinction implies that exophers may constitute distinct subtypes of EVs with unique molecular compositions and functions."

Throughout the results, the authors have effectively demonstrated that the contribution of immune pathways to exopher production varies depending on pathogen-derived stimuli. However, the research has not explored whether the EVs produced differ between various conditions or treatments. Instead, the survival assay results for ASK-ablated animals from Figure 8b, which ablation constitutively reduces EV production regardless of the type of pathogen-derived stimulus (Figure 4a-c), could hint at a functional uniformity of the produced EVs. The potential heterogeneity might be suggested by assessing the distinctiveness of the generated exophers.

28. Text line 331-337 "Previous studies have shown that increased exopher release can benefit the organism by removing protein aggregates, extending lifespan, or promoting faster offspring development. However, when exopher release is elevated in response to pathogen exposure, we observe an opposite effect, leading to decreased survival upon infection. This suggests that exophers, or possibly EV in general, may function under a hormetic response: while a moderate increase in EV production appears beneficial, an excessive increase may overwhelm the organism, negating any positive effects." The authors focus their investigation on exophers, a specific subtype within the diverse family of extracellular vesicles. Considering the well-established diversity of EVs in origin, characteristics, destination, and function, it is premature to extend the findings of this study to all EVs. Furthermore, it is important to note that Reference 16 suggests distinct functions for neuron-derived exophers compared to the muscle-derived exophers examined here. This observation raises a fascinating question; why do neuron-derived exophers and muscle-derived exophers appear to have opposite effects on survival? This intriguing contrast merits further discussion and exploration within the manuscript, rather than limiting the scope of potential implications.

29. Text line 345 "In contrast, our study involved prolonged exposure to a mixture of volatile metabolites from pathogens, which may trigger different physiological responses."

The authors emphasize the potential importance of i) the duration of exposure and ii) the specificity of volatiles in mediating the physiological responses of *C. elegans* to *P. aeruginosa*. Indeed, Ref. 31 demonstrates that short-term exposure (2 hours) to 1-undecene can improve survival under pathogen exposure, which contrasts with the effects indicated in the current research. However, to suggest the possible influence of the factors on the variation in physiological responses, the authors should provide additional supporting evidence. For example, they could compare whether varying the duration of exposure to 1-undecene leads to different effects or not. Additionally, discussing which specific volatile compounds from pathogens could underlie the observed physiological responses in this research would significantly enhance the depth and scope of the discussion.

< Minor comments >

1. Text line 116 "However, when immune response mutants were exposed to pathogens, there here was a consistent increase in exopher production across all tested mutants, as observed in wild-type worms (Fig. 1e-g)."

It could be read as all of the tested mutants are showing consistent increase in exopher production in response to pathogen exposure. However, in the following sentence, the authors mention that the *cul-6* mutant is an exception. Therefore, the phrase 'across all tested mutants' may not accurately reflect the data and could be revised to account for the exception observed in the *cul-6* mutant.

2. Text line 121 "Therefore, the pathogen-induced EV modulation does not rely on canonical immune response pathways."

Given that the authors have not fully tested for redundancy in the immune response pathways, the claim that pathogen-induced EV modulation 'does not rely on canonical immune response pathways' could be reconsidered. A more cautious interpretation may be warranted until potential compensatory mechanisms are explored.

3. Text line 124. egg retention

4. Fig. 2a-c. The conditioning duration has changed from 3 hours for pathogen exposure (Fig. 1) to 48 hours for pathogen volatile or supernatant exposure (Fig. 2; starting from L2 as described in Methods). The authors should at least comment the ground for choosing 48 hours as pathogen volatile/supernatant conditioning to trigger exopher production. Is 3 hours too short for the volatile/supernatant to exert effect?

5. Figure 2a. The illustration presents an ambiguity; the stacked two plates in fourth row appear to be oriented with both top sides facing each other, which could potentially obstruct the passage of volatiles. If this is the case, the arrangement might impede the intended experimental conditions and affect the reliability of the results.

6. The authors should standardize the terminology used throughout the manuscript to maintain consistency. Specifically, the terms 'embryo accumulation' and 'egg(s) retention' are considered to refer to similar phenomena but are used interchangeably. If the two terms are indicating different phenomena, describing the difference will help readers understanding better. Also, a unified terminology would enhance clarity and improve the manuscript's readability.

7. Text line 145. EV in response or exopher in response

8. Figure 2f and text line 148 "Notably, this increase in exopher production do not depend on increased egg retention in the worm's uterus (Fig. 2f)."

Deciding that they "do not depend on" might be hasty conclusion. Although it is true that the increase in exopher production occurred without increased egg retention under exposure to pathogen supernatant, this does not exclude a possibility that egg retention could affect the exopher production. Thus, "do not require" might be more accurate for this case.

9. Figure 4a and text line 174 "While ASI, ASH, and AWB neurons are crucial for the supernatant-induced increase they do not play a role in volatile metabolite dependent increase in EV production (Fig. 4a-c)."

In figure 4a, with ablation of ASH or AWB, while the amount of produced exophers is still significantly increased by pathogen exposure, the amount of change seems decreased compared to intact animals. To clarify this point, the magnitude of the pathogen-induced increase in EV production could be statistically compared between wild-type and ablated animals. This analysis could involve employing a two-way ANOVA approach, or its non-parametric variant, the Aligned Rank Transform (ART) ANOVA.

10. In this research, ADL neuron is functionally ablated by *hlh-4(tm604)* III mutation. According to Masoudi et al., 2018, in PLoS Biology, this mutation disrupts the 'identity features that define functional features of the ADL neuron', without physically destroying the neuron. Since this approach differs from the methods used to ablate other neurons in the research, it is crucial to clearly delineate this discrepancy in the Materials and Methods section.

11. Text line 191 "... receptor abolishes the ...".

Figure 5c shows that while the increase in pathogen supernatant-dependent EV production in AIB(-) strains is not statistically significant, there is a discernible increase in average levels. This implies that availability to regulate EV production is not completely eliminated. more accurate description of the observed effect might be 'significantly reduced,' which would more precisely reflect the experimental data.

12. Text line 194 A period is missing after "signals".

13. Text line 204 "Additionally, our prior RNA sequencing, which identified the GPCR str-173 as a regulator of socially mediated exopher production, also highlighted sri-36 and sri-39 as top candidates, further supporting their potential as strong candidates for validation⁴⁴."

The authors should elucidate how their previous RNA sequencing efforts have identified sri-36 and sri-39 as top candidates for pathogen signal receptors. It is essential to clearly describe the criteria and analysis methods used to highlight these genes, as well as any supporting data that distinguishes them from other candidates. Providing this detail will enhance the reader's understanding of the basis for these selections and strengthen the overall findings of the study.

14. Text line 207 "To investigate the roles of sri-36 and sri-39 in exopher regulation, we employed CRISPR/Cas9 technology to generate a mutant with a deletion encompassing both receptor-encoding genes, as they are located only 1.9 kb apart, with one gene situated between them (Supplementary Fig. 1)."

The proximity of two gene, sri-36 and sri-39, upon genome accounts for the efficiency of simultaneous deletion to generate double (actually triple) mutant, but it cannot be the ground for not generating single mutants. Use of this mutant deleting multiple genes leaves a possibility that the phenotypes are attributed to only one of the mutated genes, but the rests are not contributing. The authors should discuss this limitation.

15. Text line 221 (Fig. 6d) and text line 224 (Fig. 6c).

16. Figure 7d and Text line 234 "The wormScarlet CRISPR/Cas9-mediated transcriptional knock-in of the sri-19 gene revealed its specific expression in ADL neurons (Fig. 7d)."

Figure 7d describes expression of 7TM receptor-encoding genes, sri-19 in ADL neuron, which is annotated based on position and scRNAseq data from Reference 53. However, the manuscript lacks a description of how the annotation of cells based on position was executed. If promoter-driven fluorescence proteins were used as comparative markers for this

annotation, it would be helpful for the authors to include this information.

17. Text line 240 “Additionally, the expression of the srr-6 receptor in neurons of the retrovesicular ganglion (RVG), alimentary tract, and excretory system (Fig. 7l), as well as the expression of sri-36 and sri-39 in the alimentary tract (Fig. 7h), suggest that one or more of these tissues may also play a role in regulating exopher production in muscles.”
Based on the current experimental results, the claim that these tissues are involved in the regulation of exopher production may need to be presented with more caution. Or, providing stronger supporting evidence, such as a rescue experiment, could help substantiate this interpretation.

18. Text line 247 “we aimed to investigate the underlying reasons for the release of these extracellular vesicles when worms detect the pathogen in their environment.”
The results in figure 8 are deciphering the effect of released exophers, not underlying reason (or cause) of released exophers.

19. Text line 252 “To further explore this, ...”.
It is difficult to follow what “this” is referring to. To improve clarity and ensure the narrative is easily understandable, specifying explicitly what 'this' will be helpful.

20. The authors describe the age synchronization of animals by isolating embryos at the pretzel-stage or 3-fold stage. To enhance readability and accessibility, especially for readers not familiar with *C. elegans* development, it is recommended to standardize the terminology used to describe these stages, or clarify if they refer to the same developmental stage. Additionally, including a brief description of the method used for isolating pretzel-stage embryos would further aid in understanding.

21. Figure 8b. It is ambiguous in this figure whether “+ PA14” in inlet legend indicates exposure to pathogen volatile before assay, or exposure to pathogen during survival rate assay.

22. The explanation provided for Figure 8e and f in the current manuscript is insufficient. Details regarding the treatment of F1 animals during the experiment remain unclear, even with the provided illustration in Figure 8a. Including a more detailed explanation of the experimental procedures would significantly enhance the readers' understanding.

23. Text line 293 “1. Pathogen-induced exopher production: Exposure to live *Pseudomonas aeruginosa* PA14 or *Serratia marcescens* Db10 induces exopher production in *C. elegans* body wall muscles.”
In Figure 9, this fact is indicated by the green upward arrows, but it should be more stressed because it is the main aspect studied in this work.

24. Text line 312 “Our findings suggest that, although exophers produced in response to different stimuli may appear similar in number, the distinct signalling pathways involved indicate potential differences in their roles or contents.”
Although the research has thoroughly explored the mechanisms underlying exopher production in response to various pathogen stimuli, there is no positive reason for assuming differences in their roles and contents. Thus, the work does not “indicate” such potential.

25. A reference is required at text line 344 where it states 'In previous studies.'

26. Text line 347 “The involvement of multiple G protein-coupled receptors (GPCRs) in regulating exopher production suggests that exopherogenesis is not triggered by a single metabolite but by a complex cocktail of pathogen-derived signals.”
Actually, that might not be the only explanation. There are other possibilities to consider, for example, that the GPCRs are responding to the same substances.

27. Text line 354 “exophers and other EVs”
Given that this research has not evaluated other types of EVs, it is challenging to conclude that the other EVs are crucial in mediating the organism's response to environmental stressors.

28. Text line 416 "(references)"
Needs be replaced.

Reviewer #3

(Remarks to the Author)

Reviewer #4

(Remarks to the Author)

In the manuscript by Kołodziejska et al, the authors track the effects of exposure to pathogenic bacteria on production of muscle-derived giant vesicles (exophers). The authors find that exposure to *Pseudomonas* and *Serratia* pathogenic bacteria increases exopher production rate. Effects of signaling pathways that can impact innate immunity were shown, with Insulin-

like signaling pathway mutant *daf-16* reducing exopher production in all bacterial backgrounds, pathogenic and control. Its negative regulator *akt-1* is tested, which might be expected to further increase exopher rates upon exposure to PA14 and Db10, but one cannot tell given the statistics shown. Please add such analysis. The authors should also acknowledge that these signaling pathways are not exclusive to innate immunity and could mediate exopher effects via other changes to the animals independent from immunity.

tir-1 mutants affecting the p38 MAPK signaling pathway appeared to increase exopher production on OP50 (Fig 1d). It is unclear if *tir-1* further increases PA14 and Db10 effects (Fig1f) since the comparison to WT was not statistically tested. In addition, the OP50 result appears different in the two panels. Fig 1d shows *tir-1* mutants increasing exopher production on OP50, while Fig 1f appears to show no effect of the same experiment (although the statistical comparison vs WT is not shown). *pmk-1* mutants in the same p38 pathway did not affect exopher levels on OP50 or suppress PA14 or Db10 driven increases. The way the comparisons are done does not allow the reader to tell if *tir-1* mutants further increase PA14/Db10 effects. Please add the relevant statistical comparisons.

There are similar problems with interpretation for the other mutants *dbl-1* and *cul-6* on PA14 and Db10, as they are only statistically tested against OP50, but not against WT exposed to the same bacteria. This should be tested.

Pseudomonas PA14 mediated increases in muscle exopher production could be induced by non-volatile and volatile metabolites, with clear genetic differences in response to these different stimuli. They also differ in neuronal requirements. Volatile exposure to PA14 produced some egg retention (a known exopher inducer) while non-volatile metabolites (conditioned media) did not affect egg retention. The authors should modify lines 127-129 "This indicates that the *P. aeruginosa*-dependent increase in exopher production operates independently of the previously described embryo-maternal signalling" accordingly. Fig 1 showed that Db10 exposure also produced egg retention. PA14 and Db10 effects could be further tested using sterile animals to determine if exposure can overcome empty uterus effects.

dbl-1 mutant appears to suppress exopher production on OP50 control and partially suppress PA14 volatile effects (Fig 3a). Please add appropriate statistical test for this and modify accordingly line 121: "Therefore, the pathogen-induced EV modulation does not rely on canonical immune response pathways." The effects on OP50 control make overall interpretation for *dbl-1* difficult, although some studies suggest that OP50 is itself mildly pathogenic.

The authors test for effects of neuron ablation seeking to gain insight into sensing and propagating effects of bacterial metabolites. ADL, ASK, or AWC neurons were important for PA14 response in volatile and non-volatile assays. There is also some issue with interpretation here because these ablations also appeared to block most exopher production on OP50 control. If the neuron (or neuronal receptor such as *npr-9*) is required for basal exopher production it is difficult to know if it is directly involved in metabolite sensing and downstream signaling or if it is just generally required for exopher production. Please add statistical tests comparing samples exposed to the same bacteria but with intact vs ablated neurons.

The previously shown requirement for ASK in pheromone-mediated exopher production also clouds interpretation, although this is mitigated by lack of requirement for AQR/PQR/URX that are important for pheromone response.

The authors argue for a connection between volatile metabolite exposure, exopher production, and survival of P0 and F1 generations upon full PA14 exposure. They show that P0 exposure to PA14 volatile metabolites exhibited reduced survival of P0's during bacterial infection, inversely correlated with exopher production. They also showed that worms with ablated ASK neurons, that produce fewer exophers than wild type worms, survive PA14 infection longer than wild type. They also showed that knockdown of the RME-2 receptor, which produce very few exophers, also survive *P. aeruginosa* infection longer than wild-type worms. They also found that *sri-19* mutants, which do not show increased exopher production upon contact with non-volatile metabolites, survived significantly longer during infection. These are very interesting results. The authors also found interesting results for the F1 generation, with high P0 exopher generation correlated with F1 animals with improved survival on PA14. Overall interpretation is still somewhat limited for these results as they remain correlations. The authors should test if these perturbations affect embryo retention in the uterus. Effects on reproduction could be controlling exopher level rather than exopher level controlling reproduction. *rme-2* is certainly very important for brood size.

The paper does not characterize which muscles produce exophers in response to pathogens. Do exophers appear randomly from all body-wall muscle, or is production limited to a particular body region, for instance near the uterus?

The authors should more directly compare their results here with those from their recent paper on pheromone-induced exophers.

Have the authors tested OP50 conditioned media in the non-volatile experiments? It appears they used simple LB as a control rather than control conditioned media.

Please check Materials and Methods for accuracy. No micrographs are shown in this paper, but they are discussed in M&M.

Fig 1C should show negative regulation of DAF-16 by AKT.

Why do Figures 1i and 2c seem in conflict?

Fig 8 Legend is overstated: "Exophers levels are affecting survival rate post-infection." This implies cause-and-effect, which is not demonstrated. A more accurate title would be "Exophers levels correlate or inversely correlate with survival rate post-

infection.”

In many experiments the changes emphasized in the interpretations are statistically significant but represent a small change, in some cases a difference in exopher number from 9 to 11 or 7 to 10 is statistically significant while other experiments conclude no significance between similar median comparisons, 11 to 13 (Fig 3B). Which cases are biologically important differences? Overall, the statistics are misleading, graphs and text should indicate mean values for each experiment with standard error of the mean. In this current submission, the authors display interquartile ranges emphasizing the range around the median and are most appropriately used with skewed datasets that contain outliers (while these datasets appear to be normally distributed, with no outliers indicated). Moreover, precise trial numbers and sample sizes are not provided for each experiment (often a range is given) and several experiments in Fig 1 only have two biological replicates rather than the standard minimum of three. It is not clear from the text or figure legends if all data is shown, or if one representative dataset is shown in the main figure. Replicate experiments should be provided in the supplement to allow for data transparency regarding natural variation between experiments. Furthermore, it might be appropriate to evaluate these data as log fold changes rather than comparing averages to more rigorously determine if these small differences are meaningful, the authors should consult a biostatistician to guide them in this analysis.

Overstatement: line 265-266 “Given that muscle exophers have been shown to improve animal reproduction and boost offspring development”. Previous work showed a correlation, not a cause-and-effect relationship.

The model at the end of Results should be in the Discussion.

Several mutants in the TGF-Beta pathway have organismal-wide phenotypes including small body size (DBL-1). Data examining this mutant for exophers should be normalized to account for the difference in body size between the WT and dbl-1(nk3) animals.

Additional controls should be performed to address the effect of dietary restriction on muscle exophers (a known effector of egg retention and exophers in other tissues). Both volatile and non-volatile metabolites could have a profound impact on feeding which in turn would muddy the conclusions from these experiments and need to be addressed experimentally.

Summary tables for P-values are nice. They could be improved by including the WT control alongside experimental results (Fig 5D).

Fig 7e – The legend states worms lacking SRI-36 and SRI-39 receptors are comparable to WT but data shown reports a significant difference ($p=0.0016$) between WT and sri-36/39(wwa57) mutants. This result is at odds with both the main text and the figure legend.

Reviewer #5

(Remarks to the Author)

The manuscript provides valuable insights into extracellular vesicle (EV) production in response to pathogen-derived metabolites. It highlights a regulatory network influenced by volatile and non-volatile metabolites, with implications for host-pathogen signaling and innate immunity. While the study is innovative and relevant, several areas require clarification and additional data to strengthen its impact.

1. The study mentions key immune pathways such as insulin, TGF β , p38 MAPK, and IPR, but excludes other established regulators like SKN-1, ELT-2, and HLH-30. Exploring their roles could enhance the analysis and provide a more comprehensive understanding of the mechanisms driving EV production. The selection criteria for these pathways should be justified.
2. Figures 1D and 1F show inconsistent trends in EV production among mutants, such as tir-1(tm3036) and dbl-1(nk3). These discrepancies need clarification to ensure the validity of the conclusions drawn from these experiments. Graphical improvements, such as merging related panels or adding clearer annotations, would improve readability.
3. Tissue-specific roles of key regulators like NPR-9 and GPCRs (e.g., SRI-19, SRI-36, SRI-39, and SRR-6) in immune activation and exopher production should be further investigated. Rescue experiments could identify their tissue-specific functions, providing insights into their biological significance.
4. While the study demonstrates offspring survival benefits linked to increased maternal exopher production, it does not explore these effects beyond the F1 generation. The authors might want to investigate potential intergenerational transmission into the F2 generation.
5. The specificity of the observed EV responses to pathogens versus general stressors remains unclear. Including controls for non-pathogenic stressors would help establish whether the responses are pathogen-specific.

Minor Comments

1. The Introduction contains some redundancy between the second and third paragraphs, particularly in discussing the advantages of using *C. elegans* as a model organism. Streamlining this section would enhance conciseness.

2. Several references lack complete citation details, such as volume, page numbers, or correct publication years (e.g., reference #3 and #42).
3. Minor typographical errors, such as missing punctuation on line 194, should be corrected to improve overall readability.

Reviewer #6

(Remarks to the Author)

Version 1:

Reviewer comments:

Reviewer #1

(Remarks to the Author)

The authors have made tremendous efforts during the revision process to address the questions raised by the reviewers, and overall they have successfully addressed most of my concerns. However, several minor points still need to be revised before publication:

1. I agree with the other reviewer that representative images of muscle-derived exophers should be included in Figure 1.
2. The authors should clarify how they confirmed that sri-19 is expressed in ADL neurons, and sri-36/39 in ASI and SKI neurons, in the absence of specific markers.
3. Regarding my previous question (#5) on rescuing SRI-19 in ADL neurons: although the authors attempted rescue experiments, the results were unexpected. I appreciate that they have already made an effort, but since the evidence for SRI-19 function in ADL neurons remains inconclusive, the authors should either (i) use an ADL reporter to demonstrate co-localization with wrmScarlet-SRI-19 fusion protein, or (ii) confirm differential wrmScarlet-SRI-19 expression in wild-type animals with varying exopher numbers.
4. In Figure 3b (middle panel), there is a clear trend toward increased exopher production in pmk-1 mutants upon PA14 non-volatile excretome treatment, although the difference is not statistically significant. The authors should increase the sample size (N) to clarify this effect.
5. Baseline levels appear inconsistent across figures. For example, in Figure 6f and 6g, the wild-type values differ markedly (~8 vs. ~2). A similar issue is present in Figure 5a, where the wild-type values vary across panels. The authors should repeat the experiments or reasonably address these discrepancies.
6. The lack of an additive effect between volatile and non-volatile metabolites should be discussed explicitly.
7. I agree with the other reviewer that a brief description of the genetic ablation methods should be included in the Methods section, even if a citation is provided.
8. I recommend revising the title. Since the study includes both volatile and non-volatile metabolites, the title should reflect both.
9. I also recommend revising the abstract. The current version is overly long and complex, and a more concise version would improve clarity and impact.

Reviewer #2

(Remarks to the Author)

Second Review report for manuscript "Pathogen threat proximity shapes host extracellular vesicle production in pre-infection response"

(Remarks to the Authors)

Upon revision, authors added considerable amount of new data and new figures, which is admirable along with the efforts already put in the original submission. On the contrary, it became clearer through one round of review and revision that authors have a strong attitude to promote their hypothesis, or belief, in a biased way and "cherry-pick" the results to support the hypothesis as described below.

Major comments:

1. Reproducibility of the results

In response to our previous comment 13, authors provided a transparent explanation about methodological differences between Figure 1d and Figure 1e-g, which can reasonably account for the mutant phenotype discrepancies between these figures. However, a closer look at the data within the main text raises additional concerns.

Question 1: How can baseline exopher production be compared between strains?

Authors explained in their rebuttal as follows:

"In Fig. 1e – g, worms, and their OP50 controls, were subjected to an additional 3-hour incubation at 25 °C during the infection window....", "Furthermore, Fig. 1e – g exophers were scored on worms immobilized on agar pads for imaging under

Dr. Pujol's microscope,....", "all other experiments involving muscle exophers presented in this manuscript, including those in Fig. 1d, were conducted on freely moving animals."

Assuming all experiments except Fig. 1 e-g were conducted under the same conditions, as also described in Methods, we note that even in the wild type, baseline data (unexposed control, namely exposed to OP50 alone) are variable between experiments if we compare across all figures (for instance Fig. 5a, b, right panel, Fig. 6a, g, i, j, etc).

This raises a fundamental question: What is the source of this variability, and which parts of the results are reproducible and trustable? If the main source of variability is a day-to-day fluctuation, the next question is whether baseline exopher production can be reproducibly compared between different strains when the experiments are conducted on the same days. If we look at the unexposed controls in Fig. 3, *daf-16*, for example, do not look different from wild type in Fig. 3a/b and *tir-1* in Fig. 3a do not look different from wild type. On the other hand, *akt-1* looks very different in both figures. These results are not consistent with Fig. 1d, where *daf-16* and *tir-1* are significantly different from wild type but *akt-1* is not. Authors describe N=3 for Fig. 3a, and N=3-4 for Fig. 3b, implying not all experiments in Fig. 3 were conducted on the same days. Then how does the baseline data look like if only experiments conducted on the same days are compared between different strains? Does it still fail to reproduce Fig. 1d?

Upon revision, Fig. 8e was replaced from the previous version (Fig. 7e). The data points themselves for all three lines appear to have been altered or added, and the sample size has changed from "n = 90, N = 3 independent experiments" to "n = 90-180, N = 3-6 independent experiments", implying authors conducted additional experiments, and the results of the statistical test changed. This also illustrates the variability between experiments and its effect on the conclusion.

In summary, if inter-strain comparison between strains is difficult even when experiments are conducted on the same days, the only solution will be to make judgements from sufficient numbers of independent experiments. Considering the above, N=3 does not seem enough for these tests.

Question 2: How can fold-change induction in exopher production be compared between strains?

Authors commented in their rebuttal that "despite these differences in absolute numbers, the relative increases in exopher production upon pathogen exposure remain highly reproducible", and that "each experiment includes internal *E. coli* OP50 controls, ensuring that our conclusions are based on fold-change induction rather than raw baseline values".

This may be true, but would authors be able to show numerically that fold-change is more stable than baseline, for example by combining all wild type data?

If this claim holds true, one option would be to avoid comparing the baseline between strains and focus on comparing the fold-change induction. Further, if the fold-change comparison between unexposed/exposed pairs that stem from the same culture turns out most stable, authors can change the statistical tests so as to compare the (same-culture) fold-change values between strains.

Adopting this option will also necessitate the removal of a number of figures and claims. For example, comparisons of baseline values such as Fig. 1d and Fig. 8a, e, i, won't be attempted and deleted from the figures, because, as discussed above, these data are not reliable without sufficient N. Upon revision, authors added Supplementary Figs. 1a, b, 3, 4, 5 and 7, in which they extracted only volatile/nonvolatile-exposed data from Figs. 1, 3, 5, 6 and 8, respectively, and compared them statistically. Based on this, authors made new claims such as involvement of *sri-19* in the volatile signaling. However, if authors admit that baseline is variable and only fold-change is reliable, comparisons between induced values alone are not meaningful. Accordingly, in this case Supplementary Figs. 1a, b, 3, 4, 5 7 and their related statements (such as asserted modulatory roles of *pmk-1*, *akt-1* and AQR/PQR/URX) need to be removed.

As a summary, it is important to evaluate which aspect of the exopher induction by the pathogen-derived stimulus is numerically reproducible. Based on the evaluation, authors need to report only highly stable and reproducible data and base their discussion on them.

2. Functional roles of exopher induction

a. Does the P0-F1 trade off exist?

In response to multiple review comments regarding previous Fig. 8, authors replaced the results with those from a new set of experiments. At 20°C (Fig. 9), the data showed that only volatile metabolites improve F1 survival in an SRI-19-dependent manner, with only a "slight nonsignificant decrease" in P0 survival. In contrast, at 25°C (Supplementary Fig. 8), it is shown that volatile metabolites have no effect on F1 survival but reduce P0 survival. Additionally, non-volatile secretomes reduced F1 survival but had no effect on P0 survival at 25°C. Based on the data, the direct correlation between P0 and F1 fitness, which is a key premise of the paper's overarching narrative, appears to be nonexistent. The data do not consistently show that increased exopher production strengthens offspring pathogen resistance, nor do they consistently show that it weakens it. Instead, the effects are highly dependent on specific conditions (volatile vs. non-volatile cues and temperature), with no clear trade-off between maternal sacrifice and offspring fitness. It is necessary to thoroughly revise the manuscript, for example the language in the abstract such as "significantly boosts offspring fitness... at the cost of maternal survival..." and "to optimize survival across generations", to reflect these results accurately.

b. Is exopher involved in the regulation of pathogen resistance?

The inconsistencies also affect the interpretation of functional roles of exophers. At 20°C (Fig. 9), the data showed that only volatile metabolites improve F1 survival in an SRI-19-dependent manner. This finding is interpreted by the authors as evidence for the functional role of exopher induction in improving F1 survival. However, as SRI-19 protein structure and expression pattern suggested, it is likely to be a sensory receptor or its modifier. Therefore, the F1 survival effect could be mediated through SRI-19-dependent signaling pathway that operates independently of exopher production.

The authors' claims are further undermined by inconsistencies in their experimental design and data interpretation. They state that the inability of exopher upregulation due to a lack of *sri-19* is related to reduced embryo accumulation and prolonged survival upon exposure to PA14 non-volatile secretomes at 25°C (Text line 424-, line 433-). However, the exopher phenotype for *sri-19* mutants under this 25°C condition was never actually observed. The authors instead rely on data from a different temperature (20°C) to support their claim. This conclusion cannot be drawn from the results provided, as an increased temperature not only changes the response of *C. elegans* to *P. aeruginosa* infection but also enhances

exopher production. Therefore, using results from a different temperature for this assertion is logically flawed. In addition, only volatile metabolite improves F1 survival, while sri-19's involvement in the regulation of exopher induction in response to the volatile metabolites, newly asserted based on Supplementary Fig. 7a, is based on inappropriate comparisons as discussed in point 1 above (ignoring baseline instability; exophers are still increased by volatile metabolites, Fig. 8b). Authors presented data showing that unlike wild type, rme-2 P0 becomes more resistant to PA14 pathogen infection. This could be a direct consequence of reduced number of embryos but not necessarily due to the lack of exophers. Overall, evidence on the biological role of exopher is very weak, and overstatement on it should be carefully avoided.

Minor comments:

1. Regarding the response to comment 12, unlike the authors' response, the limitation is not discussed in the Discussion section. On the contrary, they rather emphasize the effect of the non-volatile secretome and volatile metabolites in delivering accurate information of pathogen proximity in Text line 445. Acknowledging the limitations is required for clarity. Authors only showed that volatile metabolites and non-volatile secretome are sensed through different molecular pathways to increase muscle exopher production. Let alone it is unknown whether the contents of these exophers are different, it is unclear whether these different cues cause different consequences through the exophers (for example regarding pathogen tolerance of P0 and F1 as discussed above). The text at line 100 "Our research demonstrates that worms can differentiate the proximity of a pathogen threat based on the type of compound they detect", which remains unchanged from the original manuscript, is clearly an overstatement; the presented results DO NOT show "worms can differentiate the proximity". This and related statements need to be moved to Discussion
2. Response to comment 17. While the revised text on lines 215-219 acknowledges the potential for redundancy or compensation, the subsequent statement on lines 219-223 appears to contradict this, strongly suggesting that the insulin signaling pathway is not involved in exopher production induced by non-volatile secretomes. Also, the authors claim that "non-volatile secretome-driven upregulation selectively requires TGF β , p38 MAPK, and IPR pathways" on Text line 230-231. These conclusions are still not supported by the provided data. At the minimum, "by itself" need to be inserted after "insulin pathway does not" at line 221, and "selectively" at line 231 need to be removed.
3. Response to comment 43 and Text line 542. Addition of the data upon revision on neuronal exopher is informative. However, the manuscript's narrow focus on exophers makes it difficult to support the broader claim regarding a possible hormetic role of EVs in general. The discussion is required to be limited to the specific findings on exophers.
4. Figure 2b. Please describe how the authors identified the region of coelomocytes.
5. Text line 487-495. This paragraph of the Discussion section presents a broad conclusion about the principles of sensory neuroscience in *C. elegans*, referring to concepts such as "combinatorial coding" and the interpretation of "unique activation patterns across multiple neurons". However, these conclusions are not supported by the data presented in the manuscript. The study does not include any experiments that directly measure neuronal activity or the circuit relationships that process the sensory information.
6. Text line 499-502. The authors state that this research addresses the ASK neuron's participation in a pathway related to maternal survival upon pathogen exposure. However, the data demonstrating effect of ASK removal in maternal survival upon pathogen exposure (previous Fig. 8b) does not exist in the current manuscript. Thus, this could not be discussed in the current manuscript.

Reviewer #3

(Remarks to the Author)

Reviewer #4

(Remarks to the Author)

My concerns have been well addressed.

Reviewer #5

(Remarks to the Author)

The revision addresses several of my points, including clarifying figure inconsistencies, adding tissue-specific rescue data, and correcting the writing and references. I appreciate these efforts. Two concerns remain: the omission of key immune regulators (SKN-1, ELT-2, HLH-30) without justification, and the lack of testing beyond the F1 generation. The question of pathogen specificity is improved by the starvation controls but would be stronger with non-pathogenic stressors. Overall, the manuscript is much improved.

Reviewer #6

(Remarks to the Author)

Version 2:

Reviewer comments:

Reviewer #1

(Remarks to the Author)

The authors have satisfactorily addressed all of my concerns. I have no further comments, and I recommend acceptance of the manuscript.

Reviewer #2

(Remarks to the Author)

The authors have addressed many of the points raised in the previous review, particularly regarding the inconsistencies and textual corrections. However, two concerns remain:

1) Response to comment 10. The clarification regarding the use of different transgenes for exopher visualization addresses concerns regarding the baseline variability observed in Figures 5 and 6. However, the concern regarding Figure 3 remains unresolved. Since the mutant strains in Figure 3 (excluding pmk-1 and cul-6) utilize the same transgene background as those in Figure 1d (wac1s1 and wac1s14), they should be directly comparable. The relationship of baseline between wildtypes and the mutants differs clearly between Figure 1d and Figure 3a left panel (e.g., daf-16, tir-1 and akt-1). The provided explanation about different transgenes does not account for this specific inconsistency. Although it is the authors' full responsibility to deal with this issue, as a possible option, modifying "all" to "some" at line 114-115 could more accurately reflect the complexity of the results.

2) Regarding Figure 1a, while the inclusion of representative images helps readers visualize the muscle-derived exophers, the current figure composition is difficult to interpret. Specifically, the fourth and fifth panels appear distinct from the first three. However, the manuscript and figure legend lack a description explaining their significance or how to interpret the data, particularly the heatmap in the fifth panel. Adding a clear explanation of what these specific panels represent and what they are intended to convey would significantly improve reader understanding.

Reviewer #1 (Remarks to the Author):

In this manuscript by Kolodziejska et al., the authors found that both the volatile and non-volatile pathogen metabolites stimulate exophers generation in C. elegans. They dissected the chemosensation of volatile and non-volatile pathogen metabolites, including the chemosensory neurons as well as the potential GPCRs involved, and proposed the sensory signals are integrated by the AIB interneuron via NPR-9. Finally, they proposed the significance of such regulation in enhancing offspring survival against pathogen in the basis of compromising maternal survival. This work addresses an important mechanism on how species fight against pathogen infection in the aspect of population level, and unravelled the potential mechanism of metabolites perception. Besides, the manuscript is clearly written with a clear logical flow. However, there is still critical evidence lacking, that should be completed before publication:

1. Whether the regulation of exopher genesis in muscle is similar as in other tissue, such as neurons?

Response: Thank you for this valuable suggestion. To address it, we directly tested whether pathogen-derived metabolites regulate exopher formation in mechanosensory neurons. As presented in the revised Results (Supplementary Fig. 2c and f), exposure to *P. aeruginosa* volatile metabolites elicited a significant increase in neuronal exopher production, whereas non-volatile secretome had no effect. These findings demonstrate that, unlike muscle exophers, which respond to both cue types, neuronal exophers are selectively driven by volatile signals.

2. How do the metabolites sensed by the chemosensory neurons and integrated by AIB neurons pass to muscle cells? What molecular mechanisms mediate this intercellular communication?

Response: We appreciate the reviewer's interest in the pathway linking chemosensory detection to muscle-cell exopher formation. Although it might be tempting to imagine that pathogen-derived metabolites are directly shuttled from sensory endings to muscle cells, this is unlikely given their chemical diversity and the spatial separation of these tissues. Instead, two other scenarios are more plausible:

1. Classical neurotransmitters or neuropeptides released at synapses downstream of the primary sensory neurons would traverse the interneuron network, ultimately modulating the neuromuscular junction, to trigger exophogenesis in body wall muscle.
2. Long-range neuropeptides secreted by integrative interneurons (such as AIB, AIA or RMG) would diffuse or be transported through the pseudocoelom to reach muscle cells directly and induce exopher production.

Distinguishing between these mechanisms would require extensive follow-up experiments, such as tissue-specific knockouts of individual neuropeptide genes, conditional blockade of synaptic release, or peptide-receptor mapping in muscle, which are beyond the scope of the present study. Accordingly, we instead focused our efforts on further mapping the sensory-to-interneuron circuit: additionally we show that ASJ sensory neuron is not involved (Fig. 5a - c), whereas AIA and RMG interneurons both participate (Fig. 6e - k). We furthermore identified tripeptide isoleucyl-prolyl-proline as one of the PA14 non-volatile metabolite driving exopher upregulation (Fig. 4). Therefore, by clarifying the chemosensory and interneuronal components of this pathway, our work lays the foundation for future studies on the synaptic and peptidergic signalling that convey environmental information to muscle cells.

3. Are there additive effects of volatile and non-volatile metabolites on exopher generation?

Response: Thank you for the suggestion. Our new data (Fig. 2i) demonstrate that combining volatile and non-volatile pathogen metabolites does not produce an additive effect on exopher production.

4. Why are multiple chemosensory neurons required for both volatile and non-volatile metabolites? For example, ASK, ADL, and AWC – do they use the same sets of GPCRs to sense both volatile and non-volatile metabolites?

Response: Multiple chemosensory neurons, such as ASK, ADL, and AWC, are required because each neuron carries a distinct, yet partially overlapping, complement of G-protein-coupled receptors (GPCRs) and downstream signalling machinery. No single neuron can express every receptor needed to detect the full range of volatile and non-volatile metabolites, so by distributing receptors across multiple cells, *C. elegans* maximizes its chemical detection repertoire without overloading any one pathway.

Moreover, chemosensation in the worm relies on combinatorial coding: individual metabolites elicit unique activation patterns across several neurons rather than a one-to-one receptor-to-odour map. Interneurons, like AIA, then interpret these distributed patterns to discriminate among different cues. This architecture not only sharpens sensitivity and specificity for both small, hydrophobic odorants and larger, water-soluble molecules, but also provides redundancy, if one neuron's pathway is impaired, others can still contribute to the overall sensory representation.

Finally, each neuron subtype is tuned to its preferred ligand class through its choice of G α subunits, second-messenger pathways, and ion channels. For example, AWC is optimized for rapid volatile odour detection, while ASK and ADL are more sensitive to soluble metabolites. Although there is some overlap in GPCR expression, differences in receptor affinity, G-protein coupling, and adaptation kinetics ensure that each neuron plays a specialized role. Together, this division of labour and combinatorial integration underlies the worm's ability to detect and distinguish a vast and ever-changing chemical environment including bacterial volatile metabolites and non-volatile secretome.

As EVs are known both for removing unneeded cellular material and for distributing biological material between cells and tissues, it makes sense that the nervous system would employ multiple chemosensory neurons, and their distinct GPCR repertoires, to fine-tune EV-mediated responses. Accordingly, while we have demonstrated that different GPCRs recognize volatile metabolites versus non-volatile secretome (Fig. 8b, c, f, g, j, k), we further hypothesize that discrete sets of GPCRs in specific neurons regulate exopherogenesis in response to these cues. We have now added a section to the Discussion that places these findings into the context of EV-mediated signalling.

5. For the neuron-specific function experiments, the authors should perform rescue experiments to confirm. For example, rescue *SRI-19* in *ADL* neurons.

Response: Thank you for this suggestion. As a general practice with newly generated mutants, we proceed cautiously: for each GPCR we created two independent CRISPR/Cas9 alleles, backcrossed them, and only considered phenotypes that reproduced across both lines. We adopted this strategy in preference to array-based rescue because extrachromosomal arrays frequently introduce overexpression and mosaicism, which can confound interpretation.

Nonetheless, we performed the requested tissue-specific rescue for *sri-19* in ADL and we also chose to rescue *srr-6* in the intestine (to test for a very interesting possibility that this receptor is acting in the gut). For *srr-6*, intestinal rescue restored the non-volatile secretome-dependent response and lowered the elevated baseline, consistent with an intestinal site of action (now shown in Supplementary Fig. 7j). For *sri-19*, ADL-specific rescue using the *srg-234* promoter did not restore the phenotype; instead, it exacerbated the defect (data shown on the graph below). For this

experiment we tested three different injection concentrations (100, 25, and 10 ng/μl). Higher doses produced overt toxicity (enlarged gonad, protruding vulva), while the lowest dose failed to yield a stably transmitting line. Because these outcomes are most consistent with array-driven overexpression artefacts, we chose not to include the *sri-19* rescue data in the manuscript as we believe they are difficult to conclude and interpret.

Importantly, the same phenotypes across two independent alleles for all tested GPCRs (together with the successful *srr-6* intestinal rescue) support our conclusion that SRI-19, SRR-6, and SRI-36 and/or SRI-39 GPCRs regulate exopher upregulation in response to pathogen secretomes.

6. More evidence is required to confirm that AIB interneurons integrate both volatile and non-volatile signals via NPR-9. For instance, the authors could rescue NPR-9 in AIB neurons. Additionally, does activation of AIB neurons phenocopy the effect of volatile and non-volatile metabolites on exopher generation as well as offspring survival on pathogens?

Response: We appreciate the reviewer's suggestion to further validate AIB's role as the integration node for pathogen cues via NPR-9. In addition to our loss-of-function data (AIB ablation and *npr-9* mutants), we employed optogenetic activation of AIB (using ReaChR under an AIB-specific *npr-9* promoter), which recapitulated the expected phenotype: AIB stimulation significantly reduced exopher production in muscle (Fig. 6h), which is consistent with the fact that NPR-9 is a negative regulator of AIB function¹ and with our data for *npr-9* mutant (Fig. 6g). Importantly, multiple studies have shown that *npr-9* is exclusively expressed in AIB^{2, 3}, confirming that the *npr-9* mutant phenotype directly reflects AIB dysfunction rather than off-target effects in other neurons.

We also attempted to assess the impact of AIB activation on offspring survival during pathogen exposure. Although conceptually intriguing, these experiments proved challenging to time and interpret: it remains unclear when AIB-mediated signals would need to reach the germline to influence progeny resilience, and optogenetic protocols may not faithfully mimic the natural temporal dynamics of metabolite sensing. In our hands, AIB stimulation had no detectable effect on progeny survival against *P. aeruginosa*, but we believe this negative result reflects technical limits rather than a lack of biological relevance.

Taken together, two independent loss-of-function approaches and one gain-of-function assay, our data robustly support a model in which AIB, via NPR-9, integrates both volatile and non-volatile pathogen signals to regulate exopher biogenesis.

¹ <https://doi.org/10.1038/cmi.2016.8>

² <https://doi.org/10.1073/pnas.0709492105>

³ <https://doi.org/10.1371/journal.pgen.1006050>

Minor points:

7. *Why do the two sri-19 alleles present different behavior upon PA14-induced exopher generation?*

Response: We have evaluated our data again and can clearly say that both *sri-19* alleles produce identical phenotypes, confirming the robustness of our findings. In each mutant:

- PA14 volatile metabolites exposure elicits a modest but statistically significant increase in exopher production (Fig. 8b) (statistically significantly less pronounced than the robust induction observed in wild-type worms (Supplementary Fig. 7a).
- PA14 non-volatile secretome leads to a slight, non-significant reduction in exopher numbers, in stark contrast to the pronounced increase seen in wild-type animals (Fig. 8c).

The consistent behaviour of two independent *sri-19* alleles, both blunting the volatile-induced response and abolishing the non-volatile secretome-induced upregulation, strongly supports the conclusion that SRI-19 GPCR is essential for the exopher-dependent response to PA14-produced compounds.

8. *Line 264, the citation should be Fig. 8d instead of Fig. 8c.*

Response: Corrected

9. *The PA14-induced exopher generation is even increased in the akt-1 mutants. The authors should at least discuss the potential reason.*

Response: Thank you for highlighting this important point. In the revised version of our manuscript, we have expanded our analysis and discussion on the extent of exopher induction in response to pathogen exposure or pathogen-derived secretomes, particularly in the context of immunity pathway and neuronal mutants. These additions allow for a more nuanced and detailed discussion of the differential responses across genetic backgrounds. Plots illustrating the results of this new analysis are presented in Supplementary Fig. 1a – b, Supplementary Fig. 3 a – b, Supplementary Fig. 4a – b.

10. *The format of citation 44 should be corrected.*

Response: Corrected

Reviewer #2 (Remarks to the Author)

Review report for manuscript "Pathogen threat proximity shapes host extracellular vesicle production in pre-infection response"

< Summary >

Kołodziejska et al. investigated how environmental signals from pathogens regulate the production of muscle-derived exophers in *C. elegans* hermaphrodite. The authors put forth a compelling set of findings on mechanisms of exopherogenesis regulation in response to direct or indirect pathogen exposure:

- Dependency of exopherogenesis on four immune response pathways (p38 MAPK, Insulin signaling, TFP-beta, IPR) by analyzing exopher production of the mutants
- Distinct effect of exposure to pathogen, pathogen supernatant or pathogen volatile to exopherogenesis
- Identification of neuronal cells and molecules required for regulation of exopherogenesis from muscles
- Effect of changed exopherogenesis level upon survival under pathogen infection state

However, multiple conclusions throughout manuscript appear to extend beyond the experimental evidence. The details are addressed in comments.

II. The research report establishes, for the first time, that pathogen exposure can upregulate exopherogenesis and release from muscle cells. Given that this is the inaugural report on pathogen-responsive muscle exophers, it is crucial to document their characteristics. Fluorescence imaging data and detailed descriptions are vital in this context. Although the methods for observing muscle-derived exophers are well-established by previous research, providing image data and comprehensive descriptions of the exophers is essential.

Response: We thank the reviewer for emphasizing the importance of directly visualizing pathogen-responsive muscle exophers. In response, we have added new representative fluorescence micrographs of exophers released after both volatile metabolites (Fig. 2b) and non-volatile secretome (Fig. 2f) exposure. Moreover, each figure legend now includes a detailed description of exophers released by worms body wall muscles due to exposure to pathogen secretomes.

The authors extensively addressed the mechanisms regulating exopherogenesis in response to volatile and non-volatile pathogen substances. While the data provide evidence for the roles of specific molecules and cells in the production of muscle-derived exophers, the connection between the findings is not fully established. Additionally, the authors explore the intriguing phenomenon of muscle-derived exophers influencing lifespan regulation under pathogen-infected conditions. However, the study currently lacks key experiments needed to address several pivotal questions.

To meet the standards of Nature Communications and ensure the claims are fully supported by the data, significant additional experiments would be required. These experiments should validate the key conclusions and provide a stronger link between the proposed mechanisms and the observed phenomena, thereby enhancing the impact and rigor of the study.

< Major comments by sections >

< Introduction >

12. Text line 88 “Our research demonstrates that worms can differentiate the proximity of a pathogen threat based on the type of metabolites they detect.”

The results of this research show that animals can differentiate the pathogen threat via pathogen-derived supernatant or volatiles. However, whether they differentiate the "proximity" of pathogen based on the information has not addressed in this research.

Response: We thank the reviewer for highlighting the distinction between detecting secreted compound identity and inferring pathogen proximity. While our current data demonstrate that *C. elegans* can discriminate between pathogen-derived non-volatile secretome and volatiles, we have not directly tested whether worms use these cues to gauge distance. Nevertheless, existing principles of chemical ecology and our new mechanistic insights suggest a coherent framework: Volatile metabolites are small, low-molecular-weight compounds with high vapour pressures, diffusing rapidly through air or aqueous films to create broad, low-concentration plumes. In contrast, compounds present in non-volatile secretome are larger, more polar molecules that remain close to their source, forming steep, localized concentration peaks. Our findings show that EV production induced by non-volatile secretome depends on canonical immune-response pathways, whereas volatile metabolites-induced EV biogenesis does not engage these immune components. This divergence indicates not only that different GPCRs and downstream signalling machineries are at play, but also that the worm can mount an immune-coupled EV response only when it detects short-range, contact-level cues.

Together, these observations imply that *C. elegans* could infer pathogen proximity: a volatile-only signal would trigger a rapid, immune-independent EV release, perhaps as an exploratory or alerting mechanism, whereas detection of non-volatile secretome would activate immune pathways to produce EVs tailored for such environmental conditions. Although we did not directly assess behavioural or physiological responses across different distances, we have now acknowledged and discussed this limitation in the Discussion section.

<Figure 1 and associated text>

13. The consistency of results between Figure 1d and Figure 1e-g In Fig. 1d, the authors suggest that the deficiency of genes involved in the immune response pathway results in significant changes in the amount of exopher production from muscles. However, the experimental results under the same conditions (i.e., OP50 condition) for *daf-16*, *tir-1*, and *dbl-1* mutants shown in Fig. 1e-g indicate apparently similar amounts of exopher production compared to wild-type animals. The authors should address the clear inconsistencies observed in the results. Clarifying whether these differences arise from the experimental design or other confounding factors is essential for supporting authors' conclusions of increased exopher production in the mutants upon pathogen exposure.

Response: We appreciate the reviewer's careful comparison of Figures 1d and 1e – g. These panels were generated under slightly different conditions in two collaborating laboratories, Dr. Pujol's and Dr. Turek's laboratories, which accounts for the observed baseline discrepancies. Although the same researchers performed these assays in both laboratories, slight methodological differences account for the apparent baseline discrepancies. In Fig. 1e – g, worms, and their OP50 controls, were subjected to an additional 3-hour incubation at 25 °C during the infection window, whereas all data in Fig. 1d were collected at our standard 20 °C. Furthermore, Fig. 1e – g exophers were scored on worms immobilized on agar pads for imaging under Dr. Pujol's microscope, and we have found that immobilization can mechanically perturb fragile exophers and influence exopher counts (possibly in a genotype-dependent manner). In contrast, all other experiments involving muscle exophers presented in this manuscript, including those in Fig. 1d, were conducted on freely moving animals.

Critically, despite these differences in absolute numbers, the relative increases in exopher production upon pathogen exposure remain highly reproducible across both laboratories and two distinct pathogens (*P. aeruginosa* and *S. marcescens*). Each experiment includes internal *E. coli* OP50 controls, ensuring that our conclusions are based on fold-change induction rather than raw baseline values. Thus, while procedural variations affect baseline exopher counts, they do not compromise the robust, pathogen-specific upregulation observed in immune-pathway mutants.

To clarify the baseline differences between Figures 1d and 1e–g, we've added a note to the Figure 1 legend:

“Baseline differences between panels d and e – g arise from methodological variations which are detailed in the Materials and Methods section “Scoring of Exophers and Fluorescence Microscopy”

In the Materials and Methods section, we explicitly describe the differing incubation temperatures, durations, and worm immobilization steps used to collect each dataset.

14. Figure 1i and text line 127 “This indicates that the P. aeruginosa-dependent increase in exopher production operates independently of the previously described embryo-maternal signalling, suggesting the involvement of a novel regulatory pathway.”

It is difficult to assert that the exopher production in response to PA14 is independent of embryo-maternal signaling only based on the observation that the increase of exopher production was not accompanied by the increase of egg retention. To substantiate the claim, it is crucial that experiments be conducted to determine whether depleting embryos affects exopher production. Such studies are necessary to definitively conclude the independence of these processes.

Response: We thank the reviewer for this important point. In our original text, “embryo-maternal signaling” referred to the established correlation between exopher upregulation and increased embryo accumulation which we did not observe following PA14 exposure. However, your comment prompted us to reconsider whether exopher induction might be entirely independent of embryo presence. To test this directly, we treated animals with 5-fluoro-2'-deoxyuridine (FuDR) to block embryo production, then exposed them to *P. aeruginosa* and *S. marcescens*. In FuDR-sterilized worms, exopher production was completely abolished under both conditions, demonstrating that functional fertility is indeed required for the pathogen-induced exopher response. These new data have been added to Supplementary Fig. 1c and the Results section now clarifies that pathogen-triggered exophogenesis depends on fertility, even though it not necessarily require increased embryo accumulation.

< Figure 2 and associated text >

15. Figure 1i demonstrated that Pseudomonas aeruginosa does not lead to an increase in embryo accumulation in utero, contrasting sharply with the effects of pathogen volatiles as illustrated in Figure 2c. These findings raise the possibility that non-volatile pathogenic stimuli may suppress the embryo accumulation typically induced by exposure to pathogen volatiles. The authors are encouraged to provide a detailed explanation for these divergent outcomes.

Response: Thank you for highlighting this divergence. We have expanded the Results to address it. The new text now reads:

“The fact that *P. aeruginosa* non-volatile secretome fails to increase embryo accumulation *in utero*, in stark contrast to the retention induced by volatile metabolites, suggests that non-volatile pathogen cues may actively suppress the embryo-accumulation response typically triggered by volatiles. This

opposing outcome points to distinct downstream signalling branches that modulate reproductive physiology in response to different classes of pathogen-derived compounds.”

16. *In this manuscript, the terms 'bacterial supernatant' and 'bacterial non-volatile metabolites' appear to be used interchangeably. However, as bacterial supernatant also contains a variety of substances including extracellular vesicles, which are not metabolites. Thus, it is recommended for the authors to define the range of the terms or discuss the limitation of the experiments. A similar level of explanation is also preferred for the bacterial volatiles used in this research.*

Response: We agree that “bacterial supernatant” and “non-volatile metabolites” can be misleading, since the supernatant contains a complex mixture of molecules (small metabolites, peptides, proteins, and extracellular vesicles). Accordingly, we have standardized our terminology throughout the manuscript to refer to the volatile metabolites (all airborne compounds emitted by the pathogen) and the non-volatile secretome (the full complement of water-soluble factors present in clarified culture supernatant including small metabolites, peptides, proteins, and extracellular vesicles). We have added a brief definition of each term in the Results.

< Figure 3 and associated text >

17. *Figure 3a, c and text line 152 “Our results indicate that none of the tested pathways—including insulin, TGFβ, p38 MAPK, and IPR—significantly contribute to the volatile metabolite-dependent increase in EV production (Fig. 3a and c).”.*

It might seem as though the authors are completely excluding the possibility that the pathways contribute to the pathogen volatile-dependent increase in EV production. Considering potential for other scenarios, such as redundant contributions from these pathways, a more nuanced tone that leaves some room would be more appropriate.

Response: We thank the reviewer for this insightful suggestion. We agree that our original phrasing may have implied too strong an exclusion of these pathways. We have revised the sentence to adopt a more cautious tone, acknowledging potential redundancy or compensation. The updated text now reads:

“On one hand, our data show that insulin, TGFβ, p38 MAPK, and IPR pathways are not individually required for the robust increase in exopher production triggered by volatile secretomes as each mutant mirrored wild-type induction (Fig. 3a and c), although we cannot exclude the possibility of redundant or compensatory interactions.”

This adjustment better reflects the limitations of our data and leaves open the possibility that these pathways may act in concert or through overlapping mechanisms.

18. *Figure 3b and c. The requirement of several immune response pathways for this EV production raises interesting questions. Are these pathways working together to regulate exophergenesis? Discussing the interrelationships among these pathways within the mechanism regulating exophergenesis could yield valuable insights.*

Response: Thank you for this insightful suggestion. We have now added in the Discussion that partial overlap of immune pathways suggests coordinated or compensatory interactions that fine-tune exophergenesis according to the nature and intensity of the pathogen cue.

19. Text line 161 “... bypassing these pathways entirely for airborne signals.” Similar with Major comment 8, excluding all the possibilities entirely is too definitive.

Response: We have revised the sentence to adopt a more cautious tone.

< Figure 4 and associated text >

20. The authors have not detailed the information for genetic ablation of each neuron in Methods section. Although the information is included in Supplementary Table 2 “*Caenorhabditis elegans* strains”, a brief description of the type of ablation within Methods and/or Results with reference to Table 2 would greatly enhance reader accessibility.

Response: Thank you for this suggestion. In the revised manuscript, we have cited the original studies that generated each sensory-neuron ablation line directly in the Results section. We believe that by combining these citations with the detailed strain list presented in Supplementary Table 2, we ensure that readers have both immediate context in the Results and comprehensive information in the supplement.

21. Figures 4a-c demonstrate that the ablation of ADL or ASK neurons not only abolishes the upregulation of exopher production in response to pathogen volatiles or supernatant, but also the ablation itself considerably reduces exopher production already. The authors are not addressing this aspect.

Response: Thank you for highlighting this. We have now addressed it in the manuscript and cited our previous work where worms with sensory neurons ablations were investigated in the context of exopher production. The revised text now reads: “Notably, ablation of ADL or ASK also reduces baseline exopher levels (Fig. 5a and c), consistent with our previous work showing that these neurons contribute to exophogenesis regulation at the basal level.”

22. Figure 4a-c identify neurons required for upregulation of exopher production via stimulation from pathogen volatile or supernatant. The results are indicating that lack of ADL or ASK neurons abolishes the upregulation. However, the data also reveal that ablation of each neuron itself results in a considerable reduction in exopher production, implying that the exopher production is constantly downregulated already by the neuronal ablation. Additionally, ASK-ablated strains show prolonged survival irrespective of pathogen volatile exposure experience, as depicted in Figure 8b. These findings suggest an underlying complexity in the role of these neurons towards exophogenesis regulation that likely precedes their involvement in pathogen response. Given these complexities, it would be beneficial for the authors to further explore and discuss the implications of neuronal ablation on general physiological functions. Clarifying these intricate relationships will enhance the manuscript's depth and help substantiate the authors' conclusions.

Response: We thank the reviewer for highlighting the broader implications of sensory-neuron ablation. As reported in Szczepańska et al. (2024), ablating ASK, ADL, or AWC alters basal exopher production even in the absence of pathogen cues, a finding we have cited directly in the Results. Rather than reiterating previous findings, we have expanded the Discussion to include a dedicated section on how ASK and ADL sensory inputs converge on the AIA, RMG, and AIB interneuron network to control both basal and pathogen-triggered exopher production. As well as discussing a dual nature of these sensory neurons in regulation of exophogenesis by interpreting both reproductive cues and pathogen-derived danger signals. Importantly, all of our conclusions about metabolite/secretome-specific regulation are drawn from comparisons to internal controls consisting of neuron-ablated animals that were not exposed to volatile metabolites or non-volatile pathogen

secretomes, ensuring that the observed effects reflect true stimulus-dependent signalling rather than general perturbations of exopher machinery.

23. In Figure 4, the authors broadly investigated the contribution of multiple sensory neurons to the regulation of exopher production from muscles. It is noteworthy that these neurons are part of the "Hub-and-Spoke" system, with RMG functioning as the central hub (Ref. 35 and Macosko EZ et al., 2009). This raises the possibility that RMG may play a central role in regulating exophergenesis, potentially through its interactions with the sensory neurons studied in this research. Investigating exopher production in neuron-ablated animals including RMG will enhance the understanding of circuit mechanisms underlying the nervous system's regulation of exophergenesis from muscles to substantiate novelty of this study.

Response: Thank you for this insightful suggestion. We have now analysed exopher production in RMG-ablated animals and find that RMG indeed plays a key role in this circuit. RMG-deficient worms exhibit a reduced baseline level of muscle exophers, and unlike wild-type controls, they fail to upregulate exopher production in response to both *P. aeruginosa* volatile metabolites and non-volatile secretome. These data are now presented in Fig. 6a – c, k – m underscoring RMG's function as a hub neuron required for integrating chemosensory inputs and driving exophergenesis in muscle.

< Figure 7 and associated text >

24. The arrangement of panels in Figure 7 is somewhat challenging to interpret. It may be more effective to reorganize the figure by grouping panels according to their respective conditions. Additionally, while Figures 7, 3, and 4 share a similar experimental design, Figure 7 includes statistical comparisons under basal conditions, which are not addressed in Figures 3 and 4. Since the basal phenotypes of the mutant animals used could significantly impact the interpretation of their pathogen substance-responsive phenotypes, providing information on basal phenotypes is strongly recommended.

Response: We have added vertical separators in Fig. 8 (previously Fig. 7) to clearly delineate data for each GPCR mutant: rows now group treatments by condition (volatile vs. non-volatile), and columns group all data for a given receptor, making interpretation more intuitive. Regarding basal phenotypes for Fig. 3, they are presented in Fig. 1d, and for Fig. 5 (previously Fig. 4) the baseline effects of neuronal ablations were thoroughly characterized in Szczepańska et al. (2024), which we cite in our manuscript. We have also added a brief note in the Results to remind readers that ADL and ASK ablations produce a consistent reduction in basal exopher counts while AQR/PQR/URX ablation elevates them, matching our previous findings.

25. Text line 230 "Notably, the basal level of exophergenesis in GPCR mutant worms under standard conditions remained unchanged (Fig. 7a, e, i). These findings collectively indicate that these receptors are critical for pathogen-induced exopher production, but not under non-stressed conditions.". It is stated that the basal level of exophergenesis in GPCR mutant animals remains stable under standard conditions; however, the results from *sri-36/39* (*wwa57*), *ssr-6* (*wwa48*), and *ssr-6* (*wwa49*) show significant variations. The authors need to reconcile these observations with their initial claims. Furthermore, as the mutants itself, without pathogen stimuli, showed significant change of exophergenesis, it is misleading to claim "that these receptors are critical for pathogen-induced exopher production, but not under non-stressed conditions.". The authors must clearly delineate the observed phenotypes and articulate their contributions in light of the results obtained.

Response: We thank the reviewer for this important clarification. We have revised the text to accurately reflect the observed phenotypes in these GPCR mutants.

< Figure 8 and associated text >

26. The manuscript currently lacks a detailed description of the methods used for the parental survival assay. This information should be added to methods and described briefly in figure legends.

Response: We have improved method description.

27. Figure 8a presents an illustration that is challenging to interpret, despite a detailed legend that explains the experimental scheme. For instance, the notation 'exophers scored @AD2 F1 embryos collected' explains the procedure for Figures 8e and f. However, at first glance, it appears to pertain to the experiments shown in Figures 8b-d. To enhance clarity and ensure that readers can easily follow the experimental design, the illustration requires improvement.

Response: We have revised the schematic to clearly delineate each experimental branch making the workflow immediately understandable.

28. Figure 8b. The effect of ASK neuron ablation on the prolonged survival of *C. elegans* under *P. aeruginosa*-infected conditions was previously reported by Venkatesh et al. (*Life Science Alliance*, 2023). The authors should reference this study to provide appropriate context and acknowledge prior findings.

Response: We have added a citation to Venkatesh et al. (*Life Science Alliance*, 2023) in the Discussion to acknowledge their prior report on ASK ablation and extended survival under *P. aeruginosa*.

29. Figure 8b illustrates that exposure to pathogen volatiles, which increases exopher production from muscles, decreases the survival rate under *P. aeruginosa* infection. The manuscript notes that prolonged survival due to ASK neuron ablation, which increases exopher production (Fig. 4a-c), supports the inverse correlation between exopher production and survival under infection. These findings raise critical questions. Yet, several aspects need to be addressed to fully support the conclusions claimed by the authors. These include:

a. It is evident that ablation of the ASK neuron downregulates exophergenesis from muscles (Fig. 4a-c). This raise intriguing questions, could the survival-extending effect observed with ASK ablation be attributed to the altered exophergenesis? What will be the roles of the ASK neuron in regulating both exophergenesis and survival? While the *rme-2* RNAi experiment in Figure 4c supports the connection between exopher production and survival, the degree of survival extension observed with ASK ablation appears to differ from that of *rme-2* RNAi, implying a contradiction which raises a concern that ASK's pathogen susceptibility effect may be largely through pathways other than exophers. Addressing this crucial question will require additional experiments. For example, testing *rme-2* RNAi in ASK-ablated animals or testing pre-exposure of pathogen under *rme-2* RNAi condition could provide valuable insights for understanding the complex biology.

Response: Thank you for these thoughtful suggestions. In the revised manuscript we center our mechanistic analysis on the SRI-19–dependent link between parental volatile sensing, exopher induction, and offspring outcomes, while the role of ASK in regulation of exopher formation, survival and longevity is discussed in details in the Discussion.

30. b. Previous research by Alcedo and Kenyon (*Neuron*, 2004) suggested that ASK has a longevity-promoting effect based on combinational cell ablation experiments conducted under pathogen-free conditions. Interestingly, under pathogen-present conditions, ASK exhibits longevity-reducing effects. Given these contrasting observations, it would be valuable for the authors to explore and discuss how ASK and ASK-

dependently regulated exophers contribute to longevity mechanisms in these opposing ways, depending on the presence or absence of pathogens.

Response: We thank the reviewer for drawing attention to the dual role of ASK in longevity. We have now expanded the Discussion to reconcile these contrasting observations

31. *Figure 8c indicates that the knock-down of the rme-2 gene, known to decrease exopher production, also extends animal survival under infection, supporting the notion that reduced exopher production might enhance survival. Intriguingly, previous research has shown that the exophogenesis event from touch neuron extends lifespan (text line 332 and Ref. 16), highlighting a potential protective role of exophers in non-pathogenic contexts. This raises a crucial question, will enhancing exophogenesis from muscles through factors other than pathogen exposure, such as the knock-down of vit-1 or emb-27 genes, impact the survival of the animals under the condition tested in Figure 8c? This experiment can significantly strengthen understanding for the roles of muscle-derived exophers in survival under pathogenic conditions.*

Response: We appreciate this insightful suggestion. While testing additional genetic manipulations, such as *vit-1* or *emb-27* knockdowns, to enhance muscle exophoresis could indeed be informative, we have focused on elucidating a clear, receptor-mediated pathway via SRI-19 that links pathogen sensing to exopher production and survival outcomes. By centering on this defined mechanism, we provide a cohesive framework that can be further expanded in future work to explore other methods of modulating exophogenesis.

32. *Figure 8c-d. The manuscript lacks clarity on whether the animals are exposed to pathogen volatiles prior to the survival assay. This critical information is not described in the legends, nor is it detailed in the Methods section. The information should be addressed in Methods and depicted briefly in legends.*

Response: We have revised the Materials and Methods to clearly specify the timing and conditions of pathogen-volatile pre-exposure prior to the survival assays.

33. *Figure 8d and text line "Additionally, sri-19 mutants, which do not show increased exopher production upon contact with P. aeruginosa non-volatile metabolites, survived significantly longer during infection (Fig. 8c) which demonstrates that on mechanistic level survival is interconnected with the regulatory elements for EV production."*

The authors examined exopher production in sri-19 mutants in response to pathogen supernatant or volatile as shown in Figure 7b and c. Assuming that the protocol in Figure 8d adheres to that outlined in Figure 8a, the primary factor influencing the survival of the mutant animals under pathogen infection would be the altered level of EV production induced by prior exposure to pathogen volatiles. However, Figure 7b indicates that EV production can be significantly upregulated even in the absence of the sri-19 gene. These results raise a few questions:

a. Does pathogen infection significantly enhance EV production in sri-19 mutants?

Response: As suggested to assess whether SRI-19 is required for infection-driven exophogenesis, we exposed *sri-19* mutants to live *P. aeruginosa*. As shown in Supplementary Figure 7g, *sri-19* mutants did not exhibit the pathogen-induced increase in exopher numbers observed in wild-type worms, demonstrating that SRI-19 is essential for exopher upregulation during live bacterial infection.

34. b. *Which will dominantly impact on the survival of sri-19 mutants, pathogen supernatant or pathogen volatile?*

Answering these two questions could provide a fundamental basis to demonstrate that survival is mechanistically interconnected with the regulatory elements of EV production.

Response: In *sri-19* mutants, neither volatile nor non-volatile pre-exposure altered offspring survival, an outcome that sharply contrasts with wild-type animals, in which volatile cues significantly enhance progeny survival on *P. aeruginosa*. This result confirms that the survival benefit conferred by bacterial volatiles is dependent on SRI-19 receptor.

35. In the study, survival assays utilized different initiation stages for P0 and F1 generation animals. P0 animals are exposed to *P. aeruginosa* starting from the day 2 adult stage, whereas F1 animals begin exposure at the L4 stage. It is crucial for the authors to justify the choice of different stages for these experiments (why not adult day 2 for F1 animals) or clarify whether the pathogen's effects on the lifespan of *C. elegans* are consistent across these developmental stages.

Response: Thank you for raising this important point. In our design, P0 animals must reach day-2 adulthood (AD2) before infection because they are pre-exposed to *P. aeruginosa* volatiles or secretomes from L4 to AD2; infecting them earlier would truncate this conditioning window. For F1s, additional experiments indicate that the infection stage strongly influences the detectability of intergenerational effects. When we infected F1s as embryos (3-fold stage), we observed a robust benefit after parental volatile pre-exposure, but not after non-volatile pre-exposure. These data, presented in the current version of the manuscript, align with recent reports that parental (P0) perception of *Pseudomonas vranovensis*-derived cyanide confers protection to offspring against subsequent pathogen challenges¹. In our earlier version of the manuscript, L4-infection assays show increased survival detectable only in the subset of progeny from mothers that produced exophers above a defined threshold, indicating reduced penetrance at later stages. By contrast, additional experiments that we have performed show that infecting F1s at AD2 had no effect on their survival comparing to unexposed controls (data not shown).

Taken together, these results suggest that the parental effect is stage-dependent, being strongest during early development, an outcome that is ecologically plausible, as early life is likely the most vulnerable period, critical to invest in, in order to support proper development to adulthood and survival of the population. In current version of the manuscript we also note that temperature modulates these phenotypes, showing further how changes in the environment modulate intergenerational survival. To remain within space limits and present the most robust, reproducible signal, we therefore focus on embryo-stage infections in the revised manuscript, which we believe provides the clearest and most interpretable test of the phenomenon.

¹ <https://doi.org/10.1016/j.cell.2024.11.026>

36. Similar to Figure 8 discussed above, Fig. 8e and f lack sufficient labels to depict different conditions. Figure legend lacks information and it should describe what are shown in different subpanels. Additionally, an explanation of the error bars for Fig. 8e and f should be provided to ensure clarity.

Response: We have enhanced the visualization of these experiments in Fig. 9 (formerly Fig. 8) by adding clear panel labels and grouped color coding to indicate each condition.

37. The authors have sorted the P0 population into two categories based on criteria outlined on text line 280. However, the rationale for selecting these specific criteria is not provided. Detailed explanations are necessary for:

a. Why the numbers of exophers (7 and 10) were chosen as criteria for categorizing the biological replicates into high/low exopher groups. Although Supplementary Figure 2 and the associated text provide some basis for choosing these criteria, the process for selecting these specific numbers is not described. Furthermore, the authors should explain why the criterion changed from the median values 6 and 10 in Fig. S2 to 7 and 10 in Fig. 8e and f. This is a critical problem that could be a potential violation of neutrality in the statistical tests.

Response: Thank you for raising this. The thresholds were not changed mid-study, the mention of “6” in Supplementary Fig. S2 was a labeling error, it should have been “7”. However, as noted earlier,

we do not present the L4-infection datasets here, because embryo-stage infection provided a more robust and interpretable readout than L4 infection.

38. b. *How the populations were handled for breeding and observation of exophers? Providing experimental details is crucial because this is a set of experiments on which authors' major claim "volatile-induced exopher production enhances offspring survival against pathogen" is based.*

Response: We have expanded the Materials and Methods to include detailed protocols for each assay, ensuring full transparency and reproducibility of our experimental setups.

39. c. *We assume that Fig 8e shows the results of high-exopher group and Fig 8f show the low-exopher group. Then what are shown on the left end panels of Fig. 8e and f? Which threshold was used for which test? Why are there data that are both lower and higher than the thresholds?*

Response: Thank you for the careful read, this is a misunderstanding of how the grouping was done. The 7/10 exopher thresholds were applied at the level of the whole P0 cohort in a given experiment, not to individual animals. In other words, we did not filter out individual animals within those cohorts.

40. *The left panel of Figures 8e and f represents exopher production across all examined individuals from multiple replicated groups. However, to clearly demonstrate that each P0 group used to generate categorized F1 groups meets the established criteria, it is essential to post the specific amounts of exopher production observed in each individual population. Providing this data will enhance transparency and allow for a more precise assessment of the experimental results.*

Response: As we do not present the L4-infection datasets here, because embryo-stage infection provided a more robust and interpretable readout than L4 infection, we did not include P0 graph for this data set.

41. *The authors propose that the increased exopher production due to exposure to pathogen volatile decreases the survival of P0 animals under pathogen exposure, while the survival of the F1 generation under the same conditions is enhanced at the cost of the parent generation's survival. Figures 8e and 8f aim to demonstrate this hypothesis by comparing WT and WT+PA14 populations of P0 and F1 animals via survival assays. However, when analyzing the results within populations — for example, comparing WT animals above the criterion (Fig. 8e, those producing more exophers) with WT animals below the criterion (Fig. 8f, those producing fewer exophers) — the results reveal conflicting phenotypes. Specifically, P0 animals above the criterion (Fig. 8e, middle panel) exhibit longer survival than those below the criterion (Fig. 8f, middle panel), contradicting the notion that increased exophers reduce the survival of P0 animals under pathogen conditions proposed by the authors. Conversely, F1 animals derived from P0 animals above the criterion (Fig. 8e, right panel) show shorter survival than those derived from below the criterion (Fig. 8f, right panel), which also conflicts with the authors' interpretation. The trend is consistent across WT and WT+PA14 populations in Figures 8e and 8f.*

These findings not only conflict with the authors' major novel claim in the research, but also contradict with the results presented in Figures 8b-d. The authors must address this significant discrepancy transparently and provide a detailed explanation of their rationale for the data presentation and interpretation. Furthermore, a statistical analysis of the data within the same population, specifically comparing subgroups divided by the criteria as mentioned earlier, is necessary. Additionally, it is crucial to demonstrate that the survival-decreasing effects of pathogen volatile-responsive exophers in P0 animals under pathogen exposure are consistently observed within the experimental framework of offspring survival. Clarifying these points is essential to resolve the discrepancies and ensure the validity of the conclusions.

Response: Thank you for this detailed critique. Our claim was based on within-experiment comparisons: when P0 and F1 cohorts from the same run were analyzed, higher maternal exopher output associates with reduced P0 survival and increased F1 survival. In our experience, cross-day comparisons can be difficult to interpret as exopher production is sensitive to even small variation in the experimental conditions. To address this, we always rely on internal controls and performed the assays with matched cohorts.

However, as mentioned earlier, we have repeated this assay with infected F1s as embryos (3-fold stage), and we observed a robust benefit after parental volatile pre-exposure, but not after non-volatile pre-exposure. These data, presented in the current version of the manuscript, align with recent reports that parental (P0) perception of *Pseudomonas vranovensis*-derived cyanide confers protection to offspring against subsequent pathogen challenges¹. Also, to avoid overinterpretation, we have revised the text to describe the effect as follows: P0 volatile pre-exposure decreases P0 survival under harsher conditions (25 °C), whereas F1 show improved survival at 20°C and accelerated development at 25 °C. We further show that this depends on the SRI-19 GPCR, indicating that the regulatory mechanism for exopher production is shared with the mechanism underlying increased F1 survival at 20 °C after exposing P0 to *P. aeruginosa* volatiles.

¹ <https://doi.org/10.1016/j.cell.2024.11.026>

< Discussion >

42. Text line 316 “This distinction implies that exophers may constitute distinct subtypes of EVs with unique molecular compositions and functions.”

Throughout the results, the authors have effectively demonstrated that the contribution of immune pathways to exopher production varies depending on pathogen-derived stimuli. However, the research has not explored whether the EVs produced differ between various conditions or treatments. Instead, the survival assay results for ASK-ablated animals from Figure 8b, which ablation constitutively reduces EV production regardless of the type of pathogen-derived stimulus (Figure 4a-c), could hint at a functional uniformity of the produced EVs. The potential heterogeneity might be suggested by assessing the distinctiveness of the generated exophers.

Response: Thank you for the comment. In the current version of the manuscript we have softened conclusions and wrote that instead of implying our results raise the possibility. We agree that we do not have data proving this but as the discussion is an appropriate place to discuss hypotheses that may arise from the data we decided to keep it in the manuscript.

43. Text line 331-337 “Previous studies have shown that increased exopher release can benefit the organism by removing protein aggregates, extending lifespan, or promoting faster offspring development. However, when exopher release is elevated in response to pathogen exposure, we observe an opposite effect, leading to decreased survival upon infection. This suggests that exophers, or possibly EV in general, may function under a hormetic response: while a moderate increase in EV production appears beneficial, an excessive increase may overwhelm the organism, negating any positive effects.”

The authors focus their investigation on exophers, a specific subtype within the diverse family of extracellular vesicles. Considering the well-established diversity of EVs in origin, characteristics, destination, and function, it is premature to extend the findings of this study to all EVs. Furthermore, it is important to note that Reference 16 suggests distinct functions for neuron-derived exophers compared to the muscle-derived exophers examined here. This observation raises a fascinating question; why do neuron-derived exophers and muscle-derived

exophers appear to have opposite effects on survival? This intriguing contrast merits further discussion and exploration within the manuscript, rather than limiting the scope of potential implications.

Response: Thank you for this insightful suggestion. We have expanded the Discussion to include a comparative analysis of neuron-derived versus muscle-derived exophers, incorporating our new data on neuronal exopher production following exposure to *P. aeruginosa* volatiles and non-volatile secretomes.

44. Text line 345 “In contrast, our study involved prolonged exposure to a mixture of volatile metabolites from pathogens, which may trigger different physiological responses.”

The authors emphasize the potential importance of i) the duration of exposure and ii) the specificity of volatiles in mediating the physiological responses of *C. elegans* to *P. aeruginosa*. Indeed, Ref. 31 demonstrates that short-term exposure (2 hours) to 1-undecene can improve survival under pathogen exposure, which contrasts with the effects indicated in the current research. However, to suggest the possible influence of the factors on the variation in physiological responses, the authors should provide additional supporting evidence. For example, they could compare whether varying the duration of exposure to 1-undecene leads to different effects or not. Additionally, discussing which specific volatile compounds from pathogens could underlie the observed physiological responses in this research would significantly enhance the depth and scope of the discussion.

Response: Thank you for this thoughtful suggestion. Our intent in that paragraph was not to ascribe specific causes, but to contextualize our results against prior work and highlight two plausible variables, exposure duration and volatile identity, that may account for divergent outcomes. Given space constraints (we are already ~1,000 words over the 5,000-word limit), we have kept this section concise, but we now explicitly note in the Discussion that both exposure timing and the chemical composition of the volatile blend (e.g., 1-undecene among other *P. aeruginosa* volatiles) are likely determinants of the physiological response and represent important directions for future work.

< Minor comments >

45. Text line 116 “However, when immune response mutants were exposed to pathogens, there here was a consistent increase in exopher production across all tested mutants, as observed in wild-type worms (Fig. 1e-g).”

It could be read as all of the tested mutants are showing consistent increase in exopher production in response to pathogen exposure. However, in the following sentence, the authors mention that the *cul-6* mutant is an exception. Therefore, the phrase 'across all tested mutants' may not accurately reflect the data and could be revised to account for the exception observed in the *cul-6* mutant.

Response: We have corrected this phrasing. It now reads:

“However, when immune response mutants were exposed to pathogens, most displayed a consistent statistically significant increase in exopher production similar to wild-type worms (Fig. 1e-g). The sole exception was the *cul-6* mutant with a compromised IPR pathway, which showed an increase in exopher production upon exposure to *P. aeruginosa* and a decrease in exopher production upon exposure to *S. marcescens* when compared to the wild-type control (Fig. 1h).”

46. Text line 121 “Therefore, the pathogen-induced EV modulation does not rely on canonical immune response pathways.”

Given that the authors have not fully tested for redundancy in the immune response pathways, the claim that pathogen-induced EV modulation 'does not rely on canonical immune response pathways' could be

reconsidered. A more cautious interpretation may be warranted until potential compensatory mechanisms are explored.

Response: We agree that our original wording was too definitive given the possibility of redundant or compensatory immune mechanisms. We have therefore softened the language to reflect this nuance. The revised sentence now reads: “Together, these findings indicate that pathogen-triggered exopher upregulation does not rely on any single canonical immune pathway, although we cannot rule out redundant or compensatory contributions...”

47. Text line 124. egg retention

Response: Corrected

48. Fig. 2a-c. The conditioning duration has changed from 3 hours for pathogen exposure (Fig. 1) to 48 hours for pathogen volatile or supernatant exposure (Fig. 2; starting from L2 as described in Methods). The authors should at least comment the ground for choosing 48 hours as pathogen volatile/supernatant conditioning to trigger exopher production. Is 3 hours too short for the volatile/supernatant to exert effect?

Response: We chose a 48-hour exposure for volatile and supernatant assays to ensure robust and reproducible exopher induction under our standard sample sizes. In pilot experiments (data not shown), a 3-hour supernatant exposure was sufficient to produce a statistically significant increase in exopher counts, whereas a 3-hour volatile exposure yielded an upward trend that did not reach significance with our typical n. Extending the volatile exposure to 48 hours reliably amplified the effect size and improved assay consistency across replicates.

49. Figure 2a. The illustration presents an ambiguity; the stacked two plates in fourth row appear to be oriented with both top sides facing each other, which could potentially obstruct the passage of volatiles. If this is the case, the arrangement might impede the intended experimental conditions and affect the reliability of the results.

Response: Thank you for noting this. Although the plates in the fourth row are shown “stacked,” neither contains a lid during incubation, each agar surface is exposed, and the upper plate effectively acts as a lid for the lower one. Volatile metabolites diffuse freely between the two open agar surfaces. This open-stacked configuration has been used in *C. elegans* volatile exposure assays by others^{1,2} and reliably permits gas-phase transfer without liquid or physical contact.

¹ <https://doi.org/10.1038/s43587-023-00467-1>

² <https://doi.org/10.1038/s41598-022-26554-8>

50. The authors should standardize the terminology used throughout the manuscript to maintain consistency. Specifically, the terms 'embryo accumulation' and 'egg(s) retention' are considered to refer to similar phenomena but are used interchangeably. If the two terms are indicating different phenomena, describing the difference will help readers understanding better. Also, a unified terminology would enhance clarity and improve the manuscript's readability.

Response: We thank the reviewer for this helpful suggestion. We have standardized our terminology throughout the manuscript, using “embryo accumulation” exclusively to describe the phenomenon previously also referred to as “egg retention.” This change ensures consistency and improves clarity.

51. Text line 145. EV in response or exopher in response

Response: Corrected

52. 8. Figure 2f and text line 148 “Notably, this increase in exopher production do not depend on increased egg retention in the worm's uterus (Fig. 2f).”

Deciding that they “do not depend on” might be hasty conclusion. Although it is true that the increase in exopher production occurred without increased egg retention under exposure to pathogen supernatant, this does not exclude a possibility that egg retention could affect the exopher production. Thus, “do not require” might be more accurate for this case.

Response: We have revised the sentence according to the reviewer's suggestion.

53. Figure 4a and text line 174 “While ASI, ASH, and AWB neurons are crucial for the supernatant-induced increase they do not play a role in volatile metabolite dependent increase in EV production (Fig. 4a-c).” In figure 4a, with ablation of ASH or AWB, while the amount of produced exophers is still significantly increased by pathogen exposure, the amount of change seems decreased compared to intact animals. To clarify this point, the magnitude of the pathogen-induced increase in EV production could be statistically compared between wild-type and ablated animals. This analysis could involve employing a two-way ANOVA approach, or its non-parametric variant, the Aligned Rank Transform (ART) ANOVA.

Response: In response to your and Reviewer 4 suggestions, we have performed direct comparisons of the pathogen-induced exopher increase magnitudes for all neuronal ablations, immune-pathway mutants, and GPCR mutants. These effect-size analyses are now presented in Supplementary Fig. 1, 3, 4 and 5 to maintain clarity in the main figures. Moreover, in the revised manuscript, we now explicitly explain which comparisons are supported by robust effect sizes and reproducible trends, and we temper our conclusions where changes, though statistically significant, are modest. These revisions ensure that readers can readily distinguish between findings of clear physiological consequence and those that, while reproducible, may warrant further investigation to establish their *in vivo* importance.

54. In this research, ADL neuron is functionally ablated by *hlh-4(tm604)* III mutation. According to Masoudi et al., 2018, in *PLoS Biology*, this mutation disrupts the ‘identity features that define functional features of the ADL neuron’, without physically destroying the neuron. Since this approach differs from the methods used to ablate other neurons in the research, it is crucial to clearly delineate this discrepancy in the *Materials and Methods* section.

Response: We appreciate the reviewer’s attention to this important distinction. We have now expanded the description of each neuronal ablation strain in Supplementary Table 2 to specify the ablation method used, highlighting that *hlh-4(tm604)* disrupts ADL identity without physical cell removal, whereas other strains employ genetic caspase-mediated cell destruction.

55. Text line 191 “... receptor abolishes the ...”. Figure 5c shows that while the increase in pathogen supernatant-dependent EV production in AIB(-) strains is not statistically significant, there is a discernible increase in average levels. This implies that availability to regulate EV production is not completely eliminated. more accurate description of the observed effect might be ‘significantly reduced,’ which would more precisely reflect the experimental data.

Response: We have revised the sentence according to the reviewer's suggestion.

56. Test line 194 A period is missing after “signals”.

Response: Corrected

57. Text line 204 “Additionally, our prior RNA sequencing, which identified the GPCR *str-173* as a regulator of socially mediated exopher production, also highlighted *sri-36* and *sri-39* as top candidates, further supporting their potential as strong candidates for validation⁴⁴.”. The authors should elucidate how their previous RNA sequencing efforts have identified *sri-36* and *sri-39* as top candidates for pathogen signal receptors. It is essential to clearly describe the criteria and analysis methods used to highlight these genes, as well as any supporting data that distinguishes them from other candidates. Providing this detail will enhance the reader's understanding of the basis for these selections and strengthen the overall findings of the study.

Response: We appreciate the reviewer's request for clarification. These GPCRs emerged as top candidates in our prior RNA-seq analysis of socially-mediated exopher production (Szczepańska et al., 2024, Nature Communications), based on their differential expression in solitary worms compared to those maintained at high population density. To avoid overstating their function and clarify our selection criteria, we have made it clear in the text that *sri-36* and *sri-39* were chosen based on their association with exopher regulation, further justifying their selection for validation.

58. Text line 207 “To investigate the roles of *sri-36* and *sri-39* in exopher regulation, we employed CRISPR/Cas9 technology to generate a mutant with a deletion encompassing both receptor-encoding genes, as they are located only 1.9 kb apart, with one gene situated between them (Supplementary Fig. 1).” The propinquity of two gene, *sri-36* and *sri-39*, upon genome accounts for the efficiency of simultaneous deletion to generate double (actually triple) mutant, but it cannot be the ground for not generating single mutants. Use of this mutant deleting multiple genes leaves a possibility that the phenotypes are attributed to only one of the mutated genes, but the rests are not contributing. The authors should discuss this limitation.

Response: We appreciate the reviewer's insight regarding the use of a multi-gene deletion. In the results section we have acknowledge this limitation.

59. Text line 221 (Fig. 6d) and text line 224 (Fig. 6c).

Response: Corrected

60. Figure 7d and Text line 234 “The *wrmScarlet* CRISPR/Cas9-mediated transcriptional knock-in of the *sri-19* gene revealed its specific expression in ADL neurons (Fig. 7d).”.

Figure 7d describes expression of 7TM receptor-encoding genes, *sri-19* in ADL neuron, which is annotated based on position and scRNAseq data from Reference 53. However, the manuscript lacks a description of how the annotation of cells based on position was executed. If promoter-driven fluorescence proteins were used as comparative markers for this annotation, it would be helpful for the authors to include this information.

Response: We thank the reviewer for pointing out the lack of detail on how ADL neurons were annotated in Fig. 7d. To address this, we have added images of *gap-15p::GFP; sri-19::SL2::wrmScarlet* double-reporter line (Supplementary Fig. 7i). Co-localization of GFP and *wrmScarlet* in these animals confirms that *sri-19* is expressed in ADL. This addition now provide a clear, experimentally validated basis for our positional annotation of ADL in Fig. 7d.

61. Text line 240 “Additionally, the expression of the *srr-6* receptor in neurons of the retrovesicular ganglion (RVG), alimentary tract, and excretory system (Fig. 7l), as well as the expression of *sri-36* and *sri-39* in the

alimentary tract (Fig. 7h), suggest that one or more of these tissues may also play a role in regulating exopher production in muscles.”.

Based on the current experimental results, the claim that these tissues are involved in the regulation of exopher production may need to be presented with more caution. Or, providing stronger supporting evidence, such as a rescue experiment, could help substantiate this interpretation.

Response: We thank the reviewer for this important point. To directly test the role of *srr-6* in non-neuronal tissues, we performed an intestinal-specific rescue of *srr-6* in the *srr-6* mutant background and found that restoring *srr-6* expression in the gut is sufficient to rescue exopher upregulation in response to *P. aeruginosa* secretome. These rescue data are now presented in Supplementary Fig. 7j.

62. Text line 247 “*we aimed to investigate the underlying reasons for the release of these extracellular vesicles when worms detect the pathogen in their environment.”.*

The results in figure 8 are deciphering the effect of released exophers, not underlying reason (or cause) of released exophers.

Response: We have revised the sentence to reflect your suggestion.

63. Text line 252 “*To further explore this, ...”.*

It is difficult to follow what “this” is referring to. To improve clarity and ensure the narrative is easily understandable, specifying explicitly what 'this' will be helpful.

Response: Corrected

64. *The authors describe the age synchronization of animals by isolating embryos at the pretzel-stage or 3-fold stage. To enhance readability and accessibility, especially for readers not familiar with C. elegans development, it is recommended to standardize the terminology used to describe these stages, or clarify if they refer to the same developmental stage. Additionally, including a brief description of the method used for isolating pretzel-stage embryos would further aid in understanding.*

Response: We thank the reviewer for this suggestion. We have standardized our terminology throughout to use “3-fold stage” exclusively in place of “pretzel-stage”. In the Materials and Methods, we now include that embryos were collected by hand-picking under a stereomicroscope at the 3-fold stage.

65. *Figure 8b. It is ambiguous in this figure whether “+ PA14” in inlet legend indicates exposure to pathogen volatile before assay, or exposure to pathogen during survival rate assay.*

Response: We have revised the schematic to clearly distinguish parental volatile pre-exposure from the subsequent survival assay conditions.

66. *The explanation provided for Figure 8e and f in the current manuscript is insufficient. Details regarding the treatment of F1 animals during the experiment remain unclear, even with the provided illustration in Figure 8a. Including a more detailed explanation of the experimental procedures would significantly enhance the readers' understanding.*

Response: We have expanded the Materials and Methods to clarify the experimental workflow and ensure full reproducibility.

67. Text line 293 “1. Pathogen-induced exopher production: Exposure to live *Pseudomonas aeruginosa* PA14 or *Serratia marcescens* Db10 induces exopher production in *C. elegans* body wall muscles.”

In Figure 9, this fact is indicated by the green upward arrows, but it should be more stressed because it is the main aspect studied in this work.

Response: Thank you for highlighting this point. We have emphasized “Muscle exophergenesis upregulation” in the model depicted in Figure 10 to ensure it is more prominently featured.

68. Text line 312 “Our findings suggest that, although exophers produced in response to different stimuli may appear similar in number, the distinct signalling pathways involved indicate potential differences in their roles or contents.”

Although the research has thoroughly explored the mechanisms underlying exopher production in response to various pathogen stimuli, there is no positive reason for assuming differences in their roles and contents. Thus, the work does not “indicate” such potential.

Response: We have rephrased this point to reflect that it is a hypothesis arising from our data rather than a conclusion.

69. A reference is required at text line 344 where it states ‘In previous studies.’

Response: Corrected

70. Text line 347 “The involvement of multiple G protein-coupled receptors (GPCRs) in regulating exopher production suggests that exophergenesis is not triggered by a single metabolite but by a complex cocktail of pathogen-derived signals.”

Actually, that might not be the only explanation. There are other possibilities to consider, for example, that the GPCRs are responding to the same substances.

Response: We thank the reviewer for this insightful point. We have added this into the discussion.

71. Text line 354 “exophers and other EVs”

Given that this research has not evaluated other types of EVs, it is challenging to conclude that the other EVs are crucial in mediating the organism’s response to environmental stressors.

Response: Because we did not directly characterize non-exopher EV subtypes in this study, we have revised the text to read “exophers and possibly other EVs” to more accurately reflect our scope.

72. Text line 416 “(references)”
Needs be replaced.

Response: Corrected

Reviewer #3 (Remarks to the Author)

I co-reviewed this manuscript with one of the reviewers who provided the listed reports. This is part of the

Nature Communications initiative to facilitate training in peer review and to provide appropriate recognition for Early Career Researchers who co-review manuscripts.

Response: We sincerely appreciate the reviewer's comments and suggestions.

Reviewer #4 (Remarks to the Author)

In the manuscript by Kołodziejaska et al, the authors track the effects of exposure to pathogenic bacteria on production of muscle-derived giant vesicles (exophers). The authors find that exposure to Pseudomonas and Serratia pathogenic bacteria increases exopher production rate.

72. Effects of signaling pathways that can impact innate immunity were shown, with Insulin-like signaling pathway mutant daf-16 reducing exopher production in all bacterial backgrounds, pathogenic and control. Its negative regulator akt-1 is tested, which might be expected to further increase exopher rates upon exposure to PA14 and Db10, but one cannot tell given the statistics shown. Please add such analysis.

The authors should also acknowledge that these signaling pathways are not exclusive to innate immunity and could mediate exopher effects via other changes to the animals independent from immunity.

tir-1 mutants affecting the p38 MAPK signaling pathway appeared to increase exopher production on OP50 (Fig 1d). It is unclear if tir-1 further increases PA14 and Db10 effects (Fig1f) since the comparison to WT was not statistically tested (see above). In addition, the OP50 result appears different in the two panels. Fig 1d shows tir-1 mutants increasing exopher production on OP50, while Fig 1f appears to show no effect of the same experiment (although the statistical comparison vs WT is not shown). pmk-1 mutants in the same p38 pathway did not affect exopher levels on OP50 or suppress PA14 or Db10 driven increases. The way the comparisons are done does not allow the reader to tell if tir-1 mutants further increase PA14/Db10 effects. Please add the relevant statistical comparisons. There are similar problems with interpretation for the other mutants dbl-1 and cul-6 on PA14 and Db10, as they are only statistically tested against OP50, but not against WT exposed to the same bacteria. This should be tested.

Response: In response to your and Reviewer 1 suggestions, we have performed direct comparisons of the pathogen-induced exopher increase magnitudes for all neuronal ablations, immune-pathway mutants, and GPCR mutants. These effect-size analyses are now presented in Supplementary Fig. 1, 3, 4 and 5 to maintain clarity in the main figures.

The apparent discrepancies between Figures 1d and 1e-g reflect methodological differences between the two assays, which are now described both in the Fig. 1 legend and in the Materials and Methods (see also our response to Reviewer 1, comment 13).

73. Pseudomonas PA14 mediated increases in muscle exopher production could be induced by non-volatile and volatile metabolites, with clear genetic differences in response to these different stimuli. They also differ in neuronal requirements. Volatile exposure to PA14 produced some egg retention (a known exopher inducer) while non-volatile metabolites (conditioned media) did not affect egg retention. The authors should modify lines 127-129 “This indicates that the P. aeruginosa-dependent increase in exopher production operates independently of the previously described embryo-maternal signalling “ accordingly. Fig 1 showed that Db10 exposure also produced egg retention. PA14 and Db10 effects could be further tested using sterile animals to determine if exposure can overcome empty uterus effects.

Response: We thank the reviewer for highlighting this point. Differences in embryo accumulation following exposure to volatile versus non-volatile secretomes are described in the Results section “Pathogen non-volatile and volatile secretomes trigger exopher production” (Fig. 2d and h).

To directly test whether fertility is required for pathogen-induced exophogenesis, we sterilized worms with 5-fluoro-2'-deoxyuridine (FuDR) and then exposed them to either *P. aeruginosa* or *S. marcescens* bacteria. In FuDR-treated animals, exopher production was completely abolished under both conditions (Supplementary Fig. 1c), demonstrating that functional fertility is essential for the exopher response to both pathogens. We have updated the Results to clarify that pathogen-triggered exophogenesis depends on fertility, even though it does not always correlate with increased embryo accumulation.

74. dbl-1 mutant appears to suppress exopher production on OP50 control and partially suppress PA14 volatile effects (Fig 3a). Please add appropriate statistical test for this and modify accordingly line 121: "Therefore, the pathogen-induced EV modulation does not rely on canonical immune response pathways." The effects on OP50 control make overall interpretation for dbl-1 difficult, although some studies suggest that OP50 is itself mildly pathogenic.

The authors test for effects of neuron ablation seeking to gain insight into sensing and propagating effects of bacterial metabolites. ADL, ASK, or AWC neurons were important for PA14 response in volatile and non-volatile assays. There is also some issue with interpretation here because these ablations also appeared to block most exopher production on OP50 control. If the neuron (or neuronal receptor such as npr-9) is required for basal exopher production it is difficult to know if it is directly involved in metabolite sensing and downstream signaling or if it is just generally required for exopher production. Please add statistical tests comparing samples exposed to the same bacteria but with intact vs ablated neurons.

Response: Response: As mentioned earlier, we have performed direct comparisons of the pathogen-induced exopher increase magnitudes for all neuronal ablations, immune-pathway mutants, and GPCR mutants. These effect-size analyses are now presented in Supplementary Fig. 1, 3, 4 and 5 to maintain clarity in the main figures and are discussed in details in Results section.

Our experimental design explicitly separates a neuron's role in constitutive exophogenesis from its function in metabolite/secretome-triggered signalling. By first measuring basal exopher levels in each neuronal-ablation mutant, we establish whether the cell is generally required for exopher production. We then challenge the same mutant strains with volatile metabolites or non-volatile pathogen secretomes and compare their induced responses to wild-type controls. A neuron that only affects basal counts but still shows normal stimulus-dependent upregulation is therefore dispensable for sensing, whereas a neuron whose ablation abolishes the induced increase is directly implicated in metabolite/secretome detection or downstream signalling. This two-step approach allows us to distinguish general exopher machinery components from those specifically required for environmental cue integration.

75. The authors argue for a connection between volatile metabolite exposure, exopher production, and survival of P0 and F1 generations upon full PA14 exposure. They show that P0 exposure to PA14 volatile metabolites exhibited reduced survival of P0's during bacterial infection, inversely correlated with exopher production. They also showed that worms with ablated ASK neurons, that produce fewer exophers than wild type worms, survive PA14 infection longer than wild type. They also showed that knockdown of the RME-2 receptor, which produce very few exophers, also survive P. aeruginosa infection longer than wild-type worms. They also found that sri-19 mutants, which do not show increased exopher production upon contact with non-volatile metabolites, survived significantly longer during infection. These are very interesting results. The authors also found interesting results for the F1 generation, with high P0 exopher generation correlated with F1 animals with improved survival on PA14. Overall interpretation is still somewhat limited for these results as they remain correlations. The authors should test if these perturbations affect embryo retention in the uterus. Effects on reproduction could be controlling exopher level rather than exopher level controlling reproduction. rme-2 is certainly very important for brood size.

Response: Thank you for the suggestion. We quantified embryo accumulation in *sri-19* mutants and in *rme-2* knockdown animals, and both showed reduced embryo retention. These results are presented in Supplementary Fig. 8g and 8i.

76. The paper does not characterize which muscles produce exophers in response to pathogens. Do exophers appear randomly from all body-wall muscle, or is production limited to a particular body region, for instance near the uterus?

Response: We did not specifically map the anatomical origin of pathogen-induced exophers, so in the manuscript we refer broadly to exophers produced by the body wall muscle without regional specification.

77. The authors should more directly compare their results here with those from their recent paper on pheromone-induced exophers.

Response: Thank you for this suggestion. In the current manuscript's Results section, we have now explicitly referred to our previously published data on pheromone-induced exophers to facilitate direct comparison with the current findings. Furthermore, in the Discussion, we have expanded our analysis to draw overarching conclusions about how the nervous system encodes diverse environmental cues, whether bacterial, pathogen-derived metabolites, or conspecific pheromones, to regulate EV production via distinct yet overlapping sensory neuron circuits.

78. Have the authors tested OP50 conditioned media in the non-volatile experiments? It appears they used simple LB as a control rather than control conditioned media.

Response: We have not previously included OP50-conditioned media in the non-volatile assays, as our control plates already contain live OP50 bacteria and thus serve as an appropriate baseline. Instead, we used plain LB as the mock control. Nonetheless, in response to the reviewer's request, we have now tested OP50-conditioned media side-by-side with LB control and observed no difference in EV production. This new control data are presented below:

79. Please check Materials and Methods for accuracy. No micrographs are shown in this paper, but they are discussed in M&M.

Response: In the previous manuscript version, micrographs of GPCR expression in *C. elegans* (Fig. 7d, h, l) were described in the Materials and Methods. We have now also added representative fluorescence images of pathogen-induced exophers (Fig. 2b, f). The revised Materials and Methods section describe all microscopy details to ensure that all presented micrographs are fully supported by detailed methodological descriptions.

80. Fig 1C should show negative regulation of DAF-16 by AKT.

Response: Corrected

81. Why do Figures 1i and 2c seem in conflict?

Response: The apparent discrepancy between Fig. 1i and 2c reflects the fact that they probe embryos accumulation under different cue combinations. Fig. 2c shows number of embryos in the uterus in response to volatile metabolites alone, revealing a volatile-specific accumulation. Fig. 1i, by contrast, measures number of embryos in worms exposed to live *P. aeruginosa*, where both volatile metabolites and non-volatile supernatant signals are present simultaneously. In this combined context, the non-volatile-mediated pathway dominates (number of embryos in non-volatile alone conditions is present in Fig. 2f), effectively “overriding” the volatile signal, so that the overall there is no embryo accumulation in worms grown on live pathogen. In other words, volatile cues alone cause embryo accumulation (Fig. 2c), but when worms experience the full complement of bacterial secretome (as on live pathogen), the non-volatile component sets the response ceiling (Fig. 1i).

82. Fig 8 Legend is overstated: “Exophers levels are affecting survival rate post-infection.” This implies cause-and-effect, which is not demonstrated. A more accurate title would be “Exophers levels correlate or inversely correlate with survival rate post-infection.”

Response: We have revised the sentence according to the reviewer's suggestion.

83. In many experiments the changes emphasized in the interpretations are statistically significant but represent a small change, in some cases a difference in exopher number from 9 to 11 or 7 to 10 is statistically significant while other experiments conclude no significance between similar median comparisons, 11 to 13 (Fig 3B). Which cases are biologically important differences? Overall, the statistics are misleading, graphs and text should indicate mean values for each experiment with standard error of the mean. In this current submission, the authors display interquartile ranges emphasizing the range around the median and are most appropriately used with skewed datasets that contain outliers (while these datasets appear to be normally distributed, with no outliers indicated). Moreover, precise trial numbers and sample sizes are not provided for each experiment (often a range is given) and several experiments in Fig 1 only have two biological replicates rather than the standard minimum of three. It is not clear from the text or figure legends if all data is shown, or if one representative dataset is shown in the main figure. Replicate experiments should be provided in the supplement to allow for data transparency regarding natural variation between experiments. Furthermore, it might be appropriate to evaluate these data as log fold changes rather than comparing averages to more rigorously determine if these small differences are meaningful, the authors should consult a biostatistician to guide them in this analysis.

Response:

We thank the reviewer for highlighting the critical distinction between statistical significance and biological relevance. We fully agree that a p-value below the conventional threshold (i.e., $p < 0.05$) does not inherently guarantee physiological importance, nor does a value marginally above it (e.g., $p = 0.051$) rule out a meaningful biological effect. Striking this balance between statistical rigor and biological insight has, at times, proved challenging: in our experience, more nuanced interpretations were questioned by reviewers when they fell short of strict significance criteria. That pressure led us to rely heavily on statistical significance in the previous version of the manuscript. However, in response to your comments, we have conducted additional analyses to better characterize the magnitude and consistency of exopher changes across all conditions. Moreover, in the revised manuscript, we now explicitly explain which comparisons are supported by robust effect sizes and reproducible trends, and we temper our conclusions where changes, though statistically significant, are modest. These revisions ensure that readers can readily distinguish between findings of clear physiological consequence and those that, while reproducible, may warrant further investigation to establish their *in vivo* importance.

At the beginning of this project, we collaborated with a biostatistician to determine the most appropriate methods for analysing our EV data, and we have adhered strictly to this guidance. Although our datasets may appear roughly symmetric, Shapiro–Wilk and Kolmogorov–Smirnov tests consistently demonstrated that exopher-count distributions deviate from normality. Consequently, we have uniformly applied nonparametric tests across all comparisons: the Mann–Whitney U test for two-group analyses and the Kruskal–Wallis test with Dunn’s post hoc correction for multiple group comparisons. Moreover, because nonparametric tests compare medians rather than means, we believe it is statistically inappropriate, and potentially misleading, to report means \pm SEM for these datasets. Instead, we present medians with interquartile ranges (IQRs), which accurately reflect the central tendency and variability in non-normally distributed data.

We appreciate the reviewer’s concern regarding the reporting of sample sizes and biological replicates. In the manuscript, each figure legend specifies the exact number of animals scored (“n”) and the number of independent experimental repeats (“N”), and we have occasionally condensed these values into ranges for brevity when presenting multi-panel data. Moreover, to facilitate transparent assessment of variability, every individual measurement is plotted, enabling readers to assess natural variation and replicate consistency directly from the graphs. We deliberately present raw exopher counts rather than fold-change values in order to avoid introducing potentially misleading normalization and to maintain direct comparability between baseline and treatment conditions. Finally, the complete dataset, including all individual replicate values, has been deposited in an open-access repository, avoiding the need for an large supplement while still providing full access to the underlying data. Whenever an image or dataset serves as a representative example, this is explicitly noted in the corresponding legend. We believe that this combination of detailed legend annotation, comprehensive data visualization, and public data availability achieves maximal clarity and rigor without overburdening the manuscript or its supplementary materials.

We appreciate the reviewer’s attention to biological replicate number. The experiments in Fig. 1 were performed as part of a scientific exchange in Dr. Pujol’s laboratory in Marseille, where, due to time constraints, we were able to complete two independent biological replicates. In each case, the replicate datasets exhibited highly consistent phenotypes, providing us with confidence in the robustness of these initial observations. Importantly, Fig.1 merely establishes the foundational increase in exopher numbers upon direct infection with *P. aeruginosa*, which serves as the launching point for our subsequent, more detailed analyses. Throughout the rest of the manuscript, where we dissect the effects of pathogen-derived metabolites/secretomes on EV production, we have performed a minimum of three biological replicates for every experiment. Moreover, to ensure scientific rigor, the majority of our experiments were conducted in a blinded manner (as detailed in the Materials and Methods), and many datasets were independently collected by two researchers to minimize bias and to confirm reproducibility. Thus, we believe that the central conclusions of our

study rest on sufficiently powered datasets and accurately reflect the phenomenon under investigation.

84. *Overstatement: line 265-266 “Given that muscle exophers have been shown to improve animal reproduction and boost offspring development”. Previous work showed a correlation, not a cause-and-effect relationship.*

Response: We have revised the sentence according to the reviewer's suggestion.

85. *The model at the end of Results should be in the Discussion.*

Response: Thank you for this recommendation. We have relocated our schematic model from the end of the Results section into the Discussion, where it now serves as a conceptual synthesis of our findings within the broader interpretive context.

86. *Several mutants in the TGF-Beta pathway have organismal-wide phenotypes including small body size (DBL-1). Data examining this mutant for exophers should be normalized to account for the difference in body size between the WT and dbl-1(nk3) animals.*

Response: We thank the reviewer for highlighting the size difference in *dbl-1(nk3)* mutants. Because our analysis focuses on the relative change in exopher production upon metabolite exposure, each genotype being compared to its own unexposed control, it inherently normalizes to the basal exopher level of that strain.

87. *Additional controls should be performed to address the effect of dietary restriction on muscle exophers (a known effector of egg retention and exophers in other tissues). Both volatile and non-volatile metabolites could have a profound impact on feeding which in turn would muddy the conclusions from these experiments and need to be addressed experimentally.*

Response: We thank the reviewer for this important suggestion. To rule out an indirect effect of dietary restriction, we measured pharyngeal pumping rates in worms exposed to both volatile and non-volatile *P. aeruginosa* secretomes. As shown in Supplementary Fig. 2b and e, neither treatment altered pumping frequency compared to control animals. These data demonstrate that feeding behaviour remains unchanged under our assay conditions and support our conclusion that the observed changes in muscle exopher production are driven directly by chemosensory signalling rather than by altered food intake.

88. *Summary tables for P-values are nice. They could be improved by including the WT control alongside experimental results (Fig 5D).*

Response: We have included wild-type controls alongside the experimental results in Figs. 3, 5, and 6.

89. *Fig 7e – The legend states worms lacking SRI-36 and SRI-39 receptors are comparable to WT but data shown reports a significant difference (p=0.0016) between WT and sri-36/39(wwa57) mutants. This result is at odds with both the main text and the figure legend.*

Response: We thank the reviewer for pointing out this discrepancy. To resolve it, we increased the number of biological replicates for both the *sri-36/39(wwa57)* and the independent *sri-36/39(wwa58)* alleles. In the expanded dataset, neither double mutant differs significantly from wild-type, and

results are now fully consistent across both receptor-null strains. The updated data are shown in Figure 8e and described accordingly in the main text and legend.

Reviewer #5 (Remarks to the Author):

The manuscript provides valuable insights into extracellular vesicle (EV) production in response to pathogen-derived metabolites. It highlights a regulatory network influenced by volatile and non-volatile metabolites, with implications for host-pathogen signaling and innate immunity. While the study is innovative and relevant, several areas require clarification and additional data to strengthen its impact.

90. *The study mentions key immune pathways such as insulin, TGF β , p38 MAPK, and IPR, but excludes other established regulators like SKN-1, ELT-2, and HLH-30. Exploring their roles could enhance the analysis and provide a more comprehensive understanding of the mechanisms driving EV production. The selection criteria for these pathways should be justified.*

Response: We agree that SKN-1, ELT-2, and HLH-30 play pivotal roles in *C. elegans* stress and immune responses, but their broad regulatory scopes complicate interpretation in the context of exopher biogenesis. For instance, SKN-1 activates dozens of phase II detoxification genes in response to oxidative stress and xenobiotics^{1, 2}; perturbing SKN-1 therefore disrupts global redox homeostasis and defense pathways. ELT-2 is the major transcription factor required for *C. elegans* intestinal development, therefore, controls more processes than just immune response³. HLH-30 governs autophagy, lysosomal biogenesis, and metabolic adaptation under starvation and infection, coordinating nutrient sensing with proteostatic and longevity programs⁴. By contrast, the insulin, TGF- β , p38 MAPK, and IPR pathways each play evolutionary conserved roles in the regulation of the immunity, enabling clearer formulation of conclusions and mechanistic dissection. We have therefore focused on these axes in the present study.

¹ <https://doi.org/10.1073/pnas.0508105102>

² <https://doi.org/10.1016/j.freeradbiomed.2015.06.008>

³ <https://doi.org/10.1093/genetics/iyad088>

⁴ <https://doi.org/10.1038/ncomms3267>

91. *Figures 1D and 1F show inconsistent trends in EV production among mutants, such as tir-1(tm3036) and dbl-1(nk3). These discrepancies need clarification to ensure the validity of the conclusions drawn from these experiments. Graphical improvements, such as merging related panels or adding clearer annotations, would improve readability.*

Response: We appreciate the reviewer's careful comparison of Figures 1d and 1e – g. These panels were generated under slightly different conditions in two collaborating laboratories, Dr. Pujol's and Dr. Turek's laboratories, which accounts for the observed baseline discrepancies. Although the same researchers performed these assays in both laboratories, slight methodological differences account for the apparent baseline discrepancies. In Fig. 1e – g, worms, and their OP50 controls, were subjected to an additional 3-hour incubation at 25 °C during the infection window, whereas all data in Fig. 1d were collected at our standard 20 °C. Furthermore, Fig. 1e – g exophers were scored on worms immobilized on agar pads for imaging under Dr. Pujol's microscope, and we have found that immobilization can mechanically perturb fragile exophers and influence exopher counts (possibly in

a genotype-dependent manner). In contrast, all other experiments involving muscle exophers presented in this manuscript, including those in Fig. 1d, were conducted on freely moving animals. Critically, despite these differences in absolute numbers, the relative increases in exopher production upon pathogen exposure remain highly reproducible across both laboratories and two distinct pathogens (*P. aeruginosa* and *S. marcescens*). Each experiment includes internal OP50 controls, ensuring that our conclusions are based on fold-change induction rather than raw baseline values. Thus, while procedural variations affect baseline exopher counts, they do not compromise the robust, pathogen-specific upregulation observed in immune-pathway mutants.

To clarify the baseline differences between Figures 1d and 1e–g, we've added a note to the Fig. 1 legend:

“Baseline differences between panels d and e – g arise from methodological variations which are detailed in the Materials and Methods section “Scoring of Exophers and Fluorescence Microscopy”

In the Materials and Methods section, we explicitly describe the differing incubation temperatures, durations, and worm immobilization steps used to collect each dataset.

We appreciate the suggestion to improve figure readability by merging related panels. However, each mutant dataset uses its own OP50 control, reflecting the fact that different genes are located on the same chromosomes as our exopher reporter integrants and thus require independent strains, therefore merging panels would obscure these critical genotype-specific baselines. Instead, we have enhanced clarity by adding labels beneath each panel to group genes by their respective immune pathways.

92. *Tissue-specific roles of key regulators like NPR-9 and GPCRs (e.g., SRI-19, SRI-36, SRI-39, and SRR-6) in immune activation and exopher production should be further investigated. Rescue experiments could identify their tissue-specific functions, providing insights into their biological significance.*

Response: Thank you for this suggestion. For NPR-9, its site of action is already well established: it is reported to be exclusively expressed in AIB^{1,2}, and our data (AIB ablation, *npr-9* mutants, and AIB optogenetics) are consistent with an AIB-mediated mechanism, so we did not pursue additional tissue mapping. For SRR-6, we performed an intestinal rescue, which restored the non-volatile secretome response and lowered baseline exophers; these data are included in the manuscript (Supplementary Fig. 7j).

For SRI-19 array-based ADL rescues proved difficult to interpret due to overexpression/ mosaicism (see response to Reviewer #1, comment #5; notably, ADL-specific *sri-19* rescue exacerbated the phenotype). We did not pursue additional rescues; instead, as already presented in the manuscript, we generated and analyzed two independent, backcrossed CRISPR alleles for each GPCR and drew conclusions only when phenotypes were the same across both. On this basis, we are confident that the implicated GPCRs indeed regulate exophergenesis.

¹ <https://doi.org/10.1073/pnas.0709492105>

² <https://doi.org/10.1371/journal.pgen.1006050>

93. *While the study demonstrates offspring survival benefits linked to increased maternal exopher production, it does not explore these effects beyond the F1 generation. The authors might want to investigate potential intergenerational transmission into the F2 generation.*

Response: We agree that transmission to F2 generation is an intriguing question. We performed preliminary F2 assays using several protocols (varying the timing of exposing worms to pathogen infection) but in every case the enhanced resistance and developmental effects we saw in F1 were undetectable in F2. Given the extensive optimization required, we feel that a full thorough testing of

intergenerational persistence is beyond the scope of this manuscript. Therefore, we focus here on the robust, reproducible F1 effects.

94. The specificity of the observed EV responses to pathogens versus general stressors remains unclear. Including controls for non-pathogenic stressors would help establish whether the responses are pathogen-specific.

Response: In our earlier work (Turek et al., 2021), which we cite in our manuscript, we tested non-pathogenic stressors, including mild heat shock (33 °C for 60 min) and oxidative stress (5 mM hydrogen peroxide solution). Because for our experiments we always have internal controls (e.g. non-exposed worms) we are confident that the observed phenotype is specific to pathogen-derived cues rather than to a general stress response.

Minor Comments

95. The Introduction contains some redundancy between the second and third paragraphs, particularly in discussing the advantages of using C. elegans as a model organism. Streamlining this section would enhance conciseness.

Response: Corrected

96. Several references lack complete citation details, such as volume, page numbers, or correct publication years (e.g., reference #3 and #42).

Response: Corrected

97. Minor typographical errors, such as missing punctuation on line 194, should be corrected to improve overall readability.

Response: Corrected

Reviewer #6 (Remarks to the Author)

I co-reviewed this manuscript with one of the reviewers who provided the listed reports. This is part of the Nature Communications initiative to facilitate training in peer review and to provide appropriate recognition for Early Career Researchers who co-review manuscripts

Response: We sincerely appreciate the reviewer's comments and suggestions.

Additional data provided after the revision and not requested by the reviewers:

In the updated manuscript, we have voluntarily incorporated new findings that were not specifically requested by the reviewers. These results demonstrate that the AIA interneuron partially mediates the volatile metabolites–induced increase in exopher production and is necessary for the response to non-volatile metabolites (Fig. 6d, e, f). In addition, we examined the role of the ASJ sensory neuron and found that it does not contribute to exopher regulation (Fig. 5 a, b, c). Finally, using an untargeted LC-MS metabolomics approach, we identified the tripeptide Ile-Pro-Pro as one of the components of the bacterial supernatant responsible for promoting exopher production (Fig. 4a, b, c, d). Together,

these findings provide deeper mechanistic insight and contribute to a more comprehensive understanding of exophogenesis is regulated.

We sincerely thank the reviewers for their constructive comments, which have greatly improved our manuscript, and we are pleased to present this revised version.

Please find a detailed description of the edited paragraphs below (the reviewers' comments are in *italics* and our responses are in blue font):

Reviewer #1 (Remarks to the Author):

The authors have made tremendous efforts during the revision process to address the questions raised by the reviewers, and overall they have successfully addressed most of my concerns. However, several minor points still need to be revised before publication:

1. I agree with the other reviewer that representative images of muscle-derived exophers should be included in Figure 1.

Response: Thank you for this valuable suggestion. In response, we have added an example of a muscle-derived exopher produced by *C. elegans* in Fig. 1a.

2. The authors should clarify how they confirmed that sri-19 is expressed in ADL neurons, and sri-36/39 in ASI and SKI neurons, in the absence of specific markers.

3. Regarding my previous question (#5) on rescuing SRI-19 in ADL neurons: although the authors attempted rescue experiments, the results were unexpected. I appreciate that they have already made an effort, but since the evidence for SRI-19 function in ADL neurons remains inconclusive, the authors should either (i) use an ADL reporter to demonstrate co-localization with wrmScarlet–SRI-19 fusion protein, or (ii) confirm differential wrmScarlet–SRI-19 expression in wild-type animals with varying exopher numbers.

Response: We thank the reviewer for these comments. As indicated in the previous version of the manuscript we confirmed *sri-19* expression in ADL neurons experimentally: the SRI-19::wrmScarlet fusion protein co-localizes with the gap-15p::GFP reporter, which marks ADL neurons (Supplementary Fig. 7i). Moreover, single-cell RNA-seq data from CeNGEN (CeNGENApp) show that *sri-19* is predominantly expressed in ADL sensory neurons, with additional expression detected in PHA and URY neurons (Supplementary Fig. 7h). Together, these data support *sri-19* expression in ADL neurons and strengthen our conclusion regarding its neuronal specificity.

Regarding *sri-36* and *sri-39* expression, our conclusions are based on previously published data (Vidal et al., PLOS Biology, 2018, "An atlas of *Caenorhabditis elegans* chemoreceptor expression") and on the anatomical position of the corresponding cells within the *C. elegans* nervous system, as described in the legend of Fig. 8h.

4. In Figure 3b (middle panel), there is a clear trend toward increased exopher production in pmk-1 mutants upon PA14 non-volatile excretome treatment, although the difference is not statistically significant. The authors should increase the sample size (N) to clarify this effect.

Response: We thank the reviewer for this comment. We have repeated the experiment, and the observed effect is now statistically significant. The updated data and corresponding statistical analysis are included in the revised version of the manuscript.

5. *Baseline levels appear inconsistent across figures. For example, in Figure 6f and 6g, the wild-type values differ markedly (~8 vs. ~2). A similar issue is present in Figure 5a, where the wild-type values vary across panels. The authors should repeat the experiments or reasonably address these discrepancies.*

Response: We appreciate the reviewer's careful examination of our manuscript. The observed differences in baseline wild-type values arise from the use of different transgenes for exopher visualization. In this study, we employed multiple mutants for genes located on various chromosomes, as well as lines used for genetic ablation, each driven by distinct integrated transgenes that are also localized on different chromosomes. Consequently, it was not always possible to use our standard combination of transgenes that drive wrmScarlet expression in muscles and GFP on the mitochondrial outer membrane (wac1s1 and wac1s14, respectively).

Data presented in Fig. 6f and Fig. 5a (left and middle panels) were obtained from worms carrying both wac1s1 and wac1s14, and these wild-type values are consistent across figures. In contrast, Fig. 6g and Fig. 5a (right panel) show data from worms carrying only wac1s1, which typically produce fewer exophers than those additionally expressing mitochondrial GFP (wac1s14 background). Finally, the data shown in Fig. 6a–c come from worms with muscle RFP expression driven by a different transgene (wac1s6), which is brighter than wac1s1 and thus leads to a higher exopher count in animals with wac1s6 and wac1s14 together.

In summary, the variability in wild-type baselines reflects differences in the transgenes used for the visualization of exophers. Due to these intrinsic differences, each dataset includes its own internal controls, and we do not merge panels or combine data from strains with distinct transgenic backgrounds (e.g., Fig. 6a–c vs. d–f vs. g–j). To ensure full transparency, we have added a statement to the relevant figure legends clarifying that the observed baseline differences arise from the use of different visualization transgenes.

6. *The lack of an additive effect between volatile and non-volatile metabolites should be discussed explicitly.*

Response: We appreciate the reviewer's suggestion. This point is now addressed and discussed in the revised Discussion section.

7. *I agree with the other reviewer that a brief description of the genetic ablation methods should be included in the Methods section, even if a citation is provided.*

Response: Thank you for this suggestion. In the revised version of the manuscript, we have added to Supplementary Table 2, which lists the *C. elegans* strains, a description of the methods used for the genetic ablation of the investigated neurons.

8. *I recommend revising the title. Since the study includes both volatile and non-volatile metabolites, the title should reflect both.*

Response: We have revised the title accordingly to reflect that the study includes both volatile and non-volatile pathogen compounds.

9. *I also recommend revising the abstract. The current version is overly long and complex, and a more concise version would improve clarity and impact.*

Response: We have revised the abstract to make it more concise and focused, improving its clarity and overall impact.

Reviewer #2 (Remarks to the Author)

Upon revision, authors added considerable amount of new data and new figures, which is admirable along with the efforts already put in the original submission. On the contrary, it became clearer through one round of review and revision that authors have a strong attitude to promote their hypothesis, or belief, in a biased way and "cherry-pick" the results to support the hypothesis as described below.

Response: We thank the reviewer for the thorough evaluation of our revised manuscript and for recognising the additional data provided. We respectfully disagree, however, with the statement that we are "cherry-picking" results to support our hypothesis. As detailed below, the main basis for this concern appears to be the variability in baseline values observed across experiments. This difference, however, has a straightforward explanation: it arises from the use of different transgenes for exopher visualization, as described in detail in our response below. Furthermore, hypotheses are inherently developed based on the available data, and during the previous revision, we added a substantial amount of new data that allowed us to refine and adjust our hypothesis to more accurately reflect the full scope of our findings.

10. In response to our previous comment 13, authors provided a transparent explanation about methodological differences between Figure 1d and Figure 1e-g, which can reasonably account for the mutant phenotype discrepancies between these figures. However, a closer look at the data within the main text raises additional concerns.

Question 1: How can baseline exopher production be compared between strains?

Authors explained in their rebuttal as follows:

"In Fig. 1e – g, worms, and their OP50 controls, were subjected to an additional 3 hour incubation at 25 °C during the infection window....", "Furthermore, Fig. 1e – g exophers were scored on worms immobilized on agar pads for imaging under Dr. Pujol's microscope,....", "all other experiments involving muscle exophers presented in this manuscript, including those in Fig. 1d, were conducted on freely moving animals."

Assuming all experiments except Fig. 1e-g were conducted under the same conditions, as also described in Methods, we note that even in the wild type, baseline data (unexposed control, namely exposed to OP50 alone) are variable between experiments if we compare across all figures (for instance Fig. 5a, b, right panel, Fig. 6a, g, i, j, etc).

This raises a fundamental question: What is the source of this variability, and which parts of the results are reproducible and trustable? If the main source of variability is a day-to-day fluctuation, the next question is whether baseline exopher production can be reproducibly compared between different strains when the experiments are conducted on the same days.

If we look at the unexposed controls in Fig. 3, daf-16, for example, do not look different from wild type in Fig. 3a/b and tir-1 in Fig. 3a do not look different from wild type. On the other hand, akt-1 looks very different in both figures. These results are not consistent with Fig. 1d, where daf-16 and tir-1 are significantly different from wild type but akt-1 is not. Authors describe N=3 for Fig. 3a, and N=3-4 for Fig. 3b, implying not all experiments in Fig. 3 were conducted on the same days. Then how does the baseline data look like if only experiments conducted on the same days are compared between different strains? Does it still fail to reproduce Fig. 1d?

Upon revision, Fig. 8e was replaced from the previous version (Fig. 7e). The data points themselves for all three lines appear to have been altered or added, and the sample size has changed from "n = 90, N = 3 independent experiments" to "n = 90-180, N = 3-6 independent experiments", implying authors conducted

additional experiments, and the results of the statistical test changed. This also illustrates the variability between experiments and its effect on the conclusion.

In summary, if inter-strain comparison between strains is difficult even when experiments are conducted on the same days, the only solution will be to make judgements from sufficient numbers of independent experiments. Considering the above, $N=3$ does not seem enough for these tests.

Response: The observed differences in baseline wild-type values arise from the use of different transgenes for exopher visualization. In this study, we employed multiple mutants for genes located on various chromosomes, as well as lines used for genetic ablation, each driven by distinct integrated transgenes that are also localized on different chromosomes. Consequently, it was not always possible to use our standard combination of transgenes that drive wrmScarlet expression in muscles and GFP on the mitochondrial outer membrane (wac1s1 and wac1s14, respectively).

Data presented in Fig. 6f and Fig. 5a (left and middle panels) were obtained from worms carrying both wac1s1 and wac1s14, and these wild-type values are consistent across figures. In contrast, Fig. 6g and Fig. 5a (right panel) show data from worms carrying only wac1s1, which typically produce fewer exophers than those additionally expressing mitochondrial GFP (wac1s14 background). Finally, the data shown in Fig. 6a–c come from worms with muscle RFP expression driven by a different transgene (wac1s6), which is brighter than wac1s1 and thus leads to a higher exopher count in animals with wac1s6 and wac1s14 together.

In summary, the variability in wild-type baselines reflects differences in the transgenes used for the visualization of exophers. Due to these intrinsic differences, each dataset includes its own internal controls, and we do not merge panels or combine data from strains with distinct transgenic backgrounds (e.g., Fig. 6a–c vs. d–f vs. g–j). To ensure full transparency, we have added a statement to the relevant figure legends clarifying that the observed baseline differences arise from the use of different visualization transgenes.

II. Question 2: How can fold-change induction in exopher production be compared between strains?

Authors commented in their rebuttal that "despite these differences in absolute numbers, the relative increases in exopher production upon pathogen exposure remain highly reproducible", and that "each experiment includes internal E. coli OP50 controls, ensuring that our conclusions are based on fold change induction rather than raw baseline values".

This may be true, but would authors be able to show numerically that fold-change is more stable than baseline, for example by combining all wild type data?

Response: As stated earlier, we cannot combine all wild-type data because different transgenic lines were used as wild-type controls.

If this claim holds true, one option would be to avoid comparing the baseline between strains and focus on comparing the fold-change induction. Further, if the fold-change comparison between unexposed/exposed pairs that stem from the same culture turns out most stable, authors can change the statistical tests so as to compare the (same-culture) fold-change values between strains.

Response: We thank the reviewer for this suggestion. As explained in the previous rebuttal letter, we deliberately present raw exopher counts rather than fold-change values to avoid potentially misleading normalization and to maintain direct comparability between baseline and treatment conditions. It is easy to estimate the fold-change when raw data are presented, but it is impossible to determine the original values when only fold-change data are shown. Moreover, because all individual measurements are plotted, this allows readers to directly assess natural variation and replicate consistency from the graphs, thereby ensuring transparent evaluation of variability within our dataset.

Adopting this option will also necessitate the removal of a number of figures and claims. For example, comparisons of baseline values such as Fig. 1d and Fig. 8a, e, i, won't be attempted and deleted from the figures, because, as discussed above, these data are not reliable without sufficient N. Upon revision, authors added Supplementary Figs. 1a, b, 3, 4, 5 and 7, in which they extracted only volatile/nonvolatile-exposed data from Figs. 1, 3, 5, 6 and 8, respectively, and compared them statistically. Based on this, authors made new claims such as involvement of sri-19 in the volatile signaling. However, if authors admit that baseline is variable and only fold-change is reliable, comparisons between induced values alone are not meaningful. Accordingly, in this case Supplementary Figs. 1a, b, 3, 4, 5 7 and their related statements (such as asserted modulatory roles of pmk-1, akt-1 and AQR/PQR/URX) need to be removed.

Response: These comparisons, presented in Supplementary Figs. 1a–b, 3, 4, 5, and 7, were included in response to Reviewer 4's explicit request. The reviewer asked for these additional analyses to highlight and discuss even subtle changes or modulations observed in the data. We acknowledge that baseline levels differ between the various transgenes used for exopher visualization; however, within a given transgene background, the baseline remains consistent. Therefore, comparing induced values between strains carrying the same visualization transgene is both valid and meaningful, which is precisely the approach applied in Supplementary Figs. 1a–b, 3, 4, 5, and 7.

As a summary, it is important to evaluate which aspect of the exopher induction by the pathogen-derived stimulus is numerically reproducible. Based on the evaluation, authors need to report only highly stable and reproducible data and base their discussion on them.

Response: We appreciate the reviewer's emphasis on data reproducibility. We confirm that all aspects of exopher induction by pathogen-derived stimuli presented in the manuscript are numerically reproducible. All of the data that we are presenting here has been consistently replicated across independent experiments.

12. Does the P0-F1 trade off exist?

In response to multiple review comments regarding previous Fig. 8, authors replaced the results with those from a new set of experiments. At 20°C (Fig. 9), the data showed that only volatile metabolites improve F1 survival in an SRI-19-dependent manner, with only a "slight nonsignificant decrease" in P0 survival. In contrast, at 25°C (Supplementary Fig. 8), it is shown that volatile metabolites have no effect on F1 survival but reduce P0 survival. Additionally, non-volatile secretomes reduced F1 survival but had no effect on P0 survival at 25°C. Based on the data, the direct correlation between P0 and F1 fitness, which is a key premise of the paper's overarching narrative, appears to be nonexistent. The data do not consistently show that increased exopher production strengthens offspring pathogen resistance, nor do they consistently show that it weakens it. Instead, the effects are highly dependent on specific conditions (volatile vs. non-volatile cues and temperature), with no clear trade-off between maternal sacrifice and offspring fitness. It is necessary to thoroughly revise the manuscript, for example the language in the abstract such as "significantly boosts offspring fitness... at the cost of maternal survival..." and "to optimize survival across generations", to reflect these results accurately.

Response: We appreciate the reviewer's careful evaluation of the manuscript and agree that, with the expanded dataset presented in the previous version, it became evident that the increase in offspring survival depends on multiple factors and is not a universal outcome. Moreover, we do not observe a clear trade-off between maternal sacrifice and offspring fitness. Therefore, in the revised version of the manuscript, we have removed statements suggesting such relationships and replaced them with wording that more accurately reflects our results.

13. Is exopher involved in the regulation of pathogen resistance?

The inconsistencies also affect the interpretation of functional roles of exophers. At 20°C (Fig. 9), the data showed that only volatile metabolites improve F1 survival in an SRI-19-dependent manner. This finding is interpreted by the authors as evidence for the functional role of exopher induction in improving F1 survival. However, as SRI-19 protein structure and expression pattern suggested, it is likely to be a sensory receptor or its modifier. Therefore, the F1 survival effect could be mediated through SRI-19-dependent signaling pathway that operates independently of exopher production.

The authors' claims are further undermined by inconsistencies in their experimental design and data interpretation. They state that the inability of exopher upregulation due to a lack of sri-19 is related to reduced embryo accumulation and prolonged survival upon exposure to PA14 non-volatile secretomes at 25°C (Text line 424-, line 433-). However, the exopher phenotype for sri-19 mutants under this 25°C condition was never actually observed. The authors instead rely on data from a different temperature (20°C) to support their claim. This conclusion cannot be drawn from the results provided, as an increased temperature not only changes the response of *C. elegans* to *P. aeruginosa* infection but also enhances exopher production. Therefore, using results from a different temperature for this assertion is logically flawed. In addition, only volatile metabolite improves F1 survival, while sri-19's involvement in the regulation of exopher induction in response to the volatile metabolites, newly asserted based on Supplementary Fig. 7a, is based on inappropriate comparisons as discussed in point 1 above (ignoring baseline instability; exophers are still increased by volatile metabolites, Fig. 8b).

Authors presented data showing that unlike wild type, *rme-2* P0 becomes more resistant to PA14 pathogen infection. This could be a direct consequence of reduced number of embryos but not necessarily due to the lack of exophers.

Overall, evidence on the biological role of exopher is very weak, and overstatement on it should be carefully avoided.

Response: We agree with the reviewer that our current data indicate only a correlation between the number of exophers and improved survival of the next generation, without providing direct evidence that exophers themselves mediate this effect. Although, we observed that both phenomena share a common regulatory element, the GPCR SRI-19. In the revised version of the manuscript, we have clearly stated this distinction and addressed it in the Discussion to avoid any overstatement of our findings.

Minor comments:

14. Regarding the response to comment 12, unlike the authors' response, the limitation is not discussed in the Discussion section. On the contrary, they rather emphasize the effect of the non-volatile secretome and volatile metabolites in delivering accurate information of pathogen proximity in Text line 445. Acknowledging the limitations is required for clarity. Authors only showed that volatile metabolites and non-volatile secretome are sensed through different molecular pathways to increase muscle exopher production. Let alone it is unknown whether the contents of these exophers are different, it is unclear whether these different cues cause different consequences through the exophers (for example regarding pathogen tolerance of P0 and F1 as discussed above). The text at line 100 "Our research demonstrates that worms can differentiate the proximity of a pathogen threat based on the type of compound they detect", which remains unchanged from the original manuscript, is clearly an overstatement; the presented results DO NOT show "worms can differentiate the proximity". This and related statements need to be moved to Discussion.

Response: In the revised version of the manuscript, we have modified the title and moved the corresponding interpretative statements to the Discussion section as suggested by the reviewer.

15. Response to comment 17. While the revised text on lines 215-219 acknowledges the potential for redundancy or compensation, the subsequent statement on lines 219-223 appears to contradict this, strongly suggesting that the insulin signaling pathway is not involved in exopher production induced by non-volatile

secretomes. Also, the authors claim that “non volatile secretome driven upregulation selectively requires TGF β , p38 MAPK, and IPR pathways” on Text line 230-231. These conclusions are still not supported by the provided data. At the minimum, "by itself" need to be inserted after "insulin pathway does not" at line 221, and "selectively" at line 231 need to be removed.

Response: Corrected.

16. Response to comment 43 and Text line 542. Addition of the data upon revision on neuronal exopher is informative. However, the manuscript's narrow focus on exophers makes it difficult to support the broader claim regarding a possible hormetic role of EVs in general. The discussion is required to be limited to the specific findings on exophers.

Response: We thank the reviewer for this comment. We have removed the section discussing the potential hormetic role of EVs in general and have revised the text to focus exclusively on our findings related to exophers.

17. Figure 2b. Please describe how the authors identified the region of coelomocytes.

Response: Thank you for this suggestion. We have updated the figure legend to clarify that coelomocytes were identified based on their characteristic morphology and anatomical position within the worm body.

18. Text line 487-495. This paragraph of the Discussion section presents a broad conclusion about the principles of sensory neuroscience in *C. elegans*, referring to concepts such as "combinatorial coding" and the interpretation of "unique activation patterns across multiple neurons". However, these conclusions are not supported by the data presented in the manuscript. The study does not include any experiments that directly measure neuronal activity or the circuit relationships that process the sensory information.

Response: Thank you for pointing this out. We did not intend to draw new conclusions about the principles of sensory neuroscience in *C. elegans*, but rather to provide a possible explanation, based on existing literature, for how two distinct sensory inputs might converge into a single effector process. In the previous version of the manuscript, we accidentally missed adding the reference supporting this explanation – Lin et al. "Functional imaging and quantification of multineuronal olfactory responses in *C. elegans*". We have now included the appropriate citation and revised the phrasing in the Discussion to clearly indicate that this is a proposed interpretation, not a novel conclusion about *C. elegans* neurobiology.

19. Text line 499-502. The authors state that this research addresses the ASK neuron's participation in a pathway related to maternal survival upon pathogen exposure. However, the data demonstrating effect of ASK removal in maternal survival upon pathogen exposure (previous Fig. 8b) does not exist in the current manuscript. Thus, this could not be discussed in the current manuscript.

Response: Thank you for pointing this out. We have removed this section from the Discussion in the revised manuscript.

Reviewer #3 (Remarks to the Author):

Response: We sincerely appreciate the reviewer's comments and suggestions.

Reviewer #4 (Remarks to the Author):

My concerns have been well addressed.

Response: *We sincerely appreciate the reviewer's comments and suggestions.*

Reviewer #5 (Remarks to the Author):

The revision addresses several of my points, including clarifying figure inconsistencies, adding tissue-specific rescue data, and correcting the writing and references. I appreciate these efforts. Two concerns remain:

20. *The omission of key immune regulators (SKN-1, ELT-2, HLH-30) without justification*

Response: *Thank you for highlighting this. We have now addressed this point in the manuscript by adding a paragraph to the Discussion that acknowledges and explains this limitation.*

21. *The lack of testing beyond the F1 generation.*

Response: *We thank the reviewer for this comment. We have now tested the effect beyond the F1 generation and did not observe any significant changes in subsequent generations. The corresponding results are presented in Supplementary Fig. 8.*

Reviewer #6 (Remarks to the Author):

Response: *We sincerely appreciate the reviewer's comments and suggestions.*

Below, please find a detailed description of the edited paragraphs below (the reviewers' comments are in *italics* and our responses are in **blue font**):

Reviewer #1 (Remarks to the Author):

The authors have satisfactorily addressed all of my concerns. I have no further comments, and I recommend acceptance of the manuscript.

Response: We are pleased that the Reviewer concerns have been satisfactorily addressed and appreciate the support for our manuscript's acceptance for publication.

Reviewer #2 (Remarks to the Author):

The authors have addressed many of the points raised in the previous review, particularly regarding the inconsistencies and textual corrections. However, two concerns remain:

1) Response to comment 10. The clarification regarding the use of different transgenes for exopher visualization addresses concerns regarding the baseline variability observed in Figures 5 and 6. However, the concern regarding Figure 3 remains unresolved. Since the mutant strains in Figure 3 (excluding pmk-1 and cul-6) utilize the same transgene background as those in Figure 1d (wacIs1 and wacIs14), they should be directly comparable. The relationship of baseline between wildtypes and the mutants differs clearly between Figure 1d and Figure 3a left panel (e.g., daf-16, tir-1 and akt-1). The provided explanation about different transgenes does not account for this specific inconsistency. Although it is the authors' full responsibility to deal with this issue, as a possible option, modifying "all" to "some" at line 114-115 could more accurately reflect the complexity of the results.

Response: We thank the Reviewer for this helpful comment. Following the Reviewer's suggestion, we have revised the sentence describing the data in Figure 1e to more accurately reflect the complexity of our results. The updated sentence now reads: "In some of the tested immune response pathways, we observed altered levels of exophogenesis, either decreased or increased, compared to wild-type worms."

2) Regarding Figure 1a, while the inclusion of representative images helps readers visualize the muscle-derived exophers, the current figure composition is difficult to interpret. Specifically, the fourth and fifth panels appear distinct from the first three. However, the manuscript and figure legend lack a description explaining their significance or how to interpret the data, particularly the heatmap in the fifth panel. Adding a clear explanation of what these specific panels represent and what they are intended to convey would significantly improve reader understanding.

Response: We thank the Reviewer for drawing our attention to this point. To improve clarity, we have expanded the figure legend for Figure 1a to provide a detailed explanation of each panel and its relevance. Specifically, we now clarify that all five panels originate from the same animal and were generated from a single confocal z-stack. Panels (i-iii) present single focal plane of differential interference contrast, overview fluorescence, and magnified views of mitochondria-containing

exophers, respectively. Panels (iv) and (v) correspond to a 3D surface rendering and a depth-coded heatmap of the same z-stack, generated to visualize the spatial arrangement and extracellular topology of exophers. The colour scale in the heatmap denotes depth within the volume, with cooler colours indicating structures closer to the imaging plane and warmer colours representing deeper regions. These additions to the figure legend now explain the purpose and interpretation of each panel, improving figure readability and overall comprehension.

We sincerely thank the reviewers for their constructive comments.